# Binary Classification under Local Label Differential Privacy Using Randomized Response Mechanisms

**Shirong Xu**                                                                          *shirong@stat.ucla.edu*
*Department of Statistics and Data Science*
*University of California, Los Angeles*

**Chendi Wang**                                                                  *chendi@wharton.upenn.edu*
*Department of Statistics and Data Science*
*University of Pennsylvania*

**Will Wei Sun**                                                                      *sun244@purdue.edu*
*Daniels School of Business*
*Purdue University*

**Guang Cheng**                                                                     *guangcheng@ucla.edu*
*Department of Statistics and Data Science*
*University of California, Los Angeles*

**Reviewed on OpenReview:** *https://openreview.net/forum?id=uKCGOw9bGG*

## Abstract

Label differential privacy is a popular branch of $\epsilon$-differential privacy for protecting labels in training datasets with non-private features. In this paper, we study the generalization performance of a binary classifier trained on a dataset privatized under the label differential privacy achieved by the randomized response mechanism. Particularly, we establish minimax lower bounds for the excess risks of the deep neural network plug-in classifier, theoretically quantifying how privacy guarantee $\epsilon$ affects its generalization performance. Our theoretical result shows: (1) the randomized response mechanism slows down the convergence of excess risk by lessening the multiplicative constant term compared with the non-private case ($\epsilon = \infty$); (2) as $\epsilon$ decreases, the optimal structure of the neural network should be smaller for better generalization performance; (3) the convergence of its excess risk is guaranteed even if $\epsilon$ is adaptive to the size of training sample $n$ at a rate slower than $O(n^{-1/2})$. Our theoretical results are validated by extensive simulated examples and two real applications.

## 1 Introduction

In the past decade, differential privacy (DP; Dwork, 2008) has emerged as a standard statistical framework to protect sensitive data before releasing it to an external party. The rationale behind DP is to ensure that information obtained by an external party is robust enough to the change of a single record in a dataset. Generally, the privacy protection via DP inevitably distorts the raw data and hence reduces data utility for downstream learning tasks (Alvim et al., 2012; Kairouz et al., 2016; Li et al., 2023a). To achieve better privacy-utility tradeoff, various research efforts have been devoted to analyzing the effect of DP on learning algorithms in the machine learning community (Ghazi et al., 2021; Bassily et al., 2022; Esfandiari et al., 2022). Depending on whether the data receiver is trusted or not, differential privacy can be categorized into two main classes in the literature, including central differential privacy (central DP; Erlingsson et al. 2019; Girgis et al. 2021; Wang et al. 2023) and local differential privacy (local DP; Wang et al. 2017; Arachchige et al. 2019). Central DP relies on a trusted curator to protect all data simultaneously, whereas local DP perturbs data on the users' side. Local DP becomes a more popular solution to privacy protection due to its

successful applications, including Google Chrome browser (Erlingsson et al., 2014) and macOS (Tang et al., 2017).

An important variant of differential privacy is label differential privacy (Label DP; Chaudhuri & Hsu, 2011), which is a relaxation of DP for some real-life scenarios where input features are assumed to be publicly available and labels are highly sensitive and should be protected. Label DP has been gaining increasing attention in recent years due to the emerging demands in some real applications. For example, in recommender systems, users' ratings are sensitive for revealing users' preferences that can be utilized for advertising purposes (McSherry & Mironov, 2009; Xin & Jaakkola, 2014). In online advertising, a user click behavior is usually treated as a sensitive label whereas the product description for the displayed advertisement is publicly available (McMahan et al., 2013; Chapelle et al., 2014). These real scenarios motivate various research efforts to develop mechanisms for achieving label differential privacy and understanding the fundamental tradeoff between data utility and privacy protection. In literature, label DP can be divided into two main classes depending on whether labels are protected in a local or central manner. In central label DP, privacy protection is guaranteed by ensuring the output of a randomized learning algorithm is robust to the change of a single label in the dataset (Chaudhuri & Hsu, 2011; Ghazi et al., 2021; Bassily et al., 2022; Ghazi et al., 2023). In local label DP, labels are altered at the users' side before they are released to learning algorithms, ensuring that it is difficult to infer the true labels based on the released labels (Busa-Fekete et al., 2021; Cunningham et al., 2022).

In the literature, various efforts has been devoted to developing mechanisms to achieve label DP efficiently and analyze their essential privacy-utility tradeoffs in downstream learning tasks. The original way to achieve label DP is via randomized response mechanisms (Warner, 1965), which alters observed labels in a probabilistic manner. For binary labels, the randomized response mechanism flips labels onto the other side with a pre-determined probability (Nayak & Adeshiyan, 2009; Wang et al., 2016b; Busa-Fekete et al., 2021). Originally designed to safeguard individuals' responses in surveys (Warner, 1965; Blair et al., 2015), the RR mechanism has found extensive data collection applications (Wang et al., 2016b), such as pairwise comparisons in ranking data (Li et al., 2023b) and edges in graph data (Hehir et al., 2022; Guo et al., 2023). Ghazi et al. (2021) proposed a multi-stage training algorithm called randomized response with prior, which flips training labels via a prior distribution learned by the trained model in the previous stage. Such a multi-stage training algorithm significantly improves the generalization performance of the trained model under the same privacy guarantee. Malek Esmaeili et al. (2021) proposed to apply Laplace noise addition to one-hot encodings of labels and utilized the iterative Bayesian inference to de-noise the outputs of the privacy-preserving mechanism. Bassily et al. (2022) developed a private learning algorithm under the central label DP and established a dimension-independent deviation margin bound for the generalization performance of several differentially private classifiers, showing that the margin guarantees that are independent of the input dimension. However, their developed learning algorithm relies on the partition of the hypothesis and is hence computationally inefficient. The current state-of-the-art approach, outlined in Ghazi et al. (2023), presents a variant of RR that incorporates additional information from the loss function to enhance model performance. Analyzing this variant can be challenging due to its construction involving the solution of an optimization problem without a closed-form representation.

A critical challenge in differential privacy is to understand the essential privacy-utility tradeoff that sheds light on the fundamental utility limit for a specific problem. For example, Wang & Xu (2019) studied the sparse linear regression problem under local label DP and establish the minimax risk for the estimation error under label DP. In this paper, we intend to study the generalization performance of binary classifiers satisfying $\epsilon$-local label DP, aiming to theoretically understand how the generalization performance of differentially private classifiers are affected by the local label DP under the margin assumption (Tsybakov, 2004). To this end, we consider the local label DP via the randomized response mechanism due to its remarkable ability to incurring less utility loss (Wang et al., 2016b). Specifically, Wang et al. (2016b) demonstrated that, while adhering to the same privacy standard, the RR mechanism exhibits smaller expected mean square errors between released and actual values compared to the Laplace mechanism. Furthermore, the effectiveness of the RR mechanism surpasses that of the output perturbation approach, particularly in scenarios where the sensitivity of output functions is high. This result can be explained from the perspective of statistical hypothesis testing that the RR mechanism achieves the optimal tradeoff between type I and type II errors

under $\epsilon$-label DP (Dong et al., 2022). Additionally, the RR mechanism has a succinct representation, which allows us to develop certain theories related to deep learning.

In literature, few attempts are made to theoretically quantify the generalization performance of classifiers under local label DP even though binary classification has already become an indispensable part of the machine learning community. An important characteristic distinguishing binary classification problem is that the convergence of the generalization performance depends on the behavior of data in the vicinity of the decision boundary, which is known as the margin assumption (Tsybakov, 2004). Therefore, our first contribution is that we theoretically quantify how local label DP alters the margin assumption, which allows us to bridge the connection between the local label DP and the generalization performance. Additionally, we mainly consider two scenarios for the function class of classifiers in this paper. First, we consider the large margin classifier with its hypothesis space being a parametric function class and the deep neural network plug-in classifier. For these two scenarios, we establish their upper bound and the minimax lower bound for their excess risks, which theoretically quantifies how $\epsilon$ affects the generalization performance. The implications of our theoretical results are three-fold. First, the Bayes classifier stays invariant to the randomized response mechanism with any small $\epsilon$, which permits the possibility of learning the optimal classifier from the privatized dataset. Second, the local label DP achieved via the randomized response mechanism implicitly reduces the information for estimation. Specifically, we theoretically prove that the convergence rate of excess risk is slowed down with an additional multiplicative constant depending on $\epsilon$. Third, based on our theoretical results, we show that the excess risk fails to converge when the $\epsilon$ is adaptive to the training sample size $n$ at the order $O(n^{-1/2})$, which is independent of the margin assumption. To the best of our knowledge, no existing literature related to classification under DP or corrupted labels (Cannings et al., 2020; van Rooyen & Williamson, 2018) has investigated the effects of noise on neural network structures. Our theoretical results are supported by extensive simulations and two real applications.

The rest of this paper is organized as follows. After introducing some necessary notations in Section 1.1, Section 2 introduces the backgrounds of the binary classification problem, neural network, and the local label differential privacy. In Section 3, we introduce the framework of the differentially private learning under the label differential privacy and present theoretical results regarding the asymptotic behavior of the differentially private classifier. Section 4 quantifies how $\epsilon$ affects the generalization performance of the deep neural network plug-in classifier by establishing a minimax lower bound. Section 5 and Section 6 conduct a series of simulations and real applications to support our theoretical results. All technical proofs are provided in the Appendix.

### 1.1 Notation

For a vector $\boldsymbol{x} \in \mathbb{R}^p$, we denote its $l_1$-norm and $l_2$-norm as $\|\boldsymbol{x}\|_1 = \sum_{i=1}^p |x_i|$ and $\|\boldsymbol{x}\|_2 = \left(\sum_{i=1}^p |x_i|^2\right)^{1/2}$, respectively. For a function $f : \mathcal{X} \to \mathbb{R}$, we denote its $L_p$-norm with respect to the probability measure $\mu$ as $\|f\|_{L^p(\mu)} = \left(\int_{\mathcal{X}} |f(\boldsymbol{x})|^p d\mu(\boldsymbol{x})\right)^{1/p}$. For a real number $a$, we let $\lfloor a \rfloor$ denote the largest integer not larger than $a$. For a set $S$, we define $\mathcal{N}(\xi, S, \|\cdot\|)$ as the minimal number of $\xi$-balls needed to cover $S$ under a generic metric $\|\cdot\|$. For two given sequences $\{A_n\}_{n\in\mathbb{N}}$ and $\{B_n\}_{n\in\mathbb{N}}$, we write $A_n \gtrsim B_n$ if there exists a constant $C > 0$ such that $A_n \geq CB_n$ for any $n \in \mathbb{N}$. Additionally, we write $A_n \asymp B_n$ if $A_n \gtrsim B_n$ and $A_n \lesssim B_n$.

## 2 Preliminaries

### 2.1 Binary Classification

The goal in binary classification is to learn a discriminant function $f$, which well characterizes the functional relationship between the feature vector $\boldsymbol{X} \in \mathcal{X}$ and its associated label $Y \in \{-1, 1\}$. To measure the quality of $f$, the 0-1 risk is usually employed,

$$R(f) = \mathbb{E}\Big(I\big(f(\boldsymbol{X})Y < 0\big)\Big) = \mathbb{P}\Big(\text{sign}(f(\boldsymbol{X})) \neq Y\Big),$$

where $I(\cdot)$ denotes the indicator function and the expectation is taken with respect to the joint distribution of $(\boldsymbol{X}, Y)$.

In addition to 0-1 loss, the false negative error (FNE) and false positive error (FPE) are frequently used to assess classifier performance when dealing with highly imbalanced datasets. Specifically, FNE measures the percentage of positive samples being classified as negative, whereas FPE measures the percentage of negative samples being classified as positive. For a margin classifier $f$, the expected false positive error (EFNE) and false positive error (EFPE) are written as

$$\mathrm{EFNE}(f) = \mathbb{E}\Big(I(f(\boldsymbol{X}) < 0)\big|Y = 1\Big) = \mathbb{P}\Big(\mathrm{sign}(f(\boldsymbol{X})) \neq Y\big|Y = 1\Big),$$

$$\mathrm{EFPE}(f) = \mathbb{E}\Big(I(f(\boldsymbol{X}) > 0)\big|Y = -1\Big) = \mathbb{P}\Big(\mathrm{sign}(f(\boldsymbol{X})) \neq Y\big|Y = -1\Big).$$

Furthermore, the 0-1 risk $R(f)$ can be re-written as a weighted combination of EFNE and EFPE:

$$R(f) = \mathrm{EFNE}(f)\mathbb{P}(Y = 1) + \mathrm{EFPE}(f)\mathbb{P}(Y = -1).$$

Let $f^* = \inf_f R(f)$ denote the minimizer of $R(f)$, which refers to as the Bayes decision rule. Generally, $f^*$ is obtained by minimizing $R(f)$ in a point-wise manner and given as $f^*(\boldsymbol{X}) = \mathrm{sign}\big(\eta(\boldsymbol{X}) - 1/2\big)$ with $\eta(\boldsymbol{X}) = \mathbb{P}(Y = 1|\boldsymbol{X})$. The minimal risk $R(f^*)$ can be written as $R(f^*) = \mathbb{E}_{\boldsymbol{X}}\big(\min\{\eta(\boldsymbol{X}), 1 - \eta(\boldsymbol{X})\}\big)$. In practice, the underlying joint distribution on $(\boldsymbol{X}, Y)$ is unavailable, but a set of i.i.d. realizations $\mathcal{D} = \{(\boldsymbol{x}_i, y_i)\}_{i=1}^n$ is given. Therefore, it is a common practice to consider the estimation procedure based on minimizing the sample average of a surrogate loss, which is given as

$$\widehat{R}_\phi(f) = \frac{1}{n}\sum_{i=1}^n \phi(f(\boldsymbol{x}_i)y_i),$$

where $\phi(\cdot)$ is the surrogate loss function replacing the 0-1 loss since the 0-1 loss is computationally intractable (Arora et al., 1997).

Let $R_\phi(f) = \mathbb{E}\big(\phi(f(\boldsymbol{X})Y)\big)$ denote the $\phi$-risk. Given that $\phi$ is classification calibrated, the minimizer $f_\phi^* = \arg\min_f R_\phi(f)$ is consistent with the Bayes decision rule (Lin, 2004), i.e., $\mathrm{sign}(f_\phi^*(\boldsymbol{x})) = \mathrm{sign}(\eta(\boldsymbol{x}) - 1/2)$ for any $\boldsymbol{x} \in \mathcal{X}$. In literature, there are various classification-calibrated loss functions (Zhang, 2004; Bartlett et al., 2006), such as exponential loss, hinge loss, logistic loss, and $\psi$-loss (Shen et al., 2003).

## 2.2 Deep Neural Network and Function Class

Let $f(\boldsymbol{x}; \Theta)$ be an $L$-layer neural network with Rectified Linear Unit (ReLU) activation function, that is,

$$f(\boldsymbol{x}; \Theta) = \boldsymbol{A}_{L+1}\big(\boldsymbol{h}_L \circ \boldsymbol{h}_{L-1} \circ \cdots \boldsymbol{h}_1(\boldsymbol{x})\big) + \boldsymbol{b}_{L+1},$$

where $\circ$ denotes function composition, $\boldsymbol{h}_l(\boldsymbol{x}) = \sigma(\boldsymbol{A}_l\boldsymbol{x} + \boldsymbol{b}_l)$ denotes the $l$-th layer, and $\Theta = \{(\boldsymbol{A}_l, \boldsymbol{b}_l)\}_{l=1,\ldots,L+1}$ denotes all the parameters. Here $\boldsymbol{A}_l \in \mathbb{R}^{p_l \times p_{l-1}}$ is the weight matrix, $\boldsymbol{b}_l \in \mathbb{R}^{p_l}$ is the bias term, $p_l$ is the number of neurons in the $l$-th layer, and $\sigma(x) = \max\{0, x\}$ is the ReLU function. To characterize the network architecture of $f$, we denote the number of layers in $\Theta$ as $\Upsilon(\Theta)$, the maximum number of nodes as $\Delta(\Theta)$, the number of non-zero parameters as $\|\Theta\|_0$, the largest absolute value in $\Theta$ as $\|\Theta\|_\infty$. For a given $n$, we denote by $\mathcal{F}_n^{NN}(L_n, N_n, P_n, B_n, V_n)$ a class of neural networks, which is defined as

$$\mathcal{F}_n^{NN}(L_n, N_n, P_n, B_n, V_n) = \big\{f(\boldsymbol{x}; \Theta) : \Upsilon(\Theta) \leq L_n, \Delta(\Theta) \leq N_n,$$
$$\|\Theta\|_0 \leq P_n, \|\Theta\|_\infty \leq B_n, \|f\|_\infty \leq V_n, \big\}.$$

Let $\beta > 0$ be a degree of smoothness, then the Hölder space is defined as

$$\mathcal{H}(\beta, \mathcal{X}) = \{f \in \mathcal{C}^{\lfloor\beta\rfloor}(\mathcal{X}) : \|f\|_{\mathcal{H}(\beta, \mathcal{X})} < \infty\},$$

where $\mathcal{C}^{\lfloor\beta\rfloor}(\mathcal{X})$ the class of $\lfloor\beta\rfloor$ times continuously differentiable functions on the open set $\mathcal{X}$ and the Hölder norm $\|f\|_{\mathcal{H}(\beta, \mathcal{X})}$ is given as

$$\|f\|_{\mathcal{H}(\beta, \mathcal{X})} = \max_{\boldsymbol{m}:\|\boldsymbol{m}\|_1 \leq \lfloor\beta\rfloor} \sup_{\boldsymbol{x} \in \mathcal{X}} |\partial^{\boldsymbol{m}} f(\boldsymbol{x})| + \max_{\boldsymbol{m}:\|\boldsymbol{m}\|_1 = \lfloor\beta\rfloor} \sup_{\boldsymbol{x}, \boldsymbol{y} \in \mathcal{X}, \boldsymbol{x} \neq \boldsymbol{y}} \frac{|\partial^{\boldsymbol{m}} f(\boldsymbol{x}) - \partial^{\boldsymbol{m}} f(\boldsymbol{y})|}{\|\boldsymbol{x} - \boldsymbol{y}\|_2^{\beta - \lfloor\beta\rfloor}},$$

where $\partial^{\boldsymbol{m}} f(\boldsymbol{x}) = \frac{\partial^{m_1 + \ldots + m_p}}{\partial x_1^{m_1} \ldots \partial x_p^{m_p}} f(\boldsymbol{x})$ denotes the partial derivative of order $\boldsymbol{m}$ with respect to $\boldsymbol{x}$ and $\boldsymbol{m} = (m_1, \ldots, m_p) \in \mathbb{N}_0^p$ is a multi-index with $\mathbb{N}_0 = \mathbb{N} \cup \{0\}$. Further, we let $\mathcal{H}(\beta, \mathcal{X}, M) = \{f \in \mathcal{H}(\beta, \mathcal{X}) : \|f\|_{\mathcal{H}(\beta, \mathcal{X})} \leq M\}$ be a closed ball of radius $M$ in $\mathcal{H}(\beta, \mathcal{X})$.

The Hölder assumption serves as a common assumption for studying the generalization capabilities of neural networks for approximating functions possessing a certain degree of smoothness or regularity (Audibert & Tsybakov, 2007; Kim et al., 2021). This assumption is useful in developing a tighter generalization bounds. However, it is essential to acknowledge that the Hölder assumption presupposes a level of smoothness in the underlying function. In reality, many real-world problems involve functions that deviate from this idealized smoothness. When the actual smoothness of the function does not align with the assumption, algorithms reliant on it may fall short of the anticipated generalization performance.

### 2.3 Local Label Differential Privacy

Label differential privacy (Label DP; Chaudhuri & Hsu, 2011) is proposed as a relaxation of differential privacy (Dwork, 2008), aiming to protect the privacy of labels in the dataset, whereas training features are non-sensitive and hence publicly available. An effective approach to label DP is the randomized response mechanism (Busa-Fekete et al., 2021). As its name suggests, it protects the privacy of labels via some local randomized response mechanisms under the framework of local differential privacy.

**Definition 1.** *(Label Differential Privacy; Ghazi et al., 2021) A randomized training algorithm $\mathcal{A}$ taking as input a dataset is said to be $(\epsilon, \delta)$-label differentially private if for any two datasets $\mathcal{D}$ and $\mathcal{D}'$ that differ in the label of a single example, and for any subset $S$ of the outputs of $\mathcal{A}$, it is the case that $\mathbb{P}(\mathcal{A}(\mathcal{D}) \in S) \leq e^{\epsilon} \mathbb{P}(\mathcal{A}(\mathcal{D}') \in S) + \delta$, then $\mathcal{A}$ is said to be $\epsilon$-label differentially private ($\epsilon$-label DP).*

A direct way to achieve $\epsilon$-Label DP in binary classification is to employ the randomized response mechanism (Warner, 1965; Nayak & Adeshiyan, 2009; Wang et al., 2016b; Karwa et al., 2017). The main idea of the binary randomized response mechanism is to flip observed labels indepdently with a fixed probability. Specifically, let $\mathcal{A}_\theta$ denote the randomized response mechanism parametrized by $\theta$. For an input label $Y$, we define

$$\mathcal{A}_\theta(Y) = \begin{cases} Y, & \text{with probability } \theta, \\ -Y, & \text{with probability } 1 - \theta, \end{cases}$$

where $\theta > 1/2$ denotes the probability that the value of $Y$ stays unchanged. It is straightforward to verify that the randomized response mechanism satisfies $\epsilon$-label DP with $\epsilon = \log(\theta/(1 - \theta))$ (Ghazi et al., 2021; Busa-Fekete et al., 2021). In this paper, we denote the $\epsilon$-label DP achieved through the randomized response mechanism as $\epsilon$-local label DP.

## 3 Differentially Private Learning

### 3.1 Effect of Locally-Label Differential Privacy

Under the setting of local differential privacy, users do not trust the data curator and hence privatize their sensitive data via some randomized mechanisms locally before releasing them to central servers. We let $\mathcal{D} = \{(\boldsymbol{x}_i, y_i)\}_{i=1}^n$ denote the original dataset containing $n$ i.i.d. realizations of the random pair $(\boldsymbol{X}, Y)$. As illustrated in Figure 1, users' labels are privatized by a locally differentially private protocol $\mathcal{A}_\theta$, and then the untrusted curator receives a privatized dataset, which we denote as $\widetilde{\mathcal{D}} = \{(\boldsymbol{x}_i, \widetilde{y}_i)\}_{i=1}^n$.

A natural question is what the quantitative relation between $\epsilon$-label DP and the discrepancy between the distributions of $\mathcal{D}$ and $\widetilde{\mathcal{D}}$ is. This is useful to analyze how privacy parameter $\epsilon$ deteriorates the utility of data for downstream learning tasks.

Notice that $\widetilde{y}_i$'s are generated locally and independently, therefore the randomized response mechanism only alters the conditional distribution of $Y$ given $\boldsymbol{X}$, whereas the marginal distribution of $\boldsymbol{X}$ stays unchanged. Hence, we can assume $\widetilde{\mathcal{D}}$ is a set of i.i.d. realizations of $(\boldsymbol{X}, \widetilde{Y})$, and it is straightforward to verify that the

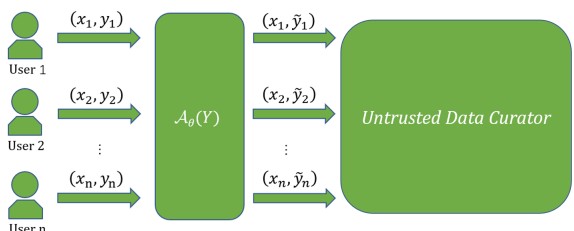

Figure 1: The framework of local label differential privacy.

conditional distribution of $\widetilde{Y}$ given $\boldsymbol{X} = \boldsymbol{x}$ can be written as

$$\widetilde{\eta}(\boldsymbol{x}) = \mathbb{P}(\widetilde{Y} = 1|\boldsymbol{X} = \boldsymbol{x}) = \theta\eta(\boldsymbol{x}) + (1-\theta)(1-\eta(\boldsymbol{x})).$$

Clearly, $\mathcal{A}_\theta$ amplifies the uncertainty of the observed labels by shrinking $\widetilde{\eta}(\boldsymbol{x})$ towards $1/2$. Particularly, $\widetilde{\eta}(\boldsymbol{X}) = 1/2$ almost surely when $\theta = 1/2$. In this case, the privatized dataset $\widetilde{\mathcal{D}}$ conveys no information for learning the decision function.

The following Lemma quantifies how the randomized response mechanism alters the conditional distribution of $Y$ and $\widetilde{Y}$ given the feature $\boldsymbol{x}$ via the Kullback-Leibler divergence (KL) divergence between these two Bernoulli distributions. Specifically, the inequalities in (1) explicates how the divergence changes with privacy guarantee $\epsilon$.

**Lemma 1.** *Suppose the randomized response mechanism $\widetilde{Y} = \mathcal{A}_\theta(Y)$ satisfies $\epsilon$-label DP with $\theta = \exp(\epsilon)/(1+\exp(\epsilon))$, then for any $\boldsymbol{x} \in \mathcal{X}$ it holds that*

$$\mathcal{L}(\epsilon, \boldsymbol{x}) \le D_{KL}\big(\mathbb{P}_{Y|\boldsymbol{X}=\boldsymbol{x}}\big|\mathbb{P}_{\widetilde{Y}|\boldsymbol{X}=\boldsymbol{x}}\big) \le \mathcal{U}(\epsilon, \boldsymbol{x}), \tag{1}$$

*where $D_{KL}\big(\mathbb{P}_{Y|\boldsymbol{X}=\boldsymbol{x}}\big|\mathbb{P}_{\widetilde{Y}|\boldsymbol{X}=\boldsymbol{x}}\big)$ denotes the Kullback-Leibler divergence (KL) divergence between $\mathbb{P}_{\widetilde{Y}|\boldsymbol{X}=\boldsymbol{x}}$ and $\mathbb{P}_{Y|\boldsymbol{X}=\boldsymbol{x}}$, $\mathcal{L}(\epsilon, \boldsymbol{x}) = 2(2\eta(\boldsymbol{x}) - 1)^2(1 + \exp(\epsilon))^{-2}$, and $\mathcal{U}(\epsilon, \boldsymbol{x}) = \min\{U_1(\epsilon, \boldsymbol{x}), U_2(\epsilon, \boldsymbol{x})\}$ with $U_1(\epsilon, \boldsymbol{x}) = (2\eta(\boldsymbol{x}) - 1)^2\eta^{-1}(\boldsymbol{x})(1 - \eta(\boldsymbol{x}))^{-1}(1 + \exp(\epsilon))^{-2}$ and $U_2(\epsilon, \boldsymbol{x}) = (2\eta(\boldsymbol{x}) - 1)^2 \exp(-\epsilon)$.*

In Lemma 1, the lower bound $\mathcal{L}(\epsilon, \boldsymbol{x})$ and the upper bound $\mathcal{U}(\epsilon, \boldsymbol{x})$ share a factor $(1 + \exp(-\epsilon))^2$, indicating that $D_{KL}(\mathbb{P}_{Y|\boldsymbol{X}=\boldsymbol{x}}|\mathbb{P}_{\widetilde{Y}|\boldsymbol{X}=\boldsymbol{x}})$ decreases exponentially with respect to $\epsilon$.

## 3.2 Differentially Private Classifier

On the side of the untrusted curator, inference tasks vary according to the purpose of data collector, such as estimation of population statistics (Joseph et al., 2018; Yan et al., 2019) and supervised learning (Ghazi et al., 2021; Esfandiari et al., 2022). In this paper, we suppose that the computation based on $\widetilde{\mathcal{D}}$ implemented by the untrusted curator is formulated as the following regularized empirical risk minimization task,

$$\min_{f \in \mathcal{F}} L_n(f) = \min_{f \in \mathcal{F}} \frac{1}{n} \sum_{i=1}^{n} \phi(f(\boldsymbol{x}_i)\widetilde{y}_i) + \lambda_n J(f), \tag{2}$$

where $\phi$ is a surrogate loss, $\lambda_n$ is a tuning parameter, $J(\cdot)$ is a penalty term, and $\mathcal{F}$ is a pre-specified hypothesis space.

We denote by $\widetilde{R}(f) = \mathbb{E}\big[\text{sign}(f(\boldsymbol{X})) \ne \widetilde{Y}\big]$ the risk with expectation taken with respect to the joint distribution of $(\boldsymbol{X}, \widetilde{Y})$ and let $\widetilde{f}^* = \arg\min_f \widetilde{R}(f)$ denote the Bayes decision rule under the distribution of $(\boldsymbol{X}, \widetilde{Y})$. The excess risk of $f$ under the distributions of $(\boldsymbol{X}, Y)$ and $(\boldsymbol{X}, \widetilde{Y})$ is denoted as $D(f, f^*) = R(f) - R(f^*)$ and $\widetilde{D}(f, \widetilde{f}^*) = \widetilde{R}(f) - \widetilde{R}(\widetilde{f}^*)$, respectively.

**Lemma 2.** *If $\widetilde{Y} = \mathcal{A}_\theta(Y)$ with $\theta > 1/2$, then $f^*(\boldsymbol{x}) = \widetilde{f}^*(\boldsymbol{x})$ for any $\boldsymbol{x} \in \mathcal{X}$ and $\widetilde{D}(f, \widetilde{f}^*) = (2\theta - 1)D(f, f^*)$ for any $f$.*

Lemma 2 shows that the Bayes decision rule stays invariant under the randomized response mechanism. It is clear to see that $\widetilde{D}(f, f^*)$ diminishes as $\theta$ gets close to $1/2$. Particularly, $\widetilde{D}(f, f^*)$ is equal to 0 when $\theta = 1/2$. This is as expected since $\theta = 1/2$ implies that $\widetilde{\eta}(\boldsymbol{X}) = 1/2$ almost surely, where all classifiers deteriorates in performance simultaneously. This result demonstrates that the optimal classifiers for the underlying distributions of $\mathcal{D}$ and $\widetilde{\mathcal{D}}$ are identical. Basically, Lemma 2 reveals an inherent debiasing process in the development of the generalization performance of the differentially private classifier $\widetilde{f}$. To be more specific, we can deduce the excess risk of $\widetilde{f}$ by the relation $\widetilde{R}(\widetilde{f}) - \widetilde{R}(f^*) = \frac{e^\epsilon - 1}{e^\epsilon + 1}(R(\widetilde{f}) - R(f^*))$ under $\epsilon$-local label DP.

Let $f^*_\phi = \arg\min_f R_\phi(f)$ and $\widetilde{f}^*_\phi = \arg\min_f \widetilde{R}_\phi(f)$ be the minimizers of $\phi$-risk under the joint distributions of $(\boldsymbol{X}, Y)$ and $(\boldsymbol{X}, \widetilde{Y})$, respectively. For illustration, we only consider hinge loss $\phi(x) = \max\{1 - x, 0\}$ in this paper, and similar results can be easily obtained for other loss functions by the comparison theorem (Bartlett et al., 2006; Bartlett & Wegkamp, 2008). With $\phi(\cdot)$ being the hinge loss, we have $\widetilde{f}^*_\phi = f^*_\phi = f^* = \widetilde{f}^*$ (Bartlett et al., 2006; Lecué, 2007). With a slight abuse of notation, we use $f^*$ to refer to these four functions simultaneously in the sequel.

## 3.3 Consistency under the Randomized Response Mechanism

In this section, we establish the asymptotic behavior of the classifier trained on privatized datasets. Specifically, we theoretically quantify how the randomized response mechanism affects the convergence rate of excess risk under the low-noise assumption (Lecué, 2007).

We suppose that the randomized response mechanism satisfies the $\epsilon$-label DP, which indicates that $\epsilon = \log(\theta/(1-\theta))$. Furthermore, we denote that $\widetilde{f}_n = \arg\min_{f \in \mathcal{F}} L_n(f)$ and $H(f, f^*) = R_\phi(f) - R_\phi(f^*)$. In classification problems, $H(f, f^*)$ is an important metric that admits the decomposition into the estimation error and the approximation error (Bartlett et al., 2006),

$$H(f, f^*) = H(f, f^*_\mathcal{F}) + H(f^*_\mathcal{F}, f^*),$$

where $f^*_\mathcal{F} = \arg\min_{f \in \mathcal{F}} R_\phi(f)$. Here $H(f, f^*_\mathcal{F})$ is the estimation error and $H(f^*_\mathcal{F}, f^*)$ is the approximation error. The estimation error depends on the learning algorithm in finding $f^*_\mathcal{F}$ based on a dataset with finite samples, where the searching difficulty is unavoidably affected by the complexity of $\mathcal{F}$ as the approximation error. Generally, the richness of $\mathcal{F}$ can be measured by VC dimension (Blumer et al., 1989; Vapnik & Chervonenkis, 2015), Rademacher complexity (Bartlett & Mendelson, 2002; Smale & Zhou, 2003), and metric entropy methods (Zhou, 2002; Shalev-Shwartz & Ben-David, 2014).

**Assumption 1.** *(Low-noise assumption) There exists a constant $c > 0$ and $0 < \gamma \leq +\infty$ such that $\mathbb{P}\Big(|2\eta(\boldsymbol{X}) - 1| \leq t\Big) \leq ct^\gamma$, for any $t \in [0, 1)$.*

Assumption 1 is known as the low-noise assumption in the binary classification (Lecué, 2007; Shen et al., 2003; Bartlett et al., 2006), which characterizes the behavior of $2\eta(\boldsymbol{x}) - 1$ around the decision boundary $\eta(\boldsymbol{x}) = 1/2$. Particularly, the case with $\gamma = +\infty$ and $c = 1$ implies that the labels $y_i$'s are deterministic in that $\eta(\boldsymbol{X})$ takes values in $\{0, 1\}$ almost surely, resulting in the fastest convergence rate of the estimation error.

**Lemma 3.** *Denote that $\kappa_\epsilon = (e^\epsilon - 1)/(e^\epsilon + 1)$. Under Assumption 1, for any $\theta \in (1/2, 1]$, it holds that*

*(1) $\mathbb{P}\Big(|2\widetilde{\eta}(\boldsymbol{X}) - 1| \leq t\Big) \leq c\kappa_\epsilon^{-\gamma} t^\gamma$, for any $t \in [0, 1)$,*

*(2) $\widetilde{R}_\phi(f) - \widetilde{R}_\phi(f^*) \geq \kappa_\epsilon (4c)^{-1/\gamma} \Big(\mathbb{E}\big[|f(\boldsymbol{X}) - f^*(\boldsymbol{X})|\big]\Big)^{\frac{\gamma+1}{\gamma}}$ for any $f \in \mathcal{F}$,*

Lemma 3 presents some insights regarding the influence of the randomized response mechanism on the low-noise assumption and the margin relation (Lecué, 2007), showing that the low-noise structure and margin relation are both invariant to the randomized response mechanism in the sense that only the multiplicative terms are enlarged by the effect of the privacy guarantee $\epsilon$.

**Assumption 2.** *We assume that $\mathcal{F}$ is properly chosen satisfying that $\|f\|_\infty \le 1$ for any $f \in \mathcal{F}$ and*

$$\log \mathcal{N}(\xi, \mathcal{F}, \|\cdot\|_{L^2(\mu)}) \asymp \mathcal{V}_1(\Theta) \log(1 + \xi^{-1}\mathcal{V}_2(\Theta)), \text{ and } VC(\mathcal{G}_\mathcal{F}) \asymp \mathcal{V}_1(\Theta)$$

*where $\mathcal{G}_\mathcal{F} = \{\{\boldsymbol{x} : \text{sign}(f(\boldsymbol{x})) = 1\} : f \in \mathcal{F}\}$, $VC(\mathcal{G}_\mathcal{F})$ denotes the VC dimension of $\mathcal{G}_\mathcal{F}$, $\mu$ denotes the marginal distribution of $\boldsymbol{X}$, $\Theta$ denotes the parameters of $f$, and $\mathcal{V}_1(\Theta)$ and $\mathcal{V}_2(\Theta)$ are some functions depending on $\Theta$.*

Assumption 2 characterizes the complexity of function class $\mathcal{F}$ through the metric entropy (Zhou, 2002; Bousquet et al., 2003; Lei et al., 2016), where $\mathcal{V}_1(\Theta)$ and $\mathcal{V}_2(\Theta)$ are quantities increasing with the size of $\Theta$. Assumption 2 generally holds for function classes of parametric models (Wang et al., 2016a; Xu et al., 2021), most notably for deep neural networks (Bartlett et al., 2019; Schmidt-Hieber, 2020). Additionally, Assumption 2 also holds for those VC classes with VC dimensions increasing with the size of $\Theta$ (Wellner et al., 2013; Bartlett et al., 2019; Lee et al., 1994).

**Theorem 1.** *Under Assumptions 1 and 2, for any minimizer $\widetilde{f}_n$ of (2), there exist some positive constants $A_1$ and $A_2$ such that*

$$A_2\left\{\left(\frac{\mathcal{V}_1(\Theta)}{n\kappa_\epsilon^2}\right)^{\frac{\gamma+1}{\gamma+2}} + \tau_n\right\} \le \sup_{\pi \in \mathcal{P}_\gamma} \mathbb{E}_{\widetilde{\mathcal{D}}}\left[R(\widetilde{f}_n) - R(f^*)\right] \le A_1\left\{\left(\frac{\mathcal{V}_1(\Theta)\log(n)}{n\kappa_\epsilon^2}\right)^{\frac{\gamma+1}{\gamma+2}} + s_n\right\},$$

*where $\mathcal{P}_\gamma$ be a class of distributions of $(\boldsymbol{X}, Y)$ satisfying Assumption 1, $s_n = \sup_{\pi \in \mathcal{P}_\gamma} \inf_{f \in \mathcal{F}} H(f, f^*)$, and $\tau_n = \sup_{\pi \in \mathcal{P}_\gamma} \inf_{f \in \mathcal{F}} D(f, f^*)$.*

Theorem 1 quantifies the asymptotic behavior of $\widetilde{f}_n$ by establishing its upper and lower bounds, which explicitly demonstrates the quantitative relation between the effect of privacy guarantee $\epsilon$ and the excess risk. The upper bound is proven through a uniform concentration inequality under the framework of empirical risk minimization analysis. The estimator $\widetilde{f}_n$ is derived from a pre-specified function class $\mathcal{F}$ that may not include the true underlying function $f^*$. Consequently, $s_n$ represents the approximation error, quantifying the capability of the optimal function within $\mathcal{F}$ to approximate $f^*$. The accuracy of estimating the optimal function in $\mathcal{F}$ for approximating $f^*$ depends on the complexity of $\mathcal{F}$ (Assumption 2) and the size of the training set $n$. A larger $\mathcal{F}$ may reduce $s_n$ but can increase the estimation error. Thus, achieving the best convergence rate involves striking the right balance between these two sources of error. The proof of the lower bound is similar to Theorem 2 in Lecué (2007), mainly applying the Assouad's lemma (Yu, 1997) to an analytic subset of $\mathcal{P}\gamma$. Further details regarding the proof are provided in the Appendix.

It is worth noting that the upper bound matches the lower bound except for a logarithmic factor when the approximation error term $s_n \lesssim \left(\frac{\mathcal{V}_1(\Theta)}{n\kappa_\epsilon^2}\right)^{\frac{\gamma+1}{\gamma+2}}$. This shows that the randomized response mechanism slows down the convergence rate by enlarging the multiplicative constant. Moreover, based on Theorem 1, we can obtain the optimal convergence rate of the excess risk in classification problem under the low-noise assumption (Lecué, 2007) by setting $\epsilon = \infty$ and $|\mathcal{F}| < \infty$.

**Lemma 4.** *Denote that $\mathcal{M} = \max\{EFNE(f) - EFNE(f^*), EFPE(f) - EFPE(f^*)\}$. Under Assumption 1, for any margin classifier $f$, it holds that*

$$R(f) - R(f^*) \le \mathcal{M} \le \frac{1}{2\min\{\mathbb{P}(Y=1), \mathbb{P}(Y=-1)\}}\left(2c^{-\frac{1}{1+\gamma}}(R(f) - R(f^*))^{\frac{\gamma}{1+\gamma}} + R(f) - R(f^*)\right), \quad (3)$$

*where $c$ and $\gamma$ are as defined in Assumption 1. Particularly, if Assumption 1 holds with $\gamma = \infty$, for any margin classifier $f$, (3) becomes*

$$R(f) - R(f^*) \le \mathcal{M} \le \frac{3}{2\min\{\mathbb{P}(Y=1), \mathbb{P}(Y=-1)\}}(R(f) - R(f^*)).$$

Lemma 4 establishes a crucial relationship between excess risk and the maximum of excess EFNE and EFPE for any classifier $f$, which also includes $\widetilde{f}_n$ as a special case. This connection enables us to demonstrate the convergence of $\text{EFNE}(\widetilde{f}_n)$ and $\text{EFPE}(\widetilde{f}_n)$ based on that of the excess risk. Remarkably, this finding

implies that the differentially private classifier $\widetilde{f}_n$ will exhibit similar false negative and false positive rates to those of the Bayes classifier as the sample size $n$ tends to infinity. In addition, increasing the degree of data imbalance will enlarge the upper bound in (3), indicating that data imbalance slows down the convergence rates of $\mathrm{EFNE}(\widetilde{f}_n)$ and $\mathrm{EFPE}(\widetilde{f}_n)$. Particularly, under the low-noise assumption with $\gamma = \infty$, which implies that samples are separable, both excess EFNE and EFPE of $\widetilde{f}_n$ exhibit the same convergence rate as the excess risk regardless of the degree of class imbalance. Furthermore, this result indicates that the privacy guarantee $\epsilon$ has a similar impact on the excess EFNE and EFPE as it does on the excess risk.

## 4 Deep Learning with Local Label Differential Privacy

A typical class of models that are popularly considered in the domain of differential privacy is deep neural network (Ghazi et al., 2021; Yuan et al., 2021) due to its success in various applications in the past decade. Unlike Section 3.3 considering the estimation and approximation errors separately, we establish theoretical results regarding the convergence rate of the excess risk of the deep neural network plug-in classifier trained from $\widetilde{\mathcal{D}}$, which is obtained by making a tradeoff between the estimation and approximation errors of deep neural networks (Schmidt-Hieber, 2020). Our theoretical results not only quantify how the optimal structure of the deep neural network changes with the privacy parameter $\epsilon$, but also derive the optimal privacy guarantee we can achieve for the deep neural network classifier.

Remind that $\widetilde{\mathcal{D}} = \{(\boldsymbol{x}_i, \widetilde{y}_i)\}_{i=1}^n$ is the privatized dataset with $\widetilde{y}_i = \mathcal{A}_\theta(y_i)$ and $\theta = \exp(\epsilon)/(1 + \exp(\epsilon))$. The deep neural network is solved as

$$\widetilde{f}_{nn} = \underset{f \in \mathcal{F}_n^{NN}(L_n, N_n, P_n, B_n, V_n)}{\arg\min} \frac{1}{n} \sum_{i=1}^n \left(f(\boldsymbol{x}_i) - \widetilde{z}_i\right)^2, \tag{4}$$

where $\widetilde{z}_i = (\widetilde{y}_i + 1)/2$ and $\mathcal{F}_n^{NN}(L_n, N_n, P_n, B_n, V_n)$ is a class of multilayer perceptrons defined in Section 2.2. The plug-in classifier based on $\widetilde{f}_{nn}$ can be obtained as $\widetilde{s}_{nn} = \mathrm{sign}(\widetilde{f}_{nn} - 1/2)$. To quantify the asymptotic behavior of $\widetilde{s}_{nn}$, we further assume that the support of $\boldsymbol{x}$ is $[0,1]^p$, which is a common assumption for deep neural networks (Yarotsky, 2017; Nakada & Imaizumi, 2020)

**Theorem 2.** *Let $\mathcal{P}_{\gamma,\beta}$ be a class of probability measures on $\mathcal{X} \times \{-1, 1\}$ satisfying Assumption 1 and $\eta(\boldsymbol{X}) \in \mathcal{H}(\beta, [0,1]^p, M)$. For any minimizer $\widetilde{f}_{nn}$ in (4) with $L_n \asymp \log(\kappa_\epsilon n / \log(n))$, $N_n \asymp (\kappa_\epsilon n / \log(n))^{\frac{2p}{2\beta+p}}$, $B_n = 1$, and $P_n \asymp N_n \log(\kappa_\epsilon n / \log(n))$, we have*

$$\left(\frac{1}{n\kappa_\epsilon^2}\right)^{\frac{\beta(\gamma+1)}{\beta(\gamma+2)+p}} \lesssim \sup_{\pi \in \mathcal{P}_{\gamma,\beta}} \mathbb{E}_{\widetilde{\mathcal{D}}}\left[R(\widetilde{s}_{nn}) - R(f^*)\right] \lesssim \left(\frac{\log n}{n\kappa_\epsilon^2}\right)^{\frac{2\beta(\gamma+1)}{2\beta(\gamma+2)+p(\gamma+2)}}. \tag{5}$$

*Particularly,* $\sup_{\pi \in \mathcal{P}_{\gamma,\beta}} \mathbb{E}_{\widetilde{\mathcal{D}}}\left[R(\widetilde{s}_{nn}) - R(f^*)\right] = o(1)$ *given that* $\epsilon \gtrsim n^{-1/2+\zeta}$ *for any* $\zeta > 0$.

In Theorem 2, we quantify the asymptotic behavior of the excess risk of $\widetilde{s}_{nn}$ by providing upper and lower bounds for $\sup_{\pi \in \mathcal{P}_{\gamma,\beta}} \mathbb{E}_{\widetilde{\mathcal{D}}}\left[R(\widetilde{s}_{nn}) - R^*\right]$. Similar to Theorem 1, the proof of the lower bound of (5) relies on the Assouad's lemma. A significant distinction of the upper bound in (5) from that of Theorem 1 is explicating the approximation error $s_n$ with respect to the structure of neural network and making the optimal tradeoff between estimation and approximation errors to achieve the fastest convergence rate. It should be noted that if $\epsilon = \infty$ which refers to the non-private case, the upper and lower bounds in (5) match with existing theoretical results established in Audibert & Tsybakov (2007). Moreover, Theorem 2 goes a step further by precisely characterizing the impact of $\epsilon$ on the convergence of $\widetilde{s}_{nn}$ to the Bayes decision rule. Additionally, it specifies how the optimal neural network's structure contracts as $\epsilon$ decreases, crucial for achieving the fastest convergence rate. Specifically, attaining this rapid convergence necessitates reducing the maximum number of hidden units at an order of $O(\epsilon^{2p/(2\beta+p)})$ when compared to the non-private case. Furthermore, we leverage Theorem 2 to derive the fastest adaptive rate of $\epsilon$ under the consistency of $\widetilde{s}_{nn}$. Specifically, we find that $\epsilon \gtrsim n^{-1/2+\zeta}$ for any $\zeta > 0$ to achieve this desired consistency rate. This result represents a crucial step towards understanding the interplay between privacy and performance in our framework.

## 5 Simulated Experiments

This section aims to validate our theoretical results through extensive simulated examples. Specifically, we show that the excess risk of a differentially private classifier converges to 0 for any fixed $\epsilon$, whereas the convergence is not achievable as long as $\epsilon$ is adaptive to the size of the training dataset with some properly chosen orders as shown in Theorems 1 and 2.

### 5.1 Support Vector Machine

This simulation experimentally analyzes the effect of label DP on the SVM classifier. The generation of simulated datasets is as follows. First, we set the regression function in classification as $\boldsymbol{\eta}(\boldsymbol{x}) = 1/\big(1 + \exp(-\boldsymbol{\beta}_0^T \boldsymbol{x})\big)$, where $\boldsymbol{\beta}_0 \in \mathbb{R}^p$ and $\boldsymbol{x}$ are both $p$-dimensional vectors generated via $\beta_{0i}, x_i \sim \text{Unif}(-1, 1)$ for $i = 1, \ldots, p$. For each feature $\boldsymbol{x}$, its label $y$ is chosen from $\{1, -1\}$ with probabilities $\boldsymbol{\eta}(\boldsymbol{x})$ and $1 - \boldsymbol{\eta}(\boldsymbol{x})$, respectively. Repeating the above process $n$ times, we obtain a non-private training dataset $\mathcal{D} = \{(\boldsymbol{x}_i, y_i)\}_{i=1}^n$. Subsequently, we apply the randomized response mechanism to generate $\widetilde{y}_i$ as $\widetilde{y}_i = \mathcal{A}_\theta(y_i)$, where $\theta = \exp(\epsilon)/(1 + \exp(\epsilon))$ and $\epsilon$ is the privacy guarantee. Therefore, the obtained privatized training dataset $\widetilde{\mathcal{D}} = \{(\boldsymbol{x}_i, \widetilde{y}_i)\}_{i=1}^n$ satisfies $\epsilon$-Label DP. Based on $\widetilde{\mathcal{D}}$, we obtain the SVM classifier (Cortes & Vapnik, 1995),

$$\widetilde{f}_n = \arg\min_{\boldsymbol{\beta}} \frac{1}{n} \sum_{i=1}^n (1 - \widetilde{y}_i \boldsymbol{\beta}^T \boldsymbol{x}_i)_+ + \lambda \|\boldsymbol{\beta}\|_2^2,$$

where $(x)_+ = \max\{x, 0\}$. Next, we evaluate the performance of $\widetilde{f}_n$ in terms of the empirical excess risk and the classification error,

$$\widehat{E}(\widetilde{f}_n) = \frac{1}{n_{\text{test}}} \sum_{i=1}^{n_{test}} I\Big( \text{sign}(\widetilde{f}_n(\boldsymbol{x}_i')) \neq \text{sign}(\eta(\boldsymbol{x}_i') - 1/2) \Big) \big|2\eta(\boldsymbol{x}_i') - 1\big|,$$

$$\text{CE}(\widetilde{f}_n) = \frac{1}{n_{test}} \sum_{i=1}^{n_{test}} I\Big( \text{sign}(\widetilde{f}_n(\boldsymbol{x}_i')) \neq \text{sign}(\eta(\boldsymbol{x}_i') - 1/2) \Big).$$

where $\boldsymbol{x}_i'$'s are testing samples generated in the same way as $\boldsymbol{x}_i$'s.

**Scenario I**. In the first scenario, we aim to verify that $\widehat{E}(\widetilde{f}_n)$ will converge to 0 as sample size $n$ increases when the privacy parameter is a fixed constant. To this end, we consider cases $(n, \epsilon) = \{100 \times 2^i, i = 0, 1, \ldots, 8\} \times \{1, 2, 3, 4, \infty\}$.

**Scenario II**. In the second scenario, we explore the asymptotic behavior of $\widehat{E}(\widetilde{f}_n)$ with $\epsilon$ adaptive to the sample size $n$. Specifically, we set $\epsilon = 5n^{-\zeta}$ and consider cases $(n, \zeta) = \{100 \times 2^i, i = 0, 1, \ldots, 8\} \times \{1/5, 1/4, 1/3, 1/2, 2/3, 1\}$. We also include the worst case $\epsilon = 0$ as a baseline.

**Scenario III**. In the third scenario, we intend to verify that $\epsilon \asymp n^{-1/2}$ is the dividing line between whether or not the excess risk converges. To this end, we consider three kinds of adaptive $\epsilon$, including $\epsilon \asymp n^{-1/2}$, $\epsilon \asymp \log(n)n^{-1/2}$, and $\epsilon \asymp \log^{-1}(n)n^{-1/2}$. The size of $\mathcal{D}$ is set as $\{100 \times 3^i, i = 0, 1, \ldots, 7\}$. For all cases, we report the averaged empirical excess risk in 1,000 replications as well as their 95% confidence intervals.

For Scenario I and Scenario II, we report the averaged empirical excess risk and the classification error in 1,000 replications for each setting in Figure 2 and Figure 3, respectively. From the left panel of Figure 2, we can see that the empirical excess risks and the classification errors with a fixed $\epsilon$ converge to 0 regardless of the value of $\epsilon$, showing that the randomized response mechanism with a fixed $\epsilon$ fails to prevent the third party from learning the optimal classifier based on $\widetilde{\mathcal{D}}$. Moreover, as seen from Figure 3, when $\zeta < 1/2$ the estimated excess risks present a decreasing pattern as the sample size increases, whereas that of the case $\zeta = 1$ deteriorates steadily and the curve finally overlaps with that of the worst case $\epsilon = 0$. It is also interesting to observe that the curve of $\zeta = 1/2$ remains unaffected by the sample size. All these phenomenons are in accordance with the results of Theorem 1.

As can be seen in Figure 4, the curve of the case $\epsilon \asymp n^{-1/2}$ remains unchanged as sample size increases as in Scenario II. This is due to the offset of information gain yielded by increasing the sample size and the

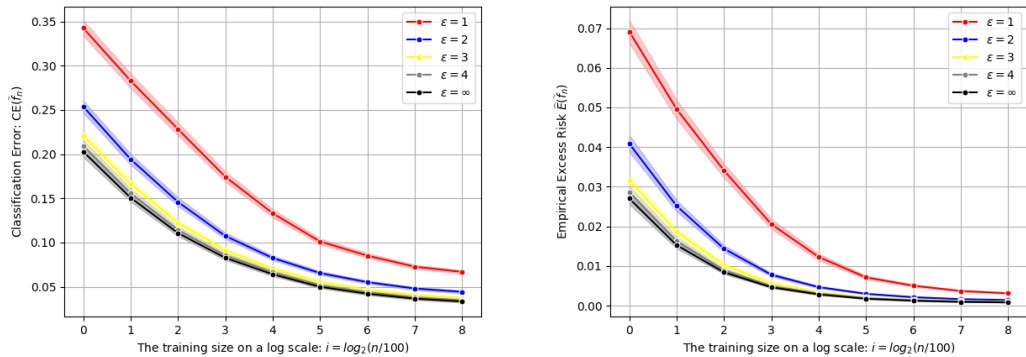

Figure 2: The averaged classification errors (Left) and averaged empirical excess risks (Right) of all settings with $n_{test} = 50,000$ in Scenario I.

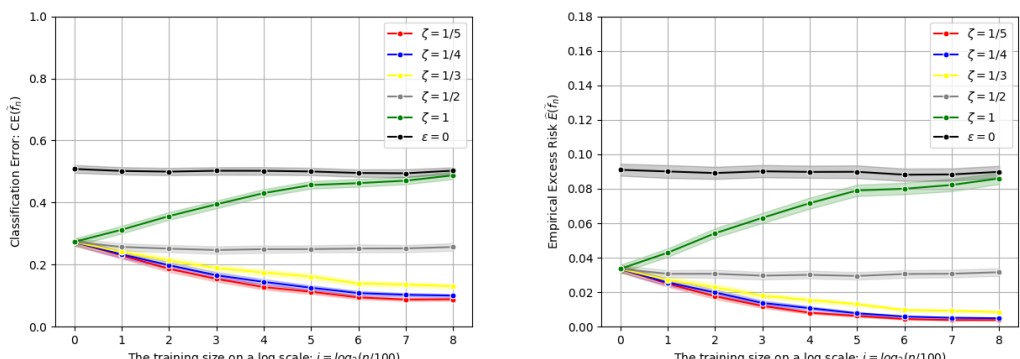

Figure 3: The averaged classification errors (Left) and averaged empirical excess risks (Right) of all settings with $n_{test} = 50,000$ in Scenario II.

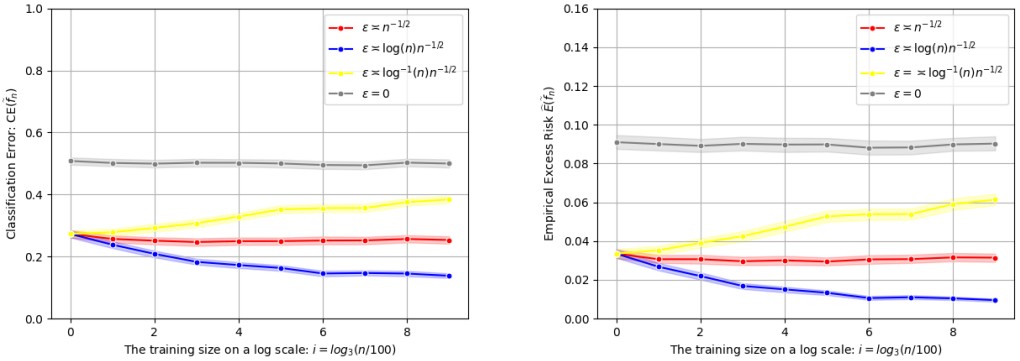

Figure 4: The averaged classification errors (Left) and averaged empirical excess risks (Right) of all settings with $n_{test} = 50,000$ in Scenario III.

information loss in the label-flipping mechanism. As expected, the additional logarithmic term significantly alters the original curve pattern. Specifically, for the case with $\epsilon \asymp \log^{-1}(n)n^{-1/2}$, the performance of $\widetilde{f}_n$ deteriorates significantly and approaches the worst case with $\epsilon = 0$. On the contrary, the performance of $\widetilde{f}_n$ in the case $\epsilon \asymp \log(n)n^{-1/2}$ improves significantly with $\widehat{E}(\widetilde{f}_n)$ converging to 0. Therefore, $\epsilon \asymp n^{-1/2}$ appears

to be the dividing line that determines whether the excess risk converges, which completely matches our theoretical results.

## 5.2 Deep Neural Network Classifier

The simulation in this section aims to verify our theoretical results in Theorem 2, which mainly lies in two aspects. First, we intend to verify the effect of $\epsilon$ of Label DP on the optimal structure of deep neural networks for classification problems. Specifically, as stated in Theorem 2, if label noise yielded by the randomized response mechanism increases, a smaller deep neural network should be employed to strike a better balance between the approximation error and the estimation error to achieve a better generalization error. Second, the consistency in estimating the decision boundary is prohibited provided that $\epsilon$ is adaptive to the training size $n$ at some specific orders. To these ends, we consider generating training datasets $\widetilde{\mathcal{D}} = \{(\boldsymbol{x}_i, \widetilde{y}_i)\}_{i=1}^n$ as follows. First, we set the regression function as $\boldsymbol{\eta}_{nn}(\boldsymbol{x}_i) = \sum_{j=1}^4 \sin(2\pi x_{ij})/8 + 1/2$ with $x_{ij} \sim \text{Unif}(0, 1)$ for any $i, j$. Then we generate $y_i$ from $\{1, -1\}$ with probabilities $\boldsymbol{\eta}_{nn}(\boldsymbol{x}_i)$ and $1 - \boldsymbol{\eta}_{nn}(\boldsymbol{x}_i)$, respectively. As in the last simulation, we then apply the randomized response mechanism $\mathcal{A}_\theta$ to each $y_i$ to generate $\widetilde{y}_i = \mathcal{A}_\theta(y_i)$ with $\theta = \exp(\epsilon)/(1 + \exp(\epsilon))$. Then we set the hypothesis space to be the class of $L$-layer fully connected neural network with equal width and the ReLU activation function, where $h$ denotes the widths in all hidden layers.

The overall training process of the neural network is implemented in Tensorflow (Abadi et al., 2016) with the Adam optimizer and learning rate being 0.001. Additionally, we employ the early-stopping technique to monitor the training error with patience 10 and maintain the parameter with the smallest training error. Let $\widetilde{f}_{nn}$ denote the resultant neural network obtained from minimizing (4). We construct the associated plug-in classifier as $\widetilde{s}_{nn} = \text{sign}(\widetilde{f}_{nn} - 1/2)$ and evaluate its performance by the empirical excess risk and the classification error as in Section 5.1.

**Scenario I**. In the first scenario, we consider privacy guarantees $\epsilon \in \{1, 2, \infty\}$ and neural network structures with $L = 2$ and $h \in \{8, 12, 16, 20, 24\}$ with . We report the averaged empirical excess risks and the classification errors of all cases in 100 replications as well as their 95% confidence intervals in Figure 5. Clearly, if $\epsilon = 1$, the optimal neural network structure is $h = 8$, whereas those of the cases $\epsilon = 2$ and $\epsilon = \infty$ are $h = 20$ and $h = 24$, respectively. Such results show that a smaller neural network is preferred when stronger privacy protection of labels (smaller $\epsilon$) is considered, which coincides with our theoretical results in Theorem 2 that the optimal neural network structure to achieve the fastest convergence rate of excess risk should diminish as $\epsilon$ decreases. Moreover, as the training sample size $n$ increases from 2,000 to 4,000, the optimal structure of the neural network enlarges for the cases $\epsilon = 1$ and $\epsilon = 2$ due to their new tradeoffs between the estimation and approximation errors.

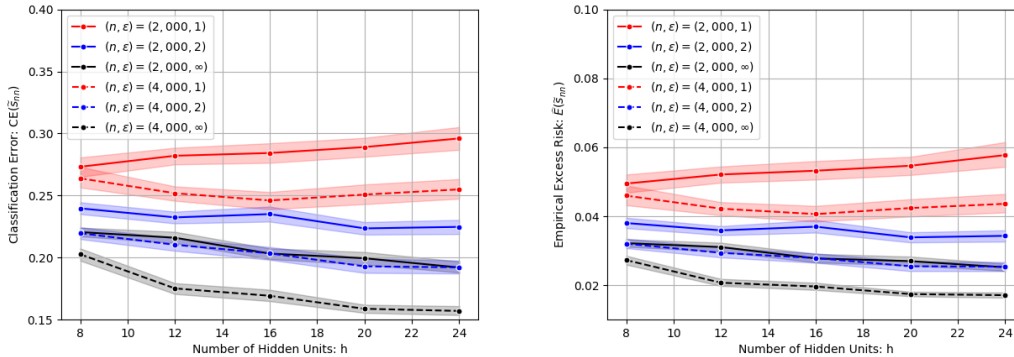

Figure 5: The averaged classification errors (Left) and averaged empirical excess risks (Right) of all cases with $n_{test} = 50,000$.

**Scenario II**. In the second scenario, we fix the neural network structure as $(L, h) = (2, 16)$ and consider training sample sizes $n \in \{2^i \times 10^3; i = 0, 1, 2, 3, 4\}$. To verify our theoretical results, we compare two privacy

schemes $\epsilon \asymp n^{-1/2}$ and $\epsilon \asymp \log(n)n^{-1/2}$. We also include the case with invariant privacy scheme $\epsilon = 4$ as a baseline. For comparing their difference in trend patterns, the multiplicative constants of two adaptive privacy schemes are chosen such that they have the same starting point as $\epsilon = 4$ when $n = 1,000$. We report the averaged empirical excess risks, the classification errors of all cases in 100 replications, and their 95% confidence intervals in Figure 6. Clearly, the performances of the cases $\epsilon = 4$ and $\epsilon \asymp \log(n)n^{-1/2}$ improve as $n$ increases. However, the performance of case $\epsilon \asymp n^{-1/2}$ presents a different pattern in generalization performance as $n$ increases. Most notably, as $n$ increases from 8,000 to 16,000, it performance deteriorates while the other two cases still observe significant improvements, showing that the additional logarithmic term plays a deterministic role in the convergence of excess risk, which perfectly aligns with our theoretical findings in Theorem 2.

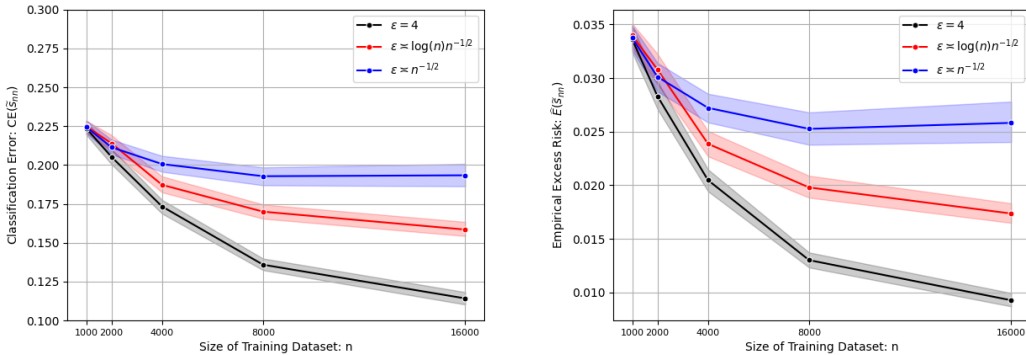

Figure 6: The averaged classification errors (Left) and averaged empirical excess risks (Right) of all cases with $n_{test} = 50,000$.

# 6 Real Applications - Mnist Dataset

This experiment considers similar settings of the privacy parameter $\epsilon$ as Section 5.2 in order to verify our theoretical findings of DNN on the MNIST dataset (LeCun, 1998). The MNIST dataset consists of 60,000 training images and 10,000 testing images, and each sample is a $28 \times 28$ grey-scale pixel image of one of the 10 digits. In this experiment, we consider a binary classification problem by only including samples of digits 2 and 3 for training and testing due to their similarity in appearance. The resultant training dataset contains 12,089 training samples and 2,042 testing samples.

For the hyperparameters setting, we consider the neural network with three convolution layers with ReLU activation and two fully-connected layers. For the three convolution layers, we set their kernel sizes and numbers of channels as $4 \times 4$ and 4, respectively. Additionally, each convolution layer is followed by a max pooling layer with size $2 \times 2$. The first fully-connected layer has 10 hidden units with ReLU activation, and the last layer outputs the probability of an image being digit 2. As in Section 5.2, the neural network is trained with the Adam optimizer and learning rate being 0.001, and the early-stopping technique to monitor the training error with patience 10 and maintain the parameter with the smallest training error. We evaluate of the trained model by the testing error on 2,042 testing samples. We consider training sample size $n \in \{2 \times i \times 10^3; i = 1, 2, 3, 4, 5\}$. We mainly consider two scenarios, including the fixed privacy guarantee with $\epsilon \in \{1, 2, \infty\}$ and the adaptive privacy schemes with $\epsilon \asymp \log(n)\sqrt{n^{-1}}$, $\epsilon \asymp n^{-1/2}$, and $\epsilon \asymp \log^{-1}(n)n^{-1/2}$. The averaged testing error of each case in 50 replications is reported in Figure 7.

Figure 7 presents similar results as in Section 5.2. First, when $\epsilon$ is fixed, differentially private classifiers improve in generalization performance as the training sample size increases and attain competitive performance as the non-private classifier ($\epsilon = \infty$) when the training sample size is large enough. This result accords with our theoretical findings in Theorem 1 that fixed privacy guarantee in the label DP slows down the convergence to the optimal classifier with a multiplicative constant. In stark contrast, as shown in the right plot of Figure 7, the convergence to the optimal classifier is prevented if $\epsilon \asymp n^{-1/2}$, as boosting the

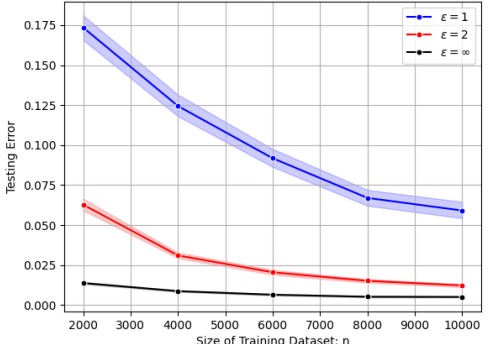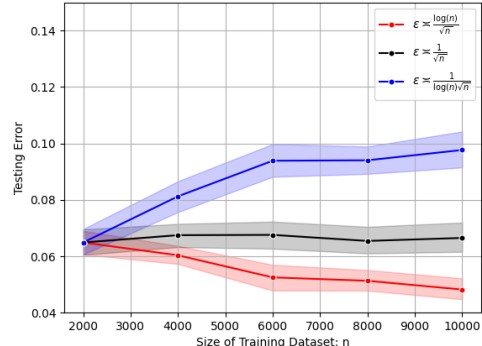

Figure 7: The averaged testing errors with fixed $\epsilon$ (Left) and adaptive $\epsilon$ (Right) under different training sample sizes in MNIST dataset

training size from 2,000 to 10,000 fails to significantly improve the testing error. This again aligns with our Theorem 1

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

# A  Appendix

**Lemma 1** (Restatement of Lemma 1)**.** *Suppose the randomized response mechanism $\widetilde{Y} = \mathcal{A}_\theta(Y)$ satisfies $\epsilon$-Label DP with $\theta = \exp(\epsilon)/(1 + \exp(\epsilon))$, then for any $\boldsymbol{x} \in \mathcal{X}$ it holds that*

$$\mathcal{L}(\epsilon, \boldsymbol{x}) \leq D_{KL}\big(\mathbb{P}_{Y|\boldsymbol{X}=\boldsymbol{x}}\big|\mathbb{P}_{\widetilde{Y}|\boldsymbol{X}=\boldsymbol{x}}\big) \leq \mathcal{U}(\epsilon, \boldsymbol{x}),$$

*where $D_{KL}\big(\mathbb{P}_{Y|\boldsymbol{X}=\boldsymbol{x}}\big|\mathbb{P}_{\widetilde{Y}|\boldsymbol{X}=\boldsymbol{x}}\big)$ denotes the Kullback-Leibler divergence (KL) divergence between $\mathbb{P}_{\widetilde{Y}|\boldsymbol{X}=\boldsymbol{x}}$ and $\mathbb{P}_{Y|\boldsymbol{X}=\boldsymbol{x}}$, $\mathcal{L}(\epsilon, \boldsymbol{x}) = 2(2\eta(\boldsymbol{x}) - 1)^2(1 + \exp(\epsilon))^{-2}$, and $\mathcal{U}(\epsilon, \boldsymbol{x}) = \min\{U_1(\epsilon, \boldsymbol{x}), U_2(\epsilon, \boldsymbol{x})\}$ with $U_1(\epsilon, \boldsymbol{x}) = (2\eta(\boldsymbol{x}) - 1)^2\eta^{-1}(\boldsymbol{x})(1 - \eta(\boldsymbol{x}))^{-1}(1 + \exp(\epsilon))^{-2}$ and $U_2(\epsilon, \boldsymbol{x}) = (2\eta(\boldsymbol{x}) - 1)^2 \exp(-\epsilon)$.*

**Proof of Lemma 1**: We first prove the lower bound of $D_{KL}\big(\mathbb{P}_{Y|\boldsymbol{X}=\boldsymbol{x}}\big|\mathbb{P}_{\widetilde{Y}|\boldsymbol{X}=\boldsymbol{x}}\big)$, which is mainly based on the Pinsker's inequality (Sason & Verdú, 2016). Without loss of generality, we assume that $\eta(\boldsymbol{x}) > 1/2$. Define

$$S(p, q) = p \log \frac{p}{q} + (1 - p) \log \frac{1 - p}{1 - q} - 2(p - q)^2,$$

where $p, q \in [0, 1]$. Let $S(p, q)$ take the partial derivative with respect to $q$, we get

$$\frac{\partial S(p, q)}{\partial q} = -\frac{p}{q} + \frac{1 - p}{1 - q} + 4(p - q) = -(p - q)(\frac{1}{q(1 - q)} - 4) \leq 0, \text{for } q \leq p,$$

where the last inequality follows from that fact that $q(1 - q) \leq 1/4$ for $q \in [0, 1]$ and the equality holds when $p = q$. Suppose that $\mathcal{A}_\theta$ satisfies $\epsilon$-Label DP, which is equivalent to set $\theta = \exp(\epsilon)/(1 + \exp(\epsilon))$. Therefore, it holds that

$$D_{KL}\Big(\mathbb{P}_{Y|\boldsymbol{X}=\boldsymbol{x}}\big|\mathbb{P}_{\widetilde{Y}|\boldsymbol{X}=\boldsymbol{x}}\Big) \geq 2(\eta(\boldsymbol{x}) - \widetilde{\eta}(\boldsymbol{x}))^2 = 2(2\eta(\boldsymbol{x}) - 1)^2(1 + \exp(\epsilon))^{-2} \triangleq \mathcal{L}(\epsilon, \boldsymbol{x}).$$

Next, we proceed to prove the upper bound. For any pair of distribution $\mathbb{P}$ and $\mathbb{Q}$, we have

$$D_{KL}(\mathbb{P}|\mathbb{Q}) = D_{KL}(\mathbb{P}|\mathbb{Q}) + 1 - 1 \leq \exp(D_{KL}(\mathbb{P}|\mathbb{Q})) - 1.$$

Recall that $Y$ and $\widetilde{Y}$ take values in $\{-1, 1\}$, we have

$$
\begin{aligned}
D_{KL}\Big(\mathbb{P}_{Y|\boldsymbol{X}=\boldsymbol{x}}\big|\mathbb{P}_{\widetilde{Y}|\boldsymbol{X}=\boldsymbol{x}}\Big) &\leq \exp\big(D_{KL}\big(\mathbb{P}_{Y|\boldsymbol{X}=\boldsymbol{x}}\big|\mathbb{P}_{\widetilde{Y}|\boldsymbol{X}=\boldsymbol{x}}\big)\big) - 1 \\
&\leq \eta(\boldsymbol{x})\frac{\eta(\boldsymbol{x})}{\widetilde{\eta}(\boldsymbol{x})} + (1-\eta(\boldsymbol{x}))\frac{1-\eta(\boldsymbol{x})}{1-\widetilde{\eta}(\boldsymbol{x})} - 1 = \eta(\boldsymbol{x})\Big(\frac{\eta(\boldsymbol{x})}{\widetilde{\eta}(\boldsymbol{x})}-1\Big) + (1-\eta(\boldsymbol{x}))\Big(\frac{1-\eta(\boldsymbol{x})}{1-\widetilde{\eta}(\boldsymbol{x})}-1\Big) \\
&= \eta(\boldsymbol{x})\Big(\frac{\eta(\boldsymbol{x})-\widetilde{\eta}(\boldsymbol{x})}{\widetilde{\eta}(\boldsymbol{x})}\Big) + (1-\eta(\boldsymbol{x}))\Big(\frac{\widetilde{\eta}(\boldsymbol{x})-\eta(\boldsymbol{x})}{1-\widetilde{\eta}(\boldsymbol{x})}\Big) = \frac{(\eta(\boldsymbol{x})-\widetilde{\eta}(\boldsymbol{x}))^2}{\widetilde{\eta}(\boldsymbol{x})(1-\widetilde{\eta}(\boldsymbol{x}))}.
\end{aligned}
$$

Note that

$$
\begin{aligned}
\widetilde{\eta}(\boldsymbol{x})(1-\widetilde{\eta}(\boldsymbol{x})) &= \Big(\theta\eta(\boldsymbol{x}) + (1-\theta)(1-\eta(\boldsymbol{x}))\Big)\Big((1-\theta)\eta(\boldsymbol{x}) + \theta(1-\eta(\boldsymbol{x}))\Big) \\
&= \theta(1-\theta)\eta^2(\boldsymbol{x}) + \eta(\boldsymbol{x})(1-\eta(\boldsymbol{x}))\big(\theta^2 + (1-\theta)^2\big) + \theta(1-\theta)(1-\eta(\boldsymbol{x}))^2 \\
&\geq \max\{\eta(\boldsymbol{x})(1-\eta(\boldsymbol{x})), \theta(1-\theta)\}.
\end{aligned}
$$

It then follows that

$$
D_{KL}\Big(\mathbb{P}_{Y|\boldsymbol{X}=\boldsymbol{x}}\big|\mathbb{P}_{\widetilde{Y}|\boldsymbol{X}=\boldsymbol{x}}\Big) \leq \frac{(\eta(\boldsymbol{x})-\widetilde{\eta}(\boldsymbol{x}))^2}{\max\{\eta(\boldsymbol{x})(1-\eta(\boldsymbol{x})), \theta(1-\theta)\}}. \tag{6}
$$

Plugging $\theta = \exp(\epsilon)/(1+\exp(\epsilon))$ into (6) yields that

$$
D_{KL}\Big(\mathbb{P}_{Y|\boldsymbol{X}=\boldsymbol{x}}\big|\mathbb{P}_{\widetilde{Y}|\boldsymbol{X}=\boldsymbol{x}}\Big) \leq \frac{(2\eta(\boldsymbol{x})-1)^2}{\max\{\eta(\boldsymbol{x})(1-\eta(\boldsymbol{x}))(1+\exp(\epsilon))^2, \exp(\epsilon)\}} \triangleq \mathcal{U}(\theta, \boldsymbol{x}).
$$

This completes the proof. $\qquad\square$

**Lemma 2** (Restatement of Lemma 2). *If $\widetilde{Y} = \mathcal{A}_\theta(Y)$ with $\theta > 1/2$, then $f^*(\boldsymbol{x}) = \widetilde{f}^*(\boldsymbol{x})$ for any $\boldsymbol{x} \in \mathcal{X}$ and $\widetilde{D}(f, \widetilde{f}^*) = (2\theta - 1)D(f, f^*)$ for any $f$.*

**Proof of Lemma 2**: By the definition of $\widetilde{\eta}(\boldsymbol{x})$, we can obtain

$$
2\widetilde{\eta}(\boldsymbol{x}) - 1 = (2\theta - 1)(2\eta(\boldsymbol{x}) - 1). \tag{7}
$$

Clearly, $2\eta(\boldsymbol{x}) > 1$ indicates that $2\widetilde{\eta}(\boldsymbol{x}) > 1$ provided that $\theta > 1/2$. By the fact that $f^*(\boldsymbol{x}) = \text{sign}(\eta(\boldsymbol{x})-1/2)$, it holds that $f^*(\boldsymbol{x}) = \widetilde{f}^*(\boldsymbol{x})$ for any $\boldsymbol{x} \in \mathcal{X}$. Recall that the excess risk in terms of 0-1 loss can be written as

$$
R(f) - R(f^*) = \mathbb{E}\big[|\text{sign}(f(\boldsymbol{X})) \neq \text{sign}(f^*(\boldsymbol{X}))||2\eta(\boldsymbol{X}) - 1|\big].
$$

Combined with (7), it holds that

$$
\begin{aligned}
\widetilde{R}(f) - \widetilde{R}(\widetilde{f}^*) &= \mathbb{E}\big[|\text{sign}(f(\boldsymbol{X})) \neq \text{sign}(\widetilde{f}^*(\boldsymbol{X}))||2\widetilde{\eta}(\boldsymbol{X}) - 1|\big] \\
&= (2\theta - 1)\mathbb{E}\big[|\text{sign}(f(\boldsymbol{X})) \neq \text{sign}(\widetilde{f}^*(\boldsymbol{X}))||2\eta(\boldsymbol{X}) - 1|\big] \\
&= (2\theta - 1)\big(R(f) - R(f^*)\big).
\end{aligned}
$$

This completes the proof. $\qquad\square$

**Lemma 3** (Restatement of Lemma 3). *Denote that $\kappa_\epsilon = (e^\epsilon - 1)/(e^\epsilon + 1)$. Under Assumption 1, for any $\theta \in (1/2, 1]$, it holds that*

*(1) $\mathbb{P}\Big(|2\widetilde{\eta}(\boldsymbol{X}) - 1| \leq t\Big) \leq c\kappa_\epsilon^{-\gamma}t^\gamma$, for any $t \in [0, 1)$,*

*(2) $\widetilde{R}_\phi(f) - \widetilde{R}_\phi(f^*) \geq \kappa_\epsilon(4c)^{-1/\gamma}\Big(\mathbb{E}\big[|f(\boldsymbol{X}) - f^*(\boldsymbol{X})|\big]\Big)^{\frac{\gamma+1}{\gamma}}$ for any $f \in \mathcal{F}$,*

**Proof of Lemma 3**: By the assumption that $\|f\|_\infty \leq 1$ and the fact that 0-1 loss is upper bounded by hinge loss, we obtain $R_\phi(f) - R_\phi(f^*) \geq R(f) - R(f^*)$. Since $\phi$ is set as hinge loss, it holds that

$$R_\phi(f) - R_\phi(f^*) = \mathbb{E}_{\boldsymbol{X}}\big[|f(\boldsymbol{X}) - f^*(\boldsymbol{X})||1 - 2\eta(\boldsymbol{X})|\big]$$
$$\geq t\mathbb{E}_{\boldsymbol{X}}\big[|f(\boldsymbol{X}) - f^*(\boldsymbol{X})|I\big(|1 - 2\eta(\boldsymbol{X})| > t\big)\big].$$

By the fact that $I\big(|1 - 2\eta(\boldsymbol{X})| > t\big) = 1 - I\big(|1 - 2\eta(\boldsymbol{X})| \leq t\big)$, it then follows that

$$R_\phi(f) - R_\phi(f^*) \geq t\Big(\mathbb{E}_{\boldsymbol{X}}\big[|f(\boldsymbol{X}) - f^*(\boldsymbol{X})|\big] - 2\mathbb{P}\big(|1 - 2\eta(\boldsymbol{X})| \leq t\big)\Big)$$
$$\geq t\Big(\mathbb{E}_{\boldsymbol{X}}\big[|f(\boldsymbol{X}) - f^*(\boldsymbol{X})|\big] - 2ct^\gamma\Big)$$
$$= t\mathbb{E}_{\boldsymbol{X}}\big[|f(\boldsymbol{X}) - f^*(\boldsymbol{X})|\big] - 2ct^{\gamma+1}$$

where the last inequality follows from Assumption 1. Choosing $t$ such that $t\mathbb{E}\big[|f(\boldsymbol{X}) - f^*(\boldsymbol{X})|\big] = 4ct^{\gamma+1}$, we get

$$R_\phi(f) - R_\phi(f^*) \geq (4c)^{-1/\gamma}\Big(\mathbb{E}\big[|f(\boldsymbol{X}) - f^*(\boldsymbol{X})|\big]\Big)^{\frac{\gamma+1}{\gamma}}.$$

Next, we proceed to establish the relation between $\widetilde{R}_\phi(f) - \widetilde{R}_\phi(f^*)$ and $\mathbb{E}\big[|f(\boldsymbol{X}) - f^*(\boldsymbol{X})|\big]$. For each $\boldsymbol{x} \in \mathcal{X}$, the conditional risk is given as

$$\mathbb{E}_{\widetilde{Y}}\Big(\phi(f(\boldsymbol{X})\widetilde{Y})|\boldsymbol{X} = \boldsymbol{x}\Big) = \widetilde{\eta}(\boldsymbol{x})\phi(f(\boldsymbol{x})) + (1 - \widetilde{\eta}(\boldsymbol{x}))\phi(-f(\boldsymbol{x}))$$
$$= [\theta\eta(\boldsymbol{x}) + (1 - \theta)(1 - \eta(\boldsymbol{x}))]\phi(f(\boldsymbol{x})) + [\theta(1 - \eta(\boldsymbol{x})) + (1 - \theta)\eta(\boldsymbol{x})]\phi(-f(\boldsymbol{x}))$$
$$= \Big(\theta\eta(\boldsymbol{x})\phi(f(\boldsymbol{x})) + \theta(1 - \eta(\boldsymbol{x}))\phi(-f(\boldsymbol{x}))\Big) + \Big((1 - \theta)(1 - \eta(\boldsymbol{x}))\phi(f(\boldsymbol{x})) + (1 - \theta)\eta(\boldsymbol{x})\phi(-f(\boldsymbol{x}))\Big).$$

Taking the expectation with respect to $\boldsymbol{X}$ yields that

$$\widetilde{R}_\phi(f) = \mathbb{E}_{(\boldsymbol{X},\widetilde{Y})}\Big(\phi(f(\boldsymbol{X})\widetilde{Y})\Big) = \theta R_\phi(f) + (1 - \theta)R_\phi(-f) \tag{8}$$

Notice that $\phi$ is hinge loss, hence

$$R_\phi(f) - R_\phi(f^*) = R_\phi(-f^*) - R_\phi(-f),$$

for any $f$ with $\|f\|_\infty \leq 1$. It follows that

$$\widetilde{R}_\phi(f) - \widetilde{R}_\phi(f^*) = \theta(R_\phi(f) - R_\phi(f^*)) + (1 - \theta)(R_\phi(-f) - R_\phi(-f^*))$$
$$= (2\theta - 1)(R_\phi(f) - R_\phi(f^*))$$
$$\geq (2\theta - 1)(4c)^{-1/\gamma}\Big(\mathbb{E}\big[|f(\boldsymbol{X}) - f^*(\boldsymbol{X})|\big]\Big)^{\frac{\gamma+1}{\gamma}}.$$

This completes the proof. □

**Lemma 4** (Restatement of Lemma 4). *Denote that* $\mathcal{M} = \max\{EFNE(f) - EFNE(f^*), EFPE(f) - EFPE(f^*)\}$. *Under Assumption 1, for any margin classifier $f$, it holds that*

$$R(f) - R(f^*) \leq \mathcal{M} \leq \frac{1}{2\min\{\mathbb{P}(Y = 1), \mathbb{P}(Y = -1)\}}\Big(2c^{-\frac{1}{1+\gamma}}(R(f) - R(f^*))^{\frac{\gamma}{1+\gamma}} + R(f) - R(f^*)\Big),$$

*where $c$ and $\gamma$ are as defined in Assumption 1. Particularly, if Assumption 1 holds with $\gamma = \infty$, for any margin classifier $f$, (3) becomes*

$$R(f) - R(f^*) \leq \mathcal{M} \leq \frac{3}{2\min\{\mathbb{P}(Y = 1), \mathbb{P}(Y = -1)\}}\big(R(f) - R(f^*)\big).$$

**Proof of Lemma 4**: We first prove the left-hand side of (3). By the relation among $R(f)$, $\text{EFNE}(f)$, and $\text{EFPE}(f)$, one has

$$
\begin{aligned}
R(f) - R(f^*) &= \big(\text{EFNE}(f) - \text{EFNE}(f^*)\big)\mathbb{P}(Y = 1) + \big(\text{EFPE}(f) - \text{EFPE}(f^*)\big)\mathbb{P}(Y = -1) \\
&\leq \max\{\text{EFNE}(f) - \text{EFNE}(f^*), \text{EFPE}(f) - \text{EFPE}(f^*)\}(\mathbb{P}(Y = 1) + \mathbb{P}(Y = -1)) \\
&= \max\{\text{EFNE}(f) - \text{EFNE}(f^*), \text{EFPE}(f) - \text{EFPE}(f^*)\}.
\end{aligned}
$$

Next, we prove the right-hand side of (3). We first define

$$
S_1(f) = \{\boldsymbol{x} \in \mathcal{X} : \text{sign}(f(\boldsymbol{x})) = 1\} \text{ and } S_{-1}(f) = \{\boldsymbol{x} \in \mathcal{X} : \text{sign}(f(\boldsymbol{x})) = -1\}.
$$

Clearly, it can be verified that $S_1(f^*) = \{\boldsymbol{x} : \eta(\boldsymbol{x}) > 1/2\}$ and $S_{-1}(f^*) = \{\boldsymbol{x} : \eta(\boldsymbol{x}) < 1/2\}$. With these, we further get

$$
\begin{aligned}
\text{EFNE}(f) - \text{EFNE}(f^*) &= \frac{1}{\mathbb{P}(Y = 1)}\left(\int_{S_{-1}(f)} \eta(\boldsymbol{x})\mathbb{P}_{\boldsymbol{X}}(\boldsymbol{x})d\boldsymbol{x} - \int_{S_{-1}(f^*)} \eta(\boldsymbol{x})\mathbb{P}_{\boldsymbol{X}}(\boldsymbol{x})d\boldsymbol{x}\right) \\
&= \frac{1}{\mathbb{P}(Y = 1)}\left(\int_{S_{-1}(f)\backslash\triangle S_{-1}} \eta(\boldsymbol{x})\mathbb{P}_{\boldsymbol{X}}(\boldsymbol{x})d\boldsymbol{x} - \int_{S_{-1}(f^*)\backslash\triangle S_{-1}} \eta(\boldsymbol{x})\mathbb{P}_{\boldsymbol{X}}(\boldsymbol{x})d\boldsymbol{x}\right), \\
\text{EFPE}(f) - \text{EFPE}(f^*) &= \frac{1}{\mathbb{P}(Y = -1)}\left(\int_{S_1(f)} (1 - \eta(\boldsymbol{x}))\mathbb{P}_{\boldsymbol{X}}(\boldsymbol{x})d\boldsymbol{x} - \int_{S_1(f^*)} (1 - \eta(\boldsymbol{x}))\mathbb{P}_{\boldsymbol{X}}(\boldsymbol{x})d\boldsymbol{x}\right) \\
&= \frac{1}{\mathbb{P}(Y = -1)}\left(\int_{S_1(f)\backslash\triangle S_1} (1 - \eta(\boldsymbol{x}))\mathbb{P}_{\boldsymbol{X}}(\boldsymbol{x})d\boldsymbol{x} - \int_{S_1(f^*)\backslash\triangle S_1} (1 - \eta(\boldsymbol{x}))\mathbb{P}_{\boldsymbol{X}}(\boldsymbol{x})d\boldsymbol{x}\right),
\end{aligned}
$$

where $\triangle S_{-1} = S_{-1}(f^*) \cap S_{-1}(f)$ and $\triangle S_1 = S_1(f^*) \cap S_1(f)$. We can easily verify that $S_{-1}(f) \setminus \triangle S_{-1} = S_1(f^*) \setminus \triangle S_1$ and $S_1(f) \setminus \triangle S_1 = S_{-1}(f^*) \setminus \triangle S_{-1}$. Therefore, it follows that

$$
\begin{aligned}
R(f) - R(f^*) &= \left(\int_{\triangle_1} \eta(\boldsymbol{x})\mathbb{P}_{\boldsymbol{X}}(\boldsymbol{x})d\boldsymbol{x} - \int_{\triangle_2} \eta(\boldsymbol{x})\mathbb{P}_{\boldsymbol{X}}(\boldsymbol{x})d\boldsymbol{x}\right) + \left(\int_{\triangle_2} (1 - \eta(\boldsymbol{x}))\mathbb{P}_{\boldsymbol{X}}(\boldsymbol{x})d\boldsymbol{x} - \int_{\triangle_1} (1 - \eta(\boldsymbol{x}))\mathbb{P}_{\boldsymbol{X}}(\boldsymbol{x})d\boldsymbol{x}\right) \\
&\geq 2\left(\int_{\triangle_1} \eta(\boldsymbol{x})\mathbb{P}_{\boldsymbol{X}}(\boldsymbol{x})d\boldsymbol{x} - \int_{\triangle_2} \eta(\boldsymbol{x})\mathbb{P}_{\boldsymbol{X}}(\boldsymbol{x})d\boldsymbol{x}\right) - \int_{\triangle_1 \cup \triangle_2} \mathbb{P}_{\boldsymbol{X}}(\boldsymbol{x})d\boldsymbol{x},
\end{aligned}
$$

where $\triangle_1 = S_{-1}(f) \setminus \triangle S_{-1}$ and $\triangle_2 = S_1(f) \setminus \triangle S_1$.

Then

$$
\begin{aligned}
&\int_{\triangle_1} \eta(\boldsymbol{x})\mathbb{P}_{\boldsymbol{X}}(\boldsymbol{x})d\boldsymbol{x} - \int_{\triangle_2} \eta(\boldsymbol{x})\mathbb{P}_{\boldsymbol{X}}(\boldsymbol{x})d\boldsymbol{x} \leq \int_{\triangle_1 \cup \triangle_2} \mathbb{P}_{\boldsymbol{X}}(\boldsymbol{x})d\boldsymbol{x} + R(f) - R(f^*) \\
&\leq \int_{\triangle_1 \cup \triangle_2} I(|2\eta(\boldsymbol{x}) - 1| \leq t)\mathbb{P}_{\boldsymbol{X}}(\boldsymbol{x})d\boldsymbol{x} + \int_{\triangle_1 \cup \triangle_2} I(|2\eta(\boldsymbol{x}) - 1| > t)\mathbb{P}_{\boldsymbol{X}}(\boldsymbol{x})d\boldsymbol{x} + R(f) - R(f^*) \\
&\leq ct^\gamma + \int_{\triangle_1 \cup \triangle_2} I(|2\eta(\boldsymbol{x}) - 1| > t)\frac{|2\eta(\boldsymbol{x}) - 1|}{t}\mathbb{P}_{\boldsymbol{X}}(\boldsymbol{x})d\boldsymbol{x} + R(f) - R(f^*) \\
&\leq ct^\gamma + t^{-1}(R(f) - R(f^*)) + R(f) - R(f^*)
\end{aligned}
$$

where the last inequality follows from the low-noise assumption. Choosing $t = (R(f) - R(f^*))^{\frac{1}{1+\gamma}}/c^{\frac{1}{1+\gamma}}$, we get

$$
\int_{\triangle_1} \eta(\boldsymbol{x})\mathbb{P}_{\boldsymbol{X}}(\boldsymbol{x})d\boldsymbol{x} - \int_{\triangle_2} \eta(\boldsymbol{x})\mathbb{P}_{\boldsymbol{X}}(\boldsymbol{x})d\boldsymbol{x} \leq 2c^{-\frac{1}{1+\gamma}}(R(f) - R(f^*))^{\frac{\gamma}{1+\gamma}} + R(f) - R(f^*)
$$

Finally, we have

$$
\text{EFNE}(f) - \text{EFNE}(f^*) \leq \frac{1}{2\mathbb{P}(Y = 1)}\left(2c^{-\frac{1}{1+\gamma}}(R(f) - R(f^*))^{\frac{\gamma}{1+\gamma}} + R(f) - R(f^*)\right). \tag{9}
$$

Using similar steps, we also have

$$\text{EFPE}(f) - \text{EFPE}(f^*) \leq \frac{1}{2\mathbb{P}(Y = -1)}\Big(2c^{-\frac{1}{1+\gamma}}(R(f) - R(f^*))^{\frac{\gamma}{1+\gamma}} + R(f) - R(f^*)\Big). \tag{10}$$

Combining (9) and (10) yields that

$$\max\{\text{EFNE}(f) - \text{EFNE}(f^*), \text{EFPE}(f) - \text{EFPE}(f^*)\}$$
$$\leq \frac{1}{2\min\{\mathbb{P}(Y = 1), \mathbb{P}(Y = -1)\}}\Big(2c^{-\frac{1}{1+\gamma}}(R(f) - R(f^*))^{\frac{\gamma}{1+\gamma}} + R(f) - R(f^*)\Big).$$

The desired results immediately follows by letting $\gamma$ go to infinity, which completes the proof. $\qquad\square$

**Theorem 1** (Restatement of Theorem 1). *Under Assumptions 1 and 2, for any minimizer $\widetilde{f}_n$ of (2), there exist some positive constants $A_1$ and $A_2$ such that*

$$A_2\Big\{\Big(\frac{\mathcal{V}_1(\Theta)}{n\kappa_\epsilon^2}\Big)^{\frac{\gamma+1}{\gamma+2}} + \tau_n\Big\} \leq \sup_{\pi \in \mathcal{P}_\gamma} \mathbb{E}_{\widetilde{\mathcal{D}}}\big[R(\widetilde{f}_n) - R(f^*)\big] \leq A_1\Big\{\Big(\frac{\mathcal{V}_1(\Theta)\log(n)}{n\kappa_\epsilon^2}\Big)^{\frac{\gamma+1}{\gamma+2}} + s_n\Big\},$$

*where $\mathcal{P}_\gamma$ be a class of distributions of $(\boldsymbol{X}, Y)$ satisfying Assumption 1, $s_n = \sup_{\pi \in \mathcal{P}_\gamma} \inf_{f \in \mathcal{F}} H(f, f^*)$, and $\tau_n = \sup_{\pi \in \mathcal{P}_\gamma} \inf_{f \in \mathcal{F}} D(f, f^*)$.*

**Proof of Theorem 1**: **Proof of the upper bound.** The proof of the upper bound mainly utilizes a uniform concentration inequality. We first denote that $\widetilde{H}(f, f^*) = \widetilde{R}_\phi(f) - \widetilde{R}_\phi(f^*)$. Next, we proceed to prove that $\mathbb{P}\big(\widetilde{H}(\widetilde{f}_n, f^*) \geq \delta_n\big)$ converges to zero as $n$ goes to infinity for some convergent sequence $\delta_n > 0$. For any $\delta_n > 0$, we let $\mathcal{F}_{\delta_n} = \{f \in \mathcal{F} : \widetilde{H}(f, f^*) \geq \delta_n\}$. If $\widetilde{f}_n \in \mathcal{F}_{\delta_n}$, one has

$$\sup_{f \in \mathcal{F}_{\delta_n}} \frac{1}{n}\sum_{i=1}^{n}\phi(f_\mathcal{F}^*(\boldsymbol{x}_i)\widetilde{y}_i) + \lambda_n J(f_\mathcal{F}^*) - \frac{1}{n}\sum_{i=1}^{n}\phi(f(\boldsymbol{x}_i)\widetilde{y}_i) - \lambda_n J(f) \geq 0.$$

This follows from the optimality of $\widetilde{f}_n$ for minimizing (2). Therefore,

$$\mathbb{P}\Big(\widetilde{H}(\widetilde{f}_n, f^*) \geq \delta_n\Big) \leq \mathbb{P}\left(\sup_{f \in \mathcal{F}_{\delta_n}} \frac{1}{n}\sum_{i=1}^{n}\phi(f_\mathcal{F}^*(\boldsymbol{x}_i)\widetilde{y}_i) + \lambda_n J(f_\mathcal{F}^*) - \frac{1}{n}\sum_{i=1}^{n}\phi(f(\boldsymbol{x}_i)\widetilde{y}_i) - \lambda_n J(f) \geq 0\right) \equiv I.$$

Next, it suffices to prove the convergence of $I$ with $n$. To this end, we proceed to provide an upper bound for $I$. Define that $\mathcal{H}_{ij} = \{f \in \mathcal{F} : 2^{i-1}\delta_n \leq \widetilde{H}(f, f^*) \leq 2^i\delta_n, 2^{j-1}J_0 \leq J(f) \leq 2^j J_0\}$ for $i \geq 1$ and $j \geq 1$ and $\mathcal{H}_{i0} = \{f \in \mathcal{F} : 2^{i-1}\delta_n \leq \widetilde{H}(f, f^*) \leq 2^i\delta_n, J(f) \leq J_0\}$. It can be easily verified that $\mathcal{F}_{\delta_n}$ admits the decomposition as $\mathcal{F}_{\delta_n} = \cup_{i=1}^{\infty} \cup_{j=0}^{\infty} \mathcal{H}_{ij}$. With this, $I$ can be upper bounded as

$$I = \mathbb{P}\left(\sup_{f \in \mathcal{F}_{\delta_n}} \frac{1}{n}\sum_{i=1}^{n}\phi(f_\mathcal{F}^*(\boldsymbol{x}_i)\widetilde{y}_i) + \lambda_n J(f_\mathcal{F}^*) - \frac{1}{n}\sum_{i=1}^{n}\phi(f(\boldsymbol{x}_i)\widetilde{y}_i) - \lambda_n J(f) \geq 0\right)$$
$$\leq \sum_{i=1}^{\infty}\sum_{j=0}^{\infty}\mathbb{P}\left(\sup_{f \in \mathcal{H}_{ij}} \frac{1}{n}\sum_{i=1}^{n}\phi(f_\mathcal{F}^*(\boldsymbol{x}_i)\widetilde{y}_i) + \lambda_n J(f_\mathcal{F}^*) - \frac{1}{n}\sum_{i=1}^{n}\phi(f(\boldsymbol{x}_i)\widetilde{y}_i) - \lambda_n J(f) \geq 0\right)$$
$$\leq \sum_{i=1}^{\infty}\sum_{j=0}^{\infty}\mathbb{P}\left(\sup_{f \in \mathcal{H}_{ij}} \frac{1}{n}\sum_{i=1}^{n}l_\phi(f_\mathcal{F}^*, z_i) - \frac{1}{n}\sum_{i=1}^{n}l_\phi(f, z_i) \geq \lambda_n\Big(\inf_{f \in \mathcal{H}_{ij}} J(f) - J_0\Big) + \inf_{f \in \mathcal{H}_{ij}}\mathbb{E}(\phi(f(\boldsymbol{X}_i)\widetilde{Y}_i)) - \mathbb{E}(\phi(f_\mathcal{F}^*(\boldsymbol{X}_i)\widetilde{Y}_i))\right),$$

where $z_i = (\boldsymbol{x}_i, \widetilde{y}_i)$ and $l_\phi(f, z_i) = \phi(f(\boldsymbol{x}_i)\widetilde{y}_i) - \mathbb{E}(\phi(f(\boldsymbol{X}_i)\widetilde{Y}_i))$. Here it is important to note that

$$\inf_{f \in \mathcal{H}_{ij}}\mathbb{E}(\phi(f(\boldsymbol{X}_i)\widetilde{Y}_i)) - \mathbb{E}(\phi(f_\mathcal{F}^*(\boldsymbol{X}_i)\widetilde{Y}_i)) = \inf_{f \in \mathcal{H}_{ij}}\widetilde{R}_\phi(f) - \widetilde{R}_\phi(f^*) - \widetilde{R}_\phi(f_\mathcal{F}^*) + \widetilde{R}_\phi(f^*)$$
$$= \inf_{f \in \mathcal{H}_{ij}}\widetilde{H}(f, f^*) + \widetilde{H}(f_\mathcal{F}^*, f^*) = \inf_{f \in \mathcal{H}_{ij}}\widetilde{H}(f, f^*) + (2\theta - 1)H(f_\mathcal{F}^*, f^*).$$

Let $V(i,j) = \lambda_n\big(\inf_{f\in\mathcal{H}_{ij}} J(f) - J_0\big) + \inf_{f\in\mathcal{H}_{ij}} \mathbb{E}(l_\phi(f, Z)) - \mathbb{E}(l_\phi(f_\mathcal{F}^*, Z))$. By the definition of $\mathcal{H}_{ij}$, we get

$$V(i,j) \geq M(i,j) = \lambda_n(2^{j-1} - 1)J_0 + (2^{i-1} - 1/4)\delta_n, \text{ for } i, j \geq 1, \tag{11}$$

where the inequality follows by assuming that $(2\theta - 1)s_n \leq 1/4\delta_n$. Next, we suppose that $\lambda_n J_0 \leq 1/4\delta_n$, we further have

$$M(i,0) \geq (2^{i-1} - 1/2)\delta_n \geq 2^{i-2}\delta_n, \text{ for } i \geq 1. \tag{12}$$

Plugging (11) and (12) into $I$, it follows that

$$I \leq \sum_{i=1}^{\infty}\sum_{j=0}^{\infty} \mathbb{P}\Big(\sup_{f\in\mathcal{H}_{ij}} \frac{1}{n}\sum_{i=1}^{n} l_\phi(f_\mathcal{F}^*, z_i) - \frac{1}{n}\sum_{i=1}^{n} l_\phi(f, z_i) \geq M(i,j)\Big) = \sum_{i=1}^{\infty}\sum_{j=0}^{\infty} P_{ij}.$$

Therefore, bounding $I$ reduces to bounding $P_{ij}$ separately. Let $Q_n = \frac{1}{n}\sum_{i=1}^{n}\big(l_\phi(f_\mathcal{F}^*, z_i) - l_\phi(f, z_i)\big)$, then $P_{ij}$ can be written as

$$P_{ij} = \mathbb{P}\left(\sup_{f\in\mathcal{H}_{ij}} Q_n - \mathbb{E}\big[\sup_{f\in\mathcal{H}_{ij}} Q_n\big] \geq M(i,j) - \mathbb{E}\big[\sup_{f\in\mathcal{H}_{ij}} Q_n\big]\right).$$

Next, we proceed to bound $P_{ij}$ by the Talagrand's inequality (see Theorem 2.6 in Koltchinskii, 2011). To this end, we first establish the relation between $\mathbb{E}\big[\sup_{f\in\mathcal{H}_{ij}} Q_n\big]$ and $M(i,j)$. Let $q(f, z_i) = \phi(f_\mathcal{F}^*(\boldsymbol{x}_i)y_i) - \phi(f(\boldsymbol{x}_i)y_i)$ and $\boldsymbol{z}' = (z_1', \ldots, z_n')$ be a ghost sample.

$$\mathbb{E}\left[\sup_{f\in\mathcal{H}_{ij}} Q_n\right] = \mathbb{E}\left[\sup_{f\in\mathcal{H}_{ij}} \frac{1}{n}\sum_{i=1}^{n} q(f, z_i) - \mathbb{E}(q(f, Z))\right]$$

$$= \mathbb{E}_{\boldsymbol{z}}\left[\sup_{f\in\mathcal{H}_{ij}} \mathbb{E}_{\boldsymbol{z}'}\Big(\frac{1}{n}\sum_{i=1}^{n} q(f, z_i) - \frac{1}{n}\sum_{i=1}^{n} q(f, z_i')|\boldsymbol{z}\Big)\right]$$

$$\leq \mathbb{E}_{\boldsymbol{z}, \boldsymbol{z}'}\left[\sup_{f\in\mathcal{H}_{ij}} \Big(\frac{1}{n}\sum_{i=1}^{n} q(f, z_i) - \frac{1}{n}\sum_{i=1}^{n} q(f, z_i')\Big)\right]$$

$$= \mathbb{E}_{\boldsymbol{z}, \boldsymbol{z}', \boldsymbol{\sigma}}\left[\sup_{f\in\mathcal{H}_{ij}} \Big(\frac{1}{n}\sum_{i=1}^{n} \sigma_i(q(f, z_i) - q(f, z_i'))\Big)\right]$$

$$\leq 2\mathbb{E}_{\boldsymbol{z}}\mathbb{E}_{\boldsymbol{\sigma}}\left[\sup_{f\in\mathcal{H}_{ij}} \Big(\frac{1}{n}\sum_{i=1}^{n} \sigma_i q(f, z_i)\Big)\right] = 2\mathcal{R}_n(\mathcal{H}_{ij}).$$

By Theorem 3.11 in Koltchinskii (2011), it follows that there exists some constant $C_2$ such that

$$\mathcal{R}_n(\mathcal{H}_{ij}) \leq \frac{C_2}{\sqrt{n}}\mathbb{E}\int_0^{\sigma_{ij}} \sqrt{\log\mathcal{N}(u, \mathcal{H}_{ij}, L_2(P_n))}du \leq \frac{C_2}{\sqrt{n}}\mathbb{E}\int_0^{\sigma_{ij}} \sqrt{\mathcal{V}_1(\Theta)\log(u^{-1}\mathcal{V}_2(\Theta))}du,$$

where $\sigma_{ij} = \sqrt{\sup_{f\in\mathcal{H}_{ij}} \frac{1}{n}\sum_{i=1}^{n} q^2(f, z_i)}$, $\|f\|_{L_2(P_n)} = \sqrt{\frac{1}{n}\sum_{i=1}^{n} q^2(f, z_i)}$, and $\mathcal{N}(\mathcal{H}_{ij}, L_2(P_n), u)$ is the minimal number of $L_2(P_n)$-balls of radius $u$ to cover $\mathcal{H}_{ij}$.

Notice that $\int_0^{\sigma_{ij}} \sqrt{\mathcal{V}_1(\Theta)\log(u^{-1}\mathcal{V}_2(\Theta))}du$ is concave function with respect to $\sigma_{ij}$, therefore

$$\mathbb{E}\left\{\int_0^{\sigma_{ij}} \sqrt{\mathcal{V}_1(\Theta)\log(u^{-1}\mathcal{V}_2(\Theta))}du\right\} \leq \int_0^{\mathbb{E}\sigma_{ij}} \sqrt{\mathcal{V}_1(\Theta)\log(u^{-1}\mathcal{V}_2(\Theta))}du \leq \int_0^{\sqrt{\mathbb{E}\sigma_{ij}^2}} \sqrt{\mathcal{V}_1(\Theta)\log(u^{-1}\mathcal{V}_2(\Theta))}du.$$

By the definition of $\sigma_{ij}$, we have $\mathbb{E}\big[\sigma_{ij}^2\big] = \mathbb{E}\big[\sup_{f\in\mathcal{H}_{ij}} \frac{1}{n}\sum_{i=1}^{n} q^2(f, z_i)\big]$. Using symmetrization and contraction inequalities, we get

$$\mathbb{E}\big[\sigma_{ij}^2\big] \leq \sup_{f\in\mathcal{H}_{ij}} \mathbb{E}[q^2(f, Z)] + 8\mathcal{R}_n(f, \mathcal{H}_{ij}).$$

Next, we proceed to bound $\mathbb{E}[q^2(f, Z)]$.

$$
\begin{aligned}
\mathbb{E}[q^2(f, Z)] &= \mathbb{E}\left[\phi(f_{\mathcal{F}}^*(\boldsymbol{X})\widetilde{Y}) - \phi(f(\boldsymbol{X})\widetilde{Y})\right]^2 \\
&\leq 2\mathbb{E}\left[\phi(f_{\mathcal{F}}^*(\boldsymbol{X})\widetilde{Y}) - \phi(f_\phi^*(\boldsymbol{X})\widetilde{Y})\right]^2 + 2\mathbb{E}\left[\phi(f_{\mathcal{F}}^*(\boldsymbol{X})\widetilde{Y}) - \phi(f(\boldsymbol{X})\widetilde{Y})\right]^2 \\
&\leq 2\mathbb{E}\left[\phi(f_{\mathcal{F}}^*(\boldsymbol{X})\widetilde{Y}) - \phi(f^*(\boldsymbol{X})\widetilde{Y})\right] + 2\mathbb{E}\left[\left|\phi(f(\boldsymbol{X})\widetilde{Y}) - \phi(f_{\mathcal{F}}^*(\boldsymbol{X})\widetilde{Y})\right|\right] \\
&\leq 2s_n + 2\mathbb{E}\left[\left|f(\boldsymbol{X}) - f_{\mathcal{F}}^*(\boldsymbol{X})\right|\right] \leq 2^{-1}\delta_n + 2\mathbb{E}\left[\left|f(\boldsymbol{X}) - f_{\mathcal{F}}^*(\boldsymbol{X})\right|\right],
\end{aligned}
\tag{13}
$$

where the second inequality follows from the assumption that $\|f\|_\infty \leq 1$. Combining this with Lemma 3 yields that

$$
\sup_{f \in \mathcal{H}_{ij}} \mathbb{E}[q^2(f, Z)] \leq 2^{-1}\delta_n + 2C_1\left((2\theta-1)^{-1}(\widetilde{R}_\phi(f) - \widetilde{R}_\phi(f^*))\right)^{\frac{\gamma}{\gamma+1}} = 4C_1(2\theta-1)^{-\frac{\gamma}{\gamma+1}}(2^i\delta_n)^{\frac{\gamma}{\gamma+1}},
$$

where $C_1 = (4c)^{\frac{1}{\gamma+1}}$. Consequently, we get

$$
\mathcal{R}_n(\mathcal{H}_{ij}) \leq \frac{C_2}{\sqrt{n}} \int_0^{U_{ij}(f)} \sqrt{\mathcal{V}_1(\Theta) \log(u^{-1}\mathcal{V}_2(\Theta))} du,
\tag{14}
$$

where $U_{ij}(f) = \min\left\{\sqrt{4C_1(2\theta-1)^{-\frac{\gamma}{\gamma+1}}(2^i\delta_n)^{\frac{\gamma}{\gamma+1}} + 8\mathcal{R}_n(\mathcal{H}_{ij})}, 1\right\}$ due to the fact that $|q(f, Z)| \leq 1$. Then, the right-hand side of (14) can be upper bounded as

$$
\begin{aligned}
\frac{C_2}{\sqrt{n}} \int_0^{U_{ij}(f)} \sqrt{\mathcal{V}_1(\Theta) \log(u^{-1}\mathcal{V}_2(\Theta))} du &= \frac{C_2 \mathcal{V}_2(\Theta) \sqrt{\mathcal{V}_1(\Theta)}}{\sqrt{n}} \int_0^{U_{ij}(f)/\mathcal{V}_2(\Theta)} \sqrt{\log\left(\frac{1}{u}\right)} du \\
&= \frac{C_2 \mathcal{V}_2(\Theta) \sqrt{\mathcal{V}_1(\Theta)}}{\sqrt{n}} \int_{\frac{\mathcal{V}_2(\Theta)}{U_{ij}(f)}}^{+\infty} \frac{1}{u^2} \sqrt{\log(u)} du \leq \frac{2C_2 \sqrt{\mathcal{V}_1(\Theta)} U_{ij}(f)}{\sqrt{n}} \sqrt{\log\left(\frac{\mathcal{V}_2(\Theta)}{U_{ij}(f)}\right)}.
\end{aligned}
\tag{15}
$$

Combining (15) with (14) yields that

$$
\left(\mathcal{R}_n(\mathcal{H}_{ij})\right)^2 \leq \frac{4C_2^2 \mathcal{V}_1(\Theta)\left(4C_1(2\theta-1)^{-\frac{\gamma}{\gamma+1}}(2^i\delta_n)^{\frac{\gamma}{\gamma+1}} + 8\mathcal{R}_n(\mathcal{H}_{ij})\right)}{n} \log\left(\frac{\mathcal{V}_2(\Theta)}{U_{ij}(f)}\right).
\tag{16}
$$

Solving (16) gives $\mathcal{R}_n(f, \mathcal{H}_{ij}) \lesssim \max\{\mathcal{V}_1(\Theta)n^{-1}, n^{-1/2}\mathcal{V}_1(\Theta)^{1/2}(2\theta-1)^{-\frac{\gamma}{2(\gamma+1)}}\delta_n^{\frac{\gamma}{2(\gamma+1)}}\}$. Therefore, we get $\mathcal{R}_n(f, \mathcal{H}_{ij}) \leq 1/4\delta_n$ provided that

$$
\left(\mathcal{V}_1(\Theta)/n\right)^{\frac{\gamma+1}{\gamma+2}}(2\theta-1)^{-\frac{\gamma}{\gamma+2}} \log(n/\mathcal{V}_1(\Theta)) \leq C\delta_n,
\tag{17}
$$

for some large constants $C$. Hence, as $n$ goes to infinity, it follows that

$$
\mathcal{R}_n(f, \mathcal{H}_{ij}) \leq 1/4\delta_n \leq 1/4M(i, j).
$$

With this, we get

$$
\mathbb{E}\left[\sup_{f \in \mathcal{H}_{ij}} Q_n\right] \leq \mathcal{R}_n(f, \mathcal{H}_{ij}) \leq 1/4M(i, j).
$$

Therefore, $P_{ij}$ can be further bounded as

$$
P_{ij} \leq \mathbb{P}\left(\sup_{f \in \mathcal{H}_{ij}} Q_n - \mathbb{E}\left[\sup_{f \in \mathcal{H}_{ij}} Q_n\right] \geq 1/2M(i, j)\right).
\tag{18}
$$

Applying Talagrand's inequality to the right-hand side, it follows that there exists some positive constants $C_4$ such that

$$P_{ij} \leq C_4 \exp\left(-\frac{nM(i,j)}{2C_4} \log\left(1 + \frac{M(i,j)}{2\mathbb{E}[\sigma_{ij}^2]}\right)\right).$$

As proved above, we can verify that there exists some constant $C_5$

$$\mathbb{E}[\sigma_{ij}^2] \leq \sup_{f \in \mathcal{H}_{ij}} \mathbb{E}[q^2(f,Z)] + 8\mathcal{R}_n(f,\mathcal{H}_{ij}) \leq C_5\left((2\theta-1)^{-\frac{\gamma}{\gamma+1}}(2^i\delta_n)^{\frac{\gamma}{\gamma+1}} + M(i,j)\right).$$

Notice that $\frac{M(i,j)}{2\mathbb{E}[\sigma_{ij}^2]}$ converges to 0 as $n$ increases, therefore there exists some constant $0 < C_7 < 1$ such that $\log(1+x) \geq C_7 x$ for $x \in [0, 1/(2C_5)]$. It then follows that

$$P_{ij} \leq C_4 \exp\left(-\frac{C_7 n M^2(i,j)}{4C_4 C_5\left((2\theta-1)^{-\frac{\gamma}{\gamma+1}}(M(i,j))^{\frac{\gamma}{\gamma+1}} + M(i,j)\right)}\right).$$

Since $M(i,j) \ll (2\theta-1)^{-\frac{\gamma}{\gamma+1}}(M(i,j))^{\frac{\gamma}{\gamma+1}}$ when $\theta - 1/2 = o(1)$ and $M(i,j) = o(1)$, we have

$$P_{ij} \leq C_4 \exp\left(-C_8 n(2\theta-1)^{\frac{\gamma}{\gamma+1}} M^{\frac{\gamma+2}{\gamma+1}}(i,j)\right),$$

where $C_8 = C_7/(4C_4 C_5)$. Therefore, we have

$$\sum_{i=1}^{n}\sum_{j=0}^{n} P_{ij} \leq \sum_{i=1}^{n}\sum_{j=1}^{n} C_4 \exp\left(-\frac{C_8 n M^{\frac{\gamma+2}{\gamma+1}}(i,j)}{(2\theta-1)^{-\frac{\gamma}{\gamma+1}}}\right) + \sum_{i=1}^{n} C_4 \exp\left(-\frac{C_8 n M^{\frac{\gamma+2}{\gamma+1}}(i,0)}{(2\theta-1)^{-\frac{\gamma}{\gamma+1}}}\right) \equiv I_1 + I_2.$$

Note that for any $i,j \geq 1$,

$$M^{\frac{\gamma+2}{\gamma+1}}(i,j) \geq \left((2^{i-1} - 1/4)\delta_n + \lambda_n(2^{j-1}-1)J_0\right)^{\frac{\gamma+2}{\gamma+1}}$$
$$\geq (2^{i-1} - 1/4)\delta_n^{\frac{\gamma+2}{\gamma+1}} + (2^{j-1}-1)(\lambda_n J_0)^{\frac{\gamma+2}{\gamma+1}}$$
$$\geq i/2\delta_n^{\frac{\gamma+2}{\gamma+1}} + (j-1)(\lambda_n J_0)^{\frac{\gamma+2}{\gamma+1}}.$$

Hence $I_1$ is upper bounded as

$$I_1 \leq \sum_{i=1}^{n}\sum_{j=1}^{n} C_4 \exp\left(-C_8 n \frac{i/2\delta_n^{\frac{\gamma+2}{\gamma+1}} + (j-1)(\lambda_n J_0)^{\frac{\gamma+2}{\gamma+1}}}{(2\theta-1)^{-\frac{\gamma}{\gamma+1}}}\right)$$
$$= C_4 \frac{\exp\left(\frac{-C_8 n\delta_n^{\frac{\gamma+2}{\gamma+1}}}{2(2\theta-1)^{-\frac{\gamma}{\gamma+1}}}\right)}{1 - \exp\left(\frac{-C_8 n\delta_n^{\frac{\gamma+2}{\gamma+1}}}{2(2\theta-1)^{-\frac{\gamma}{\gamma+1}}}\right)} \cdot \frac{1}{1 - \exp\left(\frac{-C_8 n(\lambda_n J_0)^{\frac{\gamma+2}{\gamma+1}}}{2(2\theta-1)^{-\frac{\gamma}{\gamma+1}}}\right)} \leq 4C_4 \exp\left(\frac{-C_8 n\delta_n^{\frac{\gamma+2}{\gamma+1}}}{2(2\theta-1)^{-\frac{\gamma}{\gamma+1}}}\right),$$

where the last inequality holds when $\max\left\{\exp\left(\frac{-C_8 n\delta_n^{\frac{\gamma+2}{\gamma+1}}}{2(2\theta-1)^{-\frac{\gamma}{\gamma+1}}}\right), \exp\left(\frac{-C_8 n\delta_n^{2-1/\gamma}}{2(2\theta-1)^{-\frac{\gamma}{\gamma+1}}}\right)\right\} \leq 1/2$, which holds true when $n$ goes to infinity. Similarly, for $I_2$, we get

$$I_2 \leq \sum_{i=1}^{n} C_4 \exp\left(-\frac{i/4 C_8 n\delta_n^{\frac{\gamma+2}{\gamma+1}}}{(2\theta-1)^{-\frac{\gamma}{\gamma+1}}}\right) \leq 2C_4 \exp\left(-\frac{C_8 n\delta_n^{\frac{\gamma+2}{\gamma+1}}}{4(2\theta-1)^{-\frac{\gamma}{\gamma+1}}}\right).$$

Combining $I_1$ and $I_2$, it follows that

$$I \leq 6C_4 \exp\left(-\frac{C_8 n \delta_n^{\frac{\gamma+2}{\gamma+1}}}{4(2\theta-1)^{-\frac{\gamma}{\gamma+1}}}\right) = T_1 \exp\left(-T_2 \frac{n\delta_n^{\frac{\gamma+2}{\gamma+1}}}{(2\theta-1)^{-\frac{\gamma}{\gamma+1}}}\right),$$

where $T_1 = 6C_4$ and $T_2 = C_8/4$. Therefore, we conclude that

$$\mathbb{P}\left(\widetilde{H}(\widetilde{f}_n, f^*) \geq \delta_n\right) \leq T_1 \exp\left(-T_2 \frac{n\delta_n^{\frac{\gamma+2}{\gamma+1}}}{(2\theta-1)^{-\frac{\gamma}{\gamma+1}}}\right).$$

With a choice of $\delta_n$ such that $\delta_n \geq C_9\big((2\theta-1)^{-\frac{\gamma}{\gamma+1}}|\Theta|n^{-1}\log(n/|\Theta|)\big)^{\frac{\gamma+1}{\gamma+2}}$ for some positive constants $C_9 > 0$, we have $\widetilde{H}(\widetilde{f}_n, f^*) = o_p(1)$, which implies that $\mathbb{E}\big(\widetilde{H}(\widetilde{f}_n, f^*)\big) = O_p(\delta_n)$. Notice that $T_1$ and $T_2$ are both independent of $\pi$, therefore we further have

$$\sup_{\pi \in \mathcal{P}_\gamma} \mathbb{P}\left(\widetilde{H}(\widetilde{f}_n, f^*) \geq \delta_n\right) \leq T_1 \exp\left(-T_2 \frac{n\delta_n^{\frac{\gamma+2}{\gamma+1}}}{(2\theta-1)^{-\frac{\gamma}{\gamma+1}}}\right). \tag{19}$$

By the relation between excess risk and excess $\phi$-risk $\mathbb{E}_{\widetilde{\mathcal{D}}}\big(\widetilde{D}(\widetilde{f}_n, f^*)\big) \leq \mathbb{E}_{\widetilde{\mathcal{D}}}\big(\widetilde{H}(\widetilde{f}_n, f^*)\big)$ (Bartlett et al., 2006), we further have

$$\sup_{\pi \in \mathcal{P}_\gamma} \mathbb{E}_{\widetilde{\mathcal{D}}}\left(\widetilde{D}(\widetilde{f}_n, f^*)\right) = O(\delta_n).$$

Combining this with the Lemma 2 that $D(\widetilde{f}_n, f^*) = (2\theta-1)^{-1}\widetilde{D}(\widetilde{f}_n, f^*)$, we get

$$\sup_{\pi \in \mathcal{P}_\gamma} \mathbb{E}_{\widetilde{\mathcal{D}}}\left(D(\widetilde{f}_n, f^*)\right) = O(\delta_n(2\theta-1)^{-1}).$$

The fastest rate for $\delta_n$ can be obtained by choosing the fastest rate such that the right-hand side of (19) converges to zero with $n$ and (17) holds, which yields that

$$\delta \asymp \left(\mathcal{V}_1(\Theta)/n\right)^{\frac{\gamma+1}{\gamma+2}}(2\theta-1)^{-\frac{\gamma}{\gamma+2}}\log(n).$$

This completes the proof of the upper bound.

**Proof of lower bound.** We first define the minimax excess risk as

$$W_n = \inf_f \sup_{\pi \in \mathcal{P}_\gamma} \mathbb{E}_{\widetilde{\mathcal{D}}}\big[R(f) - R(f^*)\big],$$

which admits the decomposition as

$$W_n = \inf_f \sup_{\pi \in \mathcal{P}_\gamma} \left\{\mathbb{E}_{\widetilde{\mathcal{D}}}\big[R(f) - \inf_{f \in \mathcal{F}} R(f)\big] + \inf_{f \in \mathcal{F}} R(f) - R(f^*)\right\},$$

where the second term on the right-hand side denotes the approximation error under the 0-1 risk. In what follows, we proceed to consider a sub-family of $\mathcal{P}_\gamma$ such that $\inf_{f \in \mathcal{F}} R(f) = R(f^*)$. For example, let $\mathcal{P}' \subset \mathcal{P}_\gamma$ be such a sub-family, then we have

$$\begin{aligned}
W_n &\geq \inf_f \sup_{\pi \in \mathcal{P}'} \left\{\mathbb{E}_{\widetilde{\mathcal{D}}}\big[R(f) - \inf_{f \in \mathcal{F}} R(f)\big]\right\} + \sup_{\pi \in \mathcal{P}_\gamma} \left(\inf_{f \in \mathcal{F}} R(f) - R(f^*)\right) \\
&= \inf_f \sup_{\pi \in \mathcal{P}'} \left\{\mathbb{E}_{\widetilde{\mathcal{D}}}\big[R(f) - R(f^*)\big]\right\} + \sup_{\pi \in \mathcal{P}_\gamma} \left(\inf_{f \in \mathcal{F}} R(f) - R(f^*)\right) \\
&= (2\theta-1)^{-1} \inf_f \sup_{\widetilde{\pi} \in \widetilde{\mathcal{P}}'(\theta)} \left\{\mathbb{E}_{\widetilde{\mathcal{D}}}\big[\widetilde{R}(f) - \widetilde{R}(f^*)\big]\right\} + \sup_{\pi \in \mathcal{P}_\gamma} \left(\inf_{f \in \mathcal{F}} R(f) - R(f^*)\right),
\end{aligned}$$

where the last inequality follows from Lemma 2 and $\widetilde{\mathcal{P}}'(\theta)$ is a set of probability measures $\widetilde{\pi}$ on $(\boldsymbol{X}, \widetilde{Y})$ such that any probability distribution $\widetilde{\pi} \in \widetilde{\mathcal{P}}'(\theta)$ is associated with an $\pi \in \mathcal{P}'$ with $\widetilde{\pi}$ and $\pi$ having the same marginal distribution on $\boldsymbol{X}$ and $\widetilde{Y} = \mathcal{A}_\theta(Y)$.

To provide a lower bound for $W_n$, it suffices to bound $\inf_f \sup_{\widetilde{\pi} \in \widetilde{\mathcal{P}}'(\theta)} \left\{ \mathbb{E}_{\widetilde{\mathcal{D}}} \left[ \widetilde{R}(f) - \widetilde{R}(f^*) \right] \right\}$. For ease of notation, we denote $\widetilde{W_n} = \inf_f \sup_{\widetilde{\pi} \in \widetilde{\mathcal{P}}'(\theta)} \mathbb{E}_{\widetilde{\mathcal{D}}} \left[ \widetilde{R}(f) - \widetilde{R}(f^*) \right]$ and $\widetilde{\mathcal{P}}_\gamma(\theta) = \{ \widetilde{\pi}(\boldsymbol{X}, \widetilde{Y}) : \widetilde{Y} = \mathcal{A}_\theta(Y), \pi \in \mathcal{P}_\gamma \}$. Then we proceed to construct $\widetilde{\mathcal{P}}'(\theta) \subset \widetilde{\mathcal{P}}_\gamma(\theta)$. First, following from Lemma 3, we have for any $\widetilde{\pi} \in \widetilde{\mathcal{P}}_\gamma(\theta)$

$$\mathbb{P}\left( |2\widetilde{\eta}(\boldsymbol{X}) - 1| \leq t \right) \leq c(2\theta - 1)^{-\gamma} t^\gamma,$$

where the probability is measured under $\widetilde{\pi}$.

By Assumption 2, we know $VC(\mathcal{G}_\mathcal{F}) \asymp \mathcal{V}_1(\Theta)$. For any $N$ such that $N \asymp \mathcal{V}_1(\Theta)$, there exist $N$ distinct points $\boldsymbol{x}_1, \ldots, \boldsymbol{x}_N$ such that $\{\boldsymbol{x}_1, \ldots, \boldsymbol{x}_N\}$ is shattered by $\mathcal{G}_\mathcal{F}$. Therefore, we consider distribution supported on $\{\boldsymbol{x}_1, \ldots, \boldsymbol{x}_N\}$. Let $w \in (0,1)$ be a number satisfying $(N-1)w \leq 1$. Let $Q$ be the probability measure on $\mathcal{X}$ such that

$$Q(\boldsymbol{x}_i) = \begin{cases} w, & i = 1, \ldots, N-1, \\ 1-(N-1)w, & i = N. \end{cases}$$

In what follows, we consider the hypercube $\mathcal{C} = \{-1, 1\}^{N-1}$. For any $\boldsymbol{\sigma} = (\sigma_1, \ldots, \sigma_{N-1}) \in \mathcal{C}$, we define the regression function as

$$\widetilde{\eta}_{\boldsymbol{\sigma}}(\boldsymbol{x}_i) = \begin{cases} \frac{1+\sigma_j h}{2}, & i = 1, \ldots, N-1, \\ 1, & i = N. \end{cases}$$

Next, we let $\widetilde{\pi}_{\boldsymbol{\sigma}}$ denote the associated probability measure on $\mathcal{X} \times \{-1, 1\}$ with $Q$ being marginal distribution on $\mathcal{X}$ and $\widetilde{\eta}_{\boldsymbol{\sigma}}(\boldsymbol{x})$ being the regression function. With this, we obtain

$$\mathbb{P}\left( |2\widetilde{\eta}_{\boldsymbol{\sigma}}(\boldsymbol{X}) - 1| \leq t \right) = (N-1)w I(h \leq t).$$

By assuming $(N-1)w \leq c(2\theta-1)^{-\gamma} h^\gamma$, we have $\mathbb{P}\left( |2\eta_{\boldsymbol{\sigma}}(\boldsymbol{X}) - 1| \leq t \right) = c(2\theta-1)^{-\gamma} t^\gamma$ for any $t \in [0,1)$, which indicates that $\widetilde{\pi}_{\boldsymbol{\sigma}} \in \widetilde{\mathcal{P}}_\gamma(\theta)$. By setting $\widetilde{\mathcal{P}}'(\theta) = \{\widetilde{\pi}_{\boldsymbol{\sigma}} : \boldsymbol{\sigma} \in \mathcal{C}\}$, we have

$$\widetilde{W_n} = \inf_f \sup_{\widetilde{\pi}_{\boldsymbol{\sigma}} \in \widetilde{\mathcal{P}}'(\theta)} \mathbb{E}\left[ \widetilde{R}(f) - \widetilde{R}(f^*_{\boldsymbol{\sigma}}) \right],$$

where $f^*_{\boldsymbol{\sigma}}(\boldsymbol{x}_i) = \sigma_i$ for $i = 1, \ldots, N-1$.

Under $\widetilde{\pi}_{\boldsymbol{\sigma}}$,

$$\begin{aligned}
\widetilde{R}(f) - \widetilde{R}(f^*_{\boldsymbol{\sigma}}) &= \mathbb{E}_{\widetilde{\pi}_{\boldsymbol{\sigma}}}\left[ |\operatorname{sign}(f(\boldsymbol{X})) \neq f^*_{\boldsymbol{\sigma}}(\boldsymbol{X})| |1 - 2\widetilde{\eta}(\boldsymbol{X})| \right] \\
&= \frac{1}{2}\mathbb{E}_{\widetilde{\pi}_{\boldsymbol{\sigma}}}\left[ |\operatorname{sign}(f(\boldsymbol{X})) - f^*_{\boldsymbol{\sigma}}(\boldsymbol{X})| |1 - 2\widetilde{\eta}(\boldsymbol{X})| \right] \\
&\geq \frac{t}{2}\mathbb{E}_{\widetilde{\pi}_{\boldsymbol{\sigma}}}\left[ |\operatorname{sign}(f(\boldsymbol{X})) - f^*_{\boldsymbol{\sigma}}(\boldsymbol{X})| \left(1 - \mathbb{P}(|1 - 2\widetilde{\eta}(\boldsymbol{X})| \leq t)\right) \right] \\
&\geq (2\theta-1)(4^{1-\gamma}c)^{-1/\gamma}\left( \mathbb{E}_{\widetilde{\pi}_{\boldsymbol{\sigma}}}\left[ |\operatorname{sign}(f(\boldsymbol{X})) - f^*_{\boldsymbol{\sigma}}(\boldsymbol{X})| \right] \right)^{\frac{\gamma+1}{\gamma}} \\
&\geq (2\theta-1)(4^{1-\gamma}c)^{-1/\gamma}\left( w \sum_{i=1}^{N-1} |\operatorname{sign}(f(\boldsymbol{x}_i)) - f^*_{\boldsymbol{\sigma}}(\boldsymbol{x}_i)| \right)^{\frac{\gamma+1}{\gamma}}.
\end{aligned}$$

Under the distribution $\widetilde{\pi}_{\boldsymbol{\sigma}}$, we have $f^*(\boldsymbol{x}_i) = \sigma_i$ for $i = 1, \ldots, N-1$, which then implies that

$$\widetilde{R}(f) - \widetilde{R}(f^*_{\boldsymbol{\sigma}}) \geq (2\theta-1)(4^{1-\gamma}c)^{-1/\gamma} w^{\frac{\gamma+1}{\gamma}} \left( \sum_{i=1}^{N-1} |\operatorname{sign}(f(\boldsymbol{x}_i)) - \sigma_i| \right)^{\frac{\gamma+1}{\gamma}}.$$

Taking the expectation of both sides with respect to $\widetilde{\mathcal{D}}$ yields that

$$\mathbb{E}_{\widetilde{\mathcal{D}}}\Big(\widetilde{R}(f) - \widetilde{R}(f_{\boldsymbol{\sigma}}^*)\Big) \geq (2\theta - 1)(4^{1-\gamma}c)^{-1/\gamma}w^{\frac{\gamma+1}{\gamma}}\mathbb{E}_{\widetilde{\mathcal{D}}}\Big[\Big(\sum_{i=1}^{N-1}|\operatorname{sign}(f(\boldsymbol{x}_i)) - \sigma_i|\Big)^{\frac{\gamma+1}{\gamma}}\Big]$$

$$\geq (2\theta - 1)(4^{1-\gamma}c)^{-1/\gamma}w^{\frac{\gamma+1}{\gamma}}\mathbb{E}_{\widetilde{\mathcal{D}}}^{\frac{\gamma+1}{\gamma}}\Big[\Big(\sum_{i=1}^{N-1}|\operatorname{sign}(f(\boldsymbol{x}_i)) - \sigma_i|\Big)\Big],$$

where the second inequality follows from Jensen's inequality.

For ease of notation, we let $X(w, \gamma, \theta) = (2\theta - 1)(4^{1-\gamma}c)^{-1/\gamma}w^{\frac{\gamma+1}{\gamma}}$.

$$\sup_{\widetilde{\pi}_{\boldsymbol{\sigma}} \in \widetilde{\mathcal{P}}'(\theta)} \mathbb{E}_{\widetilde{\mathcal{D}}}\Big(\widetilde{R}(f) - \widetilde{R}(f_{\boldsymbol{\sigma}}^*)\Big) \geq \sup_{\widetilde{\pi}_{\boldsymbol{\sigma}} \in \widetilde{\mathcal{P}}'(\theta)} X(w, \gamma, \theta)\mathbb{E}_{\widetilde{\mathcal{D}}}^{\frac{\gamma+1}{\gamma}}\Big[\Big(\sum_{i=1}^{N-1}|\operatorname{sign}(f(\boldsymbol{x}_i)) - \sigma_i|\Big)\Big],$$

$$\geq \frac{1}{2^N}\sum_{\boldsymbol{\sigma} \in \mathcal{C}} X(w, \gamma, \theta)\mathbb{E}_{\widetilde{\mathcal{D}}}^{\frac{\gamma+1}{\gamma}}\Big[\Big(\sum_{i=1}^{N-1}|\operatorname{sign}(f(\boldsymbol{x}_i)) - \sigma_i|\Big)\Big]$$

$$= \frac{1}{2^{N-1}}\sum_{\boldsymbol{\sigma} \in \mathcal{C}} X(w, \gamma, \theta)\mathbb{E}_{\widetilde{\mathcal{D}}}^{\frac{\gamma+1}{\gamma}}\Big[\sum_{i=1}^{N-1} I\big(\operatorname{sign}(f(\boldsymbol{x}_i)) \neq \sigma_i\big)\Big]$$

$$= X(w, \gamma, \theta)\Big(\frac{1}{2^{N-1}}\sum_{\boldsymbol{\sigma} \in \mathcal{C}}\sum_{i=1}^{N-1}\mathbb{E}_{\widetilde{\mathcal{D}}}\Big[I\big(\operatorname{sign}(f(\boldsymbol{x}_i)) \neq \sigma_i\big)\Big]\Big)^{\frac{\gamma+1}{\gamma}}$$

$$= X(w, \gamma, \theta)\Big(\sum_{i=1}^{N-1}\frac{1}{2^{N-1}}\sum_{\boldsymbol{\sigma} \in \mathcal{C}}\mathbb{P}_{\widetilde{\pi}_{\boldsymbol{\sigma}}}\big(\operatorname{sign}(f(\boldsymbol{x}_i)) \neq \sigma_i\big)\Big)^{\frac{\gamma+1}{\gamma}}.$$

For each $i$, we observe that

$$\frac{1}{2^{N-1}}\sum_{\boldsymbol{\sigma} \in \mathcal{C}}\mathbb{P}_{\widetilde{\pi}_{\boldsymbol{\sigma}}}\Big(\operatorname{sign}(f(\boldsymbol{x}_i)) \neq \sigma_i\Big)$$

$$= \frac{1}{2^{N-1}}\sum_{\boldsymbol{\sigma}:\sigma_i=1}\mathbb{P}_{\widetilde{\pi}_{\boldsymbol{\sigma}}}\Big(\operatorname{sign}(f(\boldsymbol{x}_i)) \neq \sigma_i\Big) + \frac{1}{2^{N-1}}\sum_{\boldsymbol{\sigma}:\sigma_i=-1}\mathbb{P}_{\widetilde{\pi}_{\boldsymbol{\sigma}}}\Big(\operatorname{sign}(f(\boldsymbol{x}_i)) \neq \sigma_i\Big)$$

$$= 2\mathbb{P}_{+i}\Big(\operatorname{sign}(f(\boldsymbol{x}_i)) \neq 1\Big) + 2\mathbb{P}_{-i}\Big(\operatorname{sign}(f(\boldsymbol{x}_i)) \neq -1\Big) \geq 2 - 2\operatorname{TV}(\mathbb{P}_{+i}^{\otimes n}, \mathbb{P}_{-i}^{\otimes n}).$$

where $\mathbb{P}_{+i} = \frac{1}{2^{N-1}}\sum_{\boldsymbol{\sigma}:\sigma_i=1}\mathbb{P}_{\widetilde{\pi}_{\boldsymbol{\sigma}}}$, $\mathbb{P}_{-i} = \frac{1}{2^{N-1}}\sum_{\boldsymbol{\sigma}:\sigma_i=-1}\mathbb{P}_{\widetilde{\pi}_{\boldsymbol{\sigma}}}$, and $\operatorname{TV}(\mathbb{P}_{+i}^n, \mathbb{P}_{-i}^n)$ denotes the total variation between $\mathbb{P}_{+i}^{\otimes n}$ and $\mathbb{P}_{-i}^{\otimes n}$. Notice that for any probability measures $P$ and $Q$, we have

$$\operatorname{TV}(P, Q) = \frac{1}{2}\int|p(\boldsymbol{x}) - q(\boldsymbol{x})|d\boldsymbol{x} = \frac{1}{2}\int|\sqrt{p(\boldsymbol{x})} - \sqrt{q(\boldsymbol{x})}||\sqrt{p(\boldsymbol{x})} + \sqrt{q(\boldsymbol{x})}|d\boldsymbol{x}$$

$$\leq \frac{1}{2}\Big(\int\big(\sqrt{p(\boldsymbol{x})} - \sqrt{q(\boldsymbol{x})}\big)^2 d\boldsymbol{x}\Big)^{1/2}\Big(\int\big(\sqrt{p(\boldsymbol{x})} + \sqrt{q(\boldsymbol{x})}\big)^2 d\boldsymbol{x}\Big)^{1/2}$$

$$\leq \sqrt{H^2(P, Q)},$$

where $H^2(P, Q)$ denotes the Hellienger distance between $P$ and $Q$. Let $\boldsymbol{\sigma}$ and $\boldsymbol{\sigma}'$ be two indexes such that $\boldsymbol{\sigma}_l = \boldsymbol{\sigma}_l'$ for $l \neq i$ and $\boldsymbol{\sigma}_i = -\boldsymbol{\sigma}_i' = 1$, then we have

$$\operatorname{TV}(\mathbb{P}_{+i}^{\otimes n}, \mathbb{P}_{-i}^{\otimes n}) \leq \sum_{\boldsymbol{\sigma}, \boldsymbol{\sigma}'}\frac{1}{2^{N-2}}\operatorname{TV}\big(\mathbb{P}_{\widetilde{\pi}_{\boldsymbol{\sigma}}}^{\otimes n}, \mathbb{P}_{\widetilde{\pi}_{\boldsymbol{\sigma}'}}^{\otimes n}\big) \leq \max_{\boldsymbol{\sigma}, \boldsymbol{\sigma}'}\operatorname{TV}\big(\mathbb{P}_{\widetilde{\pi}_{\boldsymbol{\sigma}}}^{\otimes n}, \mathbb{P}_{\widetilde{\pi}_{\boldsymbol{\sigma}'}}^{\otimes n}\big) \leq \max_{\boldsymbol{\sigma}, \boldsymbol{\sigma}'}H\big(\mathbb{P}_{\widetilde{\pi}_{\boldsymbol{\sigma}}}^{\otimes n}, \mathbb{P}_{\widetilde{\pi}_{\boldsymbol{\sigma}'}}^{\otimes n}\big).$$

By the definition of $\widetilde{\pi}_{\boldsymbol{\sigma}}$ and $\widetilde{\pi}_{\boldsymbol{\sigma}'}$, we get

$$H^2(\mathbb{P}_{\widetilde{\pi}_{\boldsymbol{\sigma}}}, \mathbb{P}_{\widetilde{\pi}_{\boldsymbol{\sigma}'}}) = w\sum_{i=1}^{N-1}\Big(\sqrt{\eta_{\boldsymbol{\sigma}}(\boldsymbol{x}_i)} - \sqrt{\eta_{\boldsymbol{\sigma}'}(\boldsymbol{x}_i)}\Big)^2 + \Big(\sqrt{1 - \eta_{\boldsymbol{\sigma}}(\boldsymbol{x}_i)} - \sqrt{1 - \eta_{\boldsymbol{\sigma}'}(\boldsymbol{x}_i)}\Big)^2$$

$$= 2w(1 - \sqrt{1 - h^2}) \leq 2wh^2,$$

for any $h \in [0,1]$. By the fact that $H^2(P^{\otimes n}, Q^{\otimes n}) = 2 - 2(1 - 2^{-1}H^2(P,Q))^n$, we have

$$H^2(\mathbb{P}^{\otimes n}_{\pi_{\sigma}}, \mathbb{P}^{\otimes n}_{\pi_{\sigma'}}) = 2 - 2\big[1 - w(1 - \sqrt{1-h^2})\big]^n \leq 2 - 2\big[1 - nw(1 - \sqrt{1-h^2})\big]$$

$$= 2nw(1 - \sqrt{1-h^2}) \leq 1/16,$$

where the last inequality holds by choosing $w$ and $h$ such that $wh^2 \leq n^{-1}/32$, from which it follows that $\mathrm{TV}(\mathbb{P}^n_{+i}, \mathbb{P}^n_{-i}) \leq 1/4$. Notice that $w$ and $h$ are chosen to satisfy $(N-1)w \leq c(2\theta-1)^{-\gamma}h^\gamma$ and $wh^2 \leq n^{-1}/32$. For simplicity, we obtain the following solution by considering the case equalities hold.

$$h = \frac{(N-1)^{\frac{1}{\gamma+2}}(2\theta-1)^{\frac{\gamma}{\gamma+2}}}{(32cn)^{\frac{1}{\gamma+2}}} \quad \text{and} \quad w = \frac{c^{\frac{2}{\gamma+2}}(N-1)^{-\frac{2}{\gamma+2}}(2\theta-1)^{-\frac{2\gamma}{\gamma+2}}}{(32n)^{\frac{\gamma}{\gamma+2}}}.$$

Therefore, we can conclude that

$$\sup_{\pi_{\sigma} \in \widetilde{\mathcal{P}'}(\theta)} \mathbb{E}_{\widetilde{\mathcal{D}}}\Big(\widetilde{R}(f) - \widetilde{R}(f^*_\sigma)\Big) \geq (2\theta-1)(4^{1-\gamma})^{-1/\gamma}c^{-\frac{1}{\gamma+2}} \frac{(N-1)^{-\frac{2\gamma+2}{\gamma(\gamma+2)}}(2\theta-1)^{-\frac{2\gamma+2}{\gamma+2}}}{(32n)^{\frac{\gamma+1}{\gamma+2}}}\Big(\frac{3(N-1)}{2}\Big)^{\frac{\gamma+1}{\gamma}}$$

$$= (2\theta-1)(4^{1-\gamma})^{-1/\gamma}c^{-\frac{1}{\gamma+2}} \frac{(N-1)^{\frac{\gamma+1}{\gamma+2}}(2\theta-1)^{-\frac{2\gamma+2}{\gamma+2}}}{(32n)^{\frac{\gamma+1}{\gamma+2}}}\Big(\frac{3}{2}\Big)^{\frac{\gamma+1}{\gamma}}$$

$$= (2\theta-1)(4^{1-\gamma})^{-1/\gamma}c^{-\frac{1}{\gamma+2}}\Big(\frac{N-1}{32n(2\theta-1)^2}\Big)^{\frac{\gamma+1}{\gamma+2}}\Big(\frac{3}{2}\Big)^{\frac{\gamma+1}{\gamma}}.$$

Choosing $N \asymp \mathcal{V}_1(\Theta)$ yields that

$$\inf_{f} \sup_{\pi_{\sigma} \in \widetilde{\mathcal{P}'}(\theta)} \mathbb{E}_{\widetilde{\mathcal{D}}}\Big(\widetilde{R}(f) - \widetilde{R}(f^*)\Big) \geq (2\theta-1)A_2\Big(\frac{\mathcal{V}_1(\Theta)}{n(2\theta-1)^2}\Big)^{\frac{\gamma+1}{\gamma+2}},$$

where $A_2 \asymp (4^{1-\gamma})^{-1/\gamma}c^{-\frac{1}{\gamma+2}}(2^{-1})^{\frac{6\gamma+6}{\gamma+2}}(3/2)^{\frac{\gamma+1}{\gamma}}$.

Following from Lemma 2, it holds that

$$\inf_{f} \sup_{\pi \in \mathcal{P}_\gamma} \mathbb{E}_{\widetilde{\mathcal{D}}}\Big(R(f) - R(f^*)\Big) \geq A_2\Big(\frac{\mathcal{V}_1(\Theta)}{n(2\theta-1)^2}\Big)^{\frac{\gamma+1}{\gamma+2}} + \sup_{\pi \in \mathcal{P}_\gamma}\Big(\inf_{f \in \mathcal{F}} R(f) - R(f^*)\Big).$$

This completes the proof of the lower bound. $\qquad \square$

**Corollary 1.** *Suppose that $\mathcal{F}$ is chosen such that $s_n = 0$ and $\epsilon$ is adaptive to $n$ such that $\epsilon = o(1)$. There exists some constants $A_3 > 0$ such that $\sup_{\pi \in \mathcal{P}_\gamma} \mathbb{E}_{\widetilde{\mathcal{D}}}\big[R(\widetilde{f}_n) - R(f^*)\big] \geq A_3$ provided that $\epsilon \lesssim n^{-1/2}$.*

**Proof of Corollary 1**: By Theorem 1, we have

$$\sup_{\pi \in \mathcal{P}_\gamma} \mathbb{E}_{\widetilde{\mathcal{D}}}\big[R(\widetilde{f}_n) - R(f^*)\big] \geq A_2\Big\{\Big(\frac{\mathcal{V}_1(\Theta)}{n\kappa_\epsilon^2}\Big)^{\frac{\gamma+1}{\gamma+2}} + \tau_n\Big\}.$$

Since $s_n = 0$ implies $\tau_n = 0$, we further have

$$\sup_{\pi \in \mathcal{P}_\gamma} \mathbb{E}_{\widetilde{\mathcal{D}}}\big[R(\widetilde{f}_n) - R(f^*)\big] \geq A_2\Big\{\Big(\frac{\mathcal{V}_1(\Theta)}{n\kappa_\epsilon^2}\Big)^{\frac{\gamma+1}{\gamma+2}}\Big\}.$$

Without loss of generality, we assume that $\epsilon < 1$ since $\epsilon = o(1)$. Then, by the definition of $\kappa_\epsilon$, $\kappa_\epsilon \asymp \epsilon$ when $\epsilon \to 0$. Therefore, we have

$$\frac{\mathcal{V}_1(\Theta)}{n\kappa_\epsilon^2} = \frac{\mathcal{V}_1(\Theta)(\exp(\epsilon)+1)^2}{n(\exp(\epsilon)-1)^2} \geq \frac{\mathcal{V}_1(\Theta)}{ne^2\epsilon^2}, \tag{20}$$

where the last inequality follows from the fact that $e^x - 1 > ex$ for any $x \in (0,1]$. Further, if $\epsilon = o(1/\sqrt{n})$, we have $n\epsilon^2 = o(1)$. Therefore, it follows that $\mathcal{V}_1(\Theta)n^{-1}\kappa_\epsilon^{-2} \geq C\mathcal{V}_1(\Theta)e^{-2}$ for some constants $C$. The desired result immediately follows by setting $A_3 = (C\mathcal{V}_1(\Theta)e^{-2})^{\frac{\gamma+1}{\gamma+2}}$ and this completes the proof. $\qquad \square$

**Theorem 2** (Restatement of Theorem 2). *Let $\mathcal{P}_{\gamma,\beta}$ be a class of probability measures on $\mathcal{X} \times \{-1,1\}$ satisfying Assumption 1 and $\eta(\boldsymbol{X}) \in \mathcal{H}(\beta, [0,1]^p, M)$. For any minimizer $\widetilde{f}_{nn}$ in (4) with $L_n \asymp \log(\kappa_\epsilon n / \log(n))$, $N_n \asymp (\kappa_\epsilon n / \log(n))^{\frac{2p}{2\beta+p}}$, $B_n = 1$, and $P_n \asymp N_n \log(\kappa_\epsilon n / \log(n))$, we have*

$$\left(\frac{1}{n\kappa_\epsilon^2}\right)^{\frac{\beta(\gamma+1)}{\beta(\gamma+2)+p}} \lesssim \sup_{\pi \in \mathcal{P}_{\gamma,\beta}} \mathbb{E}_{\widetilde{\mathcal{D}}}\big[R(\widetilde{s}_{nn}) - R(f^*)\big] \lesssim \left(\frac{\log n}{n\kappa_\epsilon^2}\right)^{\frac{2\beta(\gamma+1)}{2\beta(\gamma+2)+p(\gamma+2)}}.$$

*Particularly, $\sup_{\pi \in \mathcal{P}_{\gamma,\beta}} \mathbb{E}_{\widetilde{\mathcal{D}}}\big[R(\widetilde{s}_{nn}) - R(f^*)\big] = o(1)$ given that $\epsilon \gtrsim n^{-1/2+\zeta}$ for any $\zeta > 0$.*

**Proof of Theorem 2**: **Proof of the upper bound.** Let $\widetilde{Z} = (\widetilde{Y} + 1)/2$. We intend to establish the connection between the excess risk of $\widetilde{f}_{nn}$ and $\|\widetilde{f}_{nn} - \eta\|_{L^2(\mathbb{P}_{\boldsymbol{X}})}^2$. Here the proof is mainly based on Lemma 5.2 in Audibert & Tsybakov (2007).

$$\begin{aligned}
\widetilde{R}(\widetilde{s}_{nn}) - \widetilde{R}(f^*) =& \mathbb{E}\Big[\big|2\widetilde{\eta}(\boldsymbol{X}) - 1\big| \cdot I\big(\widetilde{s}_{nn}(\boldsymbol{X}) \neq f^*(\boldsymbol{X})\big)\Big] \\
\leq& 2\mathbb{E}\Big[\big|\widetilde{\eta}(\boldsymbol{X}) - 1/2\big| \cdot I\big(\widetilde{s}_{nn}(\boldsymbol{X}) \neq f^*(\boldsymbol{X})\big) \cdot I\big(|\widetilde{\eta}(\boldsymbol{X}) - 1/2| \leq t\big)\Big] \\
&+ 2\mathbb{E}\Big[\big|\widetilde{\eta}(\boldsymbol{X}) - 1/2\big| \cdot I\big(\widetilde{s}_{nn}(\boldsymbol{X}) \neq f^*(\boldsymbol{X})\big) \cdot I\big(|\widetilde{\eta}(\boldsymbol{X}) - 1/2| > t\big)\Big] \\
\leq& 2\mathbb{E}\Big[\big|\widetilde{\eta}(\boldsymbol{X}) - \widetilde{f}_{nn}(\boldsymbol{X})\big| \cdot I\big(|\widetilde{\eta}(\boldsymbol{X}) - 1/2| \leq t\big)\Big] \\
&+ 2\mathbb{E}\Big[\big|\widetilde{\eta}(\boldsymbol{X}) - \widetilde{f}_{nn}(\boldsymbol{X})\big| \cdot I\big(|\widetilde{\eta}(\boldsymbol{X}) - \widetilde{f}_{nn}(\boldsymbol{X})| > t\big)\Big],
\end{aligned} \tag{21}$$

where the last inequality follows from the fact that $|\widetilde{\eta}(\boldsymbol{X}) - 1/2| \leq |\widetilde{\eta}(\boldsymbol{X}) - \widetilde{f}_{nn}(\boldsymbol{X})|$, when $\widetilde{s}_{nn}(\boldsymbol{X}) \neq f^*(\boldsymbol{X})$. Next, by Cauchy–Schwarz inequality, (21) can be further bounded as

$$\begin{aligned}
&2\mathbb{E}\Big[\big|\widetilde{\eta}(\boldsymbol{X}) - \widetilde{f}_{nn}(\boldsymbol{X})\big| \cdot I\big(|\widetilde{\eta}(\boldsymbol{X}) - 1/2| \leq t\big)\Big] + 2\mathbb{E}\Big[\big|\widetilde{\eta}(\boldsymbol{X}) - \widetilde{f}_{nn}(\boldsymbol{X})\big| \cdot I\big(|\widetilde{\eta}(\boldsymbol{X}) - 1/2| > t\big)\Big] \\
\leq& 2\|\widetilde{\eta} - \widetilde{f}_{nn}\|_{L^2(\mathbb{P}_{\boldsymbol{X}})} \sqrt{\mathbb{P}(|\widetilde{\eta}(\boldsymbol{X}) - 1/2| \leq t)} + 2\|\widetilde{\eta} - \widetilde{f}_{nn}\|_{L^2(\mathbb{P}_{\boldsymbol{X}})}^2/t \\
\leq& 2^{1-\gamma/2}\|\widetilde{\eta} - \widetilde{f}_{nn}\|_{L^2(\mathbb{P}_{\boldsymbol{X}})} c\kappa_\epsilon^{-\gamma/2} t^{\gamma/2} + 2\|\widetilde{\eta} - \widetilde{f}_{nn}\|_{L^2(\mathbb{P}_{\boldsymbol{X}})}^2/t.
\end{aligned}$$

Choosing $t = \|\widetilde{f}_{nn} - \widetilde{\eta}\|_{L^2(\mathbb{P}_{\boldsymbol{X}})}^{\frac{2}{\gamma+2}} \kappa_\epsilon^{\frac{\gamma}{\gamma+2}}$, we get

$$\widetilde{R}(\widetilde{s}_{nn}) - \widetilde{R}(f^*) \lesssim \kappa_\epsilon^{-\frac{\gamma}{\gamma+2}} \|\widetilde{\eta} - \widetilde{f}_{nn}\|_{L^2(\mathbb{P}_{\boldsymbol{X}})}^{\frac{2\gamma+2}{\gamma+2}}.$$

Subsequently, by Lemma 2, it follows that

$$R(\widetilde{s}_{nn}) - R(f^*) = \kappa_\epsilon^{-1}\Big(\widetilde{R}(\widetilde{s}_{nn}) - \widetilde{R}(f^*)\Big) \lesssim \Big(\kappa_\epsilon^{-2} \|\widetilde{\eta} - \widetilde{f}_{nn}\|_{L^2(\mathbb{P}_{\boldsymbol{X}})}^2\Big)^{\frac{\gamma+1}{\gamma+2}}. \tag{22}$$

Next, we proceed to establish the convergence rate of $\|\widetilde{\eta} - \widetilde{f}_{nn}\|_{L^2(\mathbb{P}_{\boldsymbol{X}})}^2$. Let $\mathcal{F}_{n,\delta_n}^{NN} = \{f \in \mathcal{F}_n^{NN} : \|\widetilde{\eta} - f\|_{L^2(\mathbb{P}_{\boldsymbol{X}})}^2 \geq \delta_n\}$. For any $\delta_n > 0$,

$$\mathbb{P}\Big(\|\widetilde{\eta} - \widetilde{f}_{nn}\|_{L^2(\mathbb{P}_{\boldsymbol{X}})}^2 \geq \delta_n\Big) \leq \mathbb{P}\Big(\inf_{f \in \mathcal{F}_{n,\delta_n}^{NN}} n^{-1} \sum_{i=1}^n (f_{nn}^*(\boldsymbol{x}_i) - \widetilde{z}_i)^2 - n^{-1} \sum_{i=1}^n (f(\boldsymbol{x}_i) - \widetilde{z}_i)^2 \geq \delta_n\Big),$$

where $f_{nn}^* = \arg\min_{f \in \mathcal{F}_n^{NN}} \|f - \widetilde{\eta}\|_{L^2(\mathbb{P}_{\boldsymbol{X}})}^2$.

For ease of notation, we denote that $U_n(f) = n^{-1} \sum_{i=1}^n (f(\boldsymbol{x}_i) - \widetilde{z}_i)^2$ and $U(f) = \mathbb{E}(f(\boldsymbol{X}) - \widetilde{Z})^2$. Then, we have

$$\mathbb{P}\Big(\|\widetilde{\eta} - \widetilde{f}_{nn}\|_{L^2(\mathbb{P}_{\boldsymbol{X}})}^2 \geq \delta_n\Big) \leq \mathbb{P}\Big(\inf_{f \in \mathcal{F}_{n,\delta_n}^{NN}} U_n(f_{nn}^*) - U_n(f) \geq 0\Big).$$

Notice that $\mathcal{F}_{n,\delta_n}^{NN}$ admits the decomposition as $\mathcal{F}_{n,\delta_n}^{NN} = \cup_{i=1}^{n} \mathcal{H}_i$ with $\mathcal{H}_i = \{f \in \mathcal{F}_n^{NN} : 2^{i-1}\delta_n \leq \|\widetilde{\eta} - f\|_{L^2(\mathbb{P}_{\boldsymbol{X}})}^2 \leq 2^i \delta_n\}$. Therefore, we further have

$$\mathbb{P}\Big( \|\widetilde{\eta} - \widetilde{f}_{nn}\|_{L^2(\mathbb{P}_{\boldsymbol{X}})}^2 \geq \delta_n \Big) \leq \sum_{i=1}^{n} \mathbb{P}\Big( \inf_{f \in \mathcal{H}_i} U_n(f_{nn}^*) - U_n(f) \geq 0 \Big).$$

Clearly, it suffices to bound $\mathbb{P}\Big( \inf_{f \in \mathcal{H}_i} U_n(f_{nn}^*) - U_n(f) \geq 0 \Big)$ for upper bounding $\mathbb{P}\Big( \|\widetilde{\eta} - \widetilde{f}_{nn}\|_{L^2(\mathbb{P}_{\boldsymbol{X}})}^2 \geq \delta_n \Big)$. For $i \geq 1$,

$$\mathbb{P}\Big( \inf_{f \in \mathcal{H}_i} U_n(f_{nn}^*) - U_n(f) \geq 0 \Big)$$

$$\leq \mathbb{P}\Big( \inf_{f \in \mathcal{H}_i} \big[ U_n(f_{nn}^*) - U(f_{nn}^*) \big] - \big[ U_n(f) - U(f) \big] \geq \inf_{f \in \mathcal{H}_i} U(f) - U(f_{nn}^*) \Big)$$

$$= \mathbb{P}\Big( \inf_{f \in \mathcal{H}_i} \big[ U_n(f_{nn}^*) - U(f_{nn}^*) \big] - \big[ U_n(f) - U(f) \big] \geq \inf_{f \in \mathcal{H}_i} U(f) - U(\widetilde{\eta}) + U(\widetilde{\eta}) - U(f_{nn}^*) \Big)$$

$$\leq \mathbb{P}\Big( \inf_{f \in \mathcal{H}_i} \big[ U_n(f_{nn}^*) - U(f_{nn}^*) \big] - \big[ U_n(f) - U(f) \big] \geq 2^{i-1}\delta_n - \|f_{nn}^* - \widetilde{\eta}\|_{L^2(\mathbb{P}_{\boldsymbol{X}})}^2 \Big).$$

Assuming $\|f_{nn}^* - \widetilde{\eta}\|_{L^2(\mathbb{P}_{\boldsymbol{X}})}^2 \leq \delta_n/2$ yields that

$$\mathbb{P}\Big( \inf_{f \in \mathcal{H}_i} U_n(f_{nn}^*) - U_n(f) \geq 0 \Big) \leq \mathbb{P}\Big( \inf_{f \in \mathcal{H}_i} \big[ U_n(f_{nn}^*) - U(f_{nn}^*) \big] - \big[ U_n(f) - U(f) \big] \geq 2^{i-2}\delta_n \Big).$$

Denote that $M_i = 2^{i-2}\delta_n$. We turn to establish the relation between the variance of $\big(f_{nn}^*(\boldsymbol{X}) - \widetilde{Z}\big)^2 - \big(f(\boldsymbol{X}) - \widetilde{Z}\big)^2$ and $M_i$.

$$\sup_{f \in \mathcal{H}_i} \text{Var}\Big[ \big(f_{nn}^*(\boldsymbol{X}) - \widetilde{Z}\big)^2 - \big(f(\boldsymbol{X}) - \widetilde{Z}\big)^2 \Big]$$

$$= \sup_{f \in \mathcal{H}_i} \text{Var}\Big[ \big(f_{nn}^*(\boldsymbol{X}) - f(\boldsymbol{X})\big)\big(f_{nn}^*(\boldsymbol{X}) + f(\boldsymbol{X}) - 2\widetilde{Z}\big) \Big]$$

$$\leq \sup_{f \in \mathcal{H}_i} \mathbb{E}\Big[ \big(f_{nn}^*(\boldsymbol{X}) - f(\boldsymbol{X})\big)^2 \big(f_{nn}^*(\boldsymbol{X}) + f(\boldsymbol{X}) - 2\widetilde{Z}\big)^2 \Big] \leq \sup_{f \in \mathcal{H}_i} 4V_n^2 \|f_{nn}^* - f\|_{L^2(\mathbb{P}_{\boldsymbol{X}})}^2$$

$$\leq \sup_{f \in \mathcal{H}_i} 8V_n^2 \Big( \|f_{nn}^* - \widetilde{\eta}\|_{L^2(\mathbb{P}_{\boldsymbol{X}})}^2 + \|\widetilde{\eta} - f\|_{L^2(\mathbb{P}_{\boldsymbol{X}})}^2 \Big) \leq 64V_n^2 M_i \equiv V_i.$$

In the following, we proceed to verify conditions (4.5)-(4.7) in Shen & Wong (1994). First, the relation between $M_i$ and $V_i$ directly implies (4.6) with $T = 32V_n^2$ and $\epsilon = 1/2$. Second, by Lemma 5 of Schmidt-Hieber (2020),

$$\log \mathcal{N}\Big( \epsilon, \mathcal{F}_n^{NN}(L_n, N_n, P_n, B_n, V_n), \|\cdot\|_{L^\infty(\mathbb{P}(\boldsymbol{X}))} \Big) \leq$$

$$2L_n(P_n + 1)\log\Big( \epsilon^{-1}(L_n + 1)(N_n + 1)\max\{B_n, 1\} \Big).$$

It then follows that

$$\int_{\frac{\epsilon}{32}M_i}^{V_i^{1/2}} \sqrt{\log \mathcal{N}\Big( \epsilon, \mathcal{F}_n^{NN}(L_n, N_n, P_n, B_n, V_n), \|\cdot\|_{L^\infty(\mathbb{P}(\boldsymbol{X}))} \Big)} d\epsilon / M_i$$

$$\leq \int_{\frac{\epsilon}{32}M_i}^{V_i^{1/2}} \sqrt{2L_n(P_n + 1)\log\Big( \epsilon^{-1}(L_n + 1)(N_n + 1)\max\{B_n, 1\} \Big)} d\epsilon / M_i. \tag{23}$$

Notice that the right-hand side of (23) is non-increasing in $i$ and $M_i$, it then follows that

$$\int_{\frac{\epsilon}{32}M_i}^{V_i^{1/2}} \sqrt{2L_n(P_n + 1)\log\Big( \epsilon^{-1}(L_n + 1)(N_n + 1)\max\{B_n, 1\} \Big)} d\epsilon / M_i$$

$$\leq \int_{\frac{\epsilon}{32}M_1}^{V_1^{1/2}} \sqrt{2L_n(P_n + 1)\log\Big( \epsilon^{-1}(L_n + 1)(N_n + 1)\max\{B_n, 1\} \Big)} d\epsilon / M_1.$$

With this, condition (4.7) can be satisfied by imposing

$$\int_{\frac{\epsilon}{32}M_1}^{V_1^{1/2}} \sqrt{2L_n(P_n+1)\log\left(\epsilon^{-1}(L_n+1)(N_n+1)\max\{B_n,1\}\right)}d\epsilon/M_1 \lesssim n^{1/2},\tag{24}$$

which directly implies the condition (4.5) by appropriate choices of $L_n$ and $S_n$. By Theorem 3 in Shen & Wong (1994), we get

$$\mathbb{P}\Big(\|\widetilde{\eta}-\widetilde{f}_{nn}\|_{L^2(\mathbb{P}_{\boldsymbol{X}})}^2 \geq \delta_n\Big) \lesssim \sum_{i=1}^{\infty}\exp\Big(-\frac{nM_i^2}{512V_n^2M_i+2M_i/3}\Big) \lesssim \sum_{i=1}^{\infty}\exp\Big(-n2^{i-2}\delta_n\Big).$$

By the fact $2^{i-2} \geq (i-1/2)$ for $i \geq 1$, there exists some constants $C$ such that

$$\mathbb{P}\Big(\|\widetilde{\eta}-\widetilde{f}_{nn}\|_{L^2(\mathbb{P}_{\boldsymbol{X}})}^2 \geq \delta_n\Big) \lesssim \exp(-Cn\delta_n),$$

provided that $n\delta = o(1)$.

Next, we proceed to consider the approximation error of neural network to meet the assumption that $\|f_{nn}^* - \eta\|_{L^\infty(\mathbb{P}_{\boldsymbol{X}})}^2 \leq \delta_n$. Notice that $\eta(\boldsymbol{X}) \in \mathcal{H}(\beta,[0,1]^p,M)$. By Theorem 5 of Schmidt-Hieber (2020), there exists a class of neural networks $\mathcal{F}_n^{NN}(L_n,N_n,P_n,B_n,V_n)$ such that for any $0 < \psi_n < 1$

$$\inf_{f\in\mathcal{F}_n^{NN}(L_n,N_n,P_n,B_n,V_n)} \|f-\eta\|_{L^\infty(\mathbb{P}_{\boldsymbol{X}})} \leq \kappa_\epsilon^{-1}\psi_n,$$

where $P_n \asymp (\kappa_\epsilon^{-1}\psi_n)^{-p/\beta}\log(\kappa_\epsilon/\psi_n)$, $N_n \asymp (\kappa_\epsilon^{-1}\psi_n)^{-p/\beta}$, $B_n = 1$, $V_n \geq M+1$ and $L_n \asymp \log(\kappa_\epsilon/\psi_n)$. Let $\eta_{nn}^*$ denote the optimal function in $\mathcal{F}_n^{NN}(L_n,N_n,P_n,B_n,\infty)$ to approximate $\eta$. Suppose that $\eta_{nn}^*$ is a $L$-layer neural network and formulated as

$$\eta_{nn}^*(\boldsymbol{x}) = \boldsymbol{A}_{L+1}\boldsymbol{g}_L(\boldsymbol{x}) + \boldsymbol{b}_{L+1},$$

where $\boldsymbol{g}_L(\boldsymbol{x}) = \boldsymbol{h}_L \circ \boldsymbol{h}_{L-1} \circ \cdots \circ h_1(\boldsymbol{x})$. We construct a new neural network that $\widetilde{\eta}_{nn}$ as

$$\begin{aligned}
\widetilde{\eta}_{nn}(\boldsymbol{x}) &= \theta\eta_{nn}^*(\boldsymbol{x}) + (1-\theta)(1-\eta_{nn}^*(\boldsymbol{x}))\\
&= \theta\boldsymbol{A}_{L+1}\boldsymbol{g}_L(\boldsymbol{x}) + \theta\boldsymbol{b}_{L+1} + (1-\theta)(-\boldsymbol{A}_{L+1}\boldsymbol{g}_L(\boldsymbol{x}) + (1-\boldsymbol{b}_{L+1}))\\
&= (2\theta-1)\boldsymbol{A}_{L+1}\boldsymbol{g}_L(\boldsymbol{x}) + (2\theta-1)\boldsymbol{b}_{L+1} + (1-\theta)\mathbf{1}_{L+1}.
\end{aligned}$$

It can be easily verified that $\widetilde{\eta}_{nn} \in \mathcal{F}_n^{NN}(L_n,N_n,P_n,B_n,V_n)$. This along with the fact that $\widetilde{\eta}(\boldsymbol{x}) = \theta\eta(\boldsymbol{x}) + (1-\theta)(1-\eta(\boldsymbol{x}))$ with $B_n \geq 1$ results in

$$\begin{aligned}
\|\widetilde{\eta}_{nn}-\widetilde{\eta}\|_{L^\infty(\mathbb{P}_{\boldsymbol{X}})} &= \|\theta\eta_{nn}^* + (1-\theta)(1-\eta_{nn}^*) - \theta\eta - (1-\theta)(1-\eta)\|_{L^\infty(\mathbb{P}_{\boldsymbol{X}})}\\
&= (2\theta-1)\|\eta_{nn}^*-\eta\|_{L^\infty(\mathbb{P}_{\boldsymbol{X}})} = \kappa_\epsilon\|\eta_{nn}^*-\eta\|_{L^\infty(\mathbb{P}_{\boldsymbol{X}})} \leq \psi_n.
\end{aligned}$$

Therefore,

$$\|f_{nn}^*-\widetilde{\eta}\|_{L^\infty(\mathbb{P}_{\boldsymbol{X}})}^2 = \inf_{f\in\mathcal{F}_n^{NN}(L_n,N_n,P_n,B_n,V_n)} \|f-\widetilde{\eta}\|_{L^\infty(\mathbb{P}_{\boldsymbol{X}})}^2 \leq \|\widetilde{\eta}_{nn}-\widetilde{\eta}\|_{L^\infty(\mathbb{P}_{\boldsymbol{X}})}^2 \leq \psi_n^2.$$

Plugging $P_n \asymp (\kappa_\epsilon^{-1}\psi_n)^{-p/\beta}\log(\kappa_\epsilon/\psi_n)$, $N_n \asymp (\kappa_\epsilon^{-1}\psi_n)^{-p/\beta}$, and $L_n \asymp \log(\kappa_\epsilon/\psi_n)$ into (24) yields that $\kappa_\epsilon^{p/(2\beta)}\psi_n^{-p/(2\beta)}\log(\psi_n^{-1}) \lesssim (n\delta_n)^{1/2}$. Combining this with the assumption that $\psi_n^2 \lesssim \delta_n$, it follows that $\delta_n \asymp \psi_n^2 \asymp \kappa_\epsilon^{2p/(2\beta+p)}(\log n/n)^{2\beta/(2\beta+p)}$. Plugging this into (22) yields that

$$\mathbb{E}\Big[R(\widetilde{s}_{nn}) - R(f^*)\Big] \lesssim \Big(\frac{\log n}{n\kappa_\epsilon^2}\Big)^{\frac{2\beta(\gamma+1)}{2\beta(\gamma+2)+p(\gamma+2)}}.\tag{25}$$

Notice that the proof of (25) is independent of the distribution of $\boldsymbol{X}$, therefore (25) holds for any distribution in $\mathcal{P}_{\gamma,\beta}$, which implies that

$$\sup_{\pi\in\mathcal{P}_{\gamma,\beta}} \mathbb{E}\Big[R(\widetilde{s}_{nn}) - R(f^*)\Big] \lesssim \Big(\frac{\log n}{n\kappa_\epsilon^2}\Big)^{\frac{2\beta(\gamma+1)}{2\beta(\gamma+2)+p(\gamma+2)}}.$$

**Proof of the lower bound.** The proof is based on the well-known Assouad's lemma. The overall proof for the lower bound is mainly based on the proofs Theorem 3.5 and 4.1 in Audibert & Tsybakov (2007). We first introduce the partition $\{\mathcal{X}_i\}_{i=0}^m$ of the cube $[0,1]^p$ using the grid $G_q \subseteq [0,1]^p$ defined by

$$G_q = \left\{ \left( \frac{2k_1+1}{2q}, \cdots, \frac{2k_p+1}{2q} \right) : k_i \in \{0, \cdots, q-1\}, i \in \{1, \cdots, p\} \right\},$$

where $q \geq 1$ is an integer. For any $\boldsymbol{x} \in \mathbb{R}^p$, let $n_q(\boldsymbol{x}) \in G_q$ be the unique point which is the closest point to $\boldsymbol{x} \in \mathbb{R}^p$ among all points in $G_q$. Without loss of generality, we assume the uniqueness of $n_q(\boldsymbol{x})$ by choosing the one closest to 0. We define a partition $\{\mathcal{X}_i'\}_{i=1}^{q^p}$ of $\mathbb{R}^p$ as follows. For $\boldsymbol{x}, \boldsymbol{y} \in \mathbb{R}^p$, $\boldsymbol{x}$ and $\boldsymbol{y}$ are in the same cell $\mathcal{X}_i$ if and only if $n_q(\boldsymbol{x}) = n_q(\boldsymbol{y})$. Then, for $m \leq q^p$, we define $\mathcal{X}_i$ as $\mathcal{X}_i = \mathcal{X}_i'$ for $1 \leq i \leq m$ and $\mathcal{X}_0 = \mathbb{R}^p / \cup_{i=1}^m \mathcal{X}_i$.

Let $u : \mathbb{R}_+ \to \mathbb{R}_+$ be a non-increasing infinitely differentiable function such that $u = 1$ on $[0, 1/4]$ and $u = 0$ on $[1/2, \infty)$. Let $\phi := C_\phi u(\|\boldsymbol{x}\|)$, where $C_\phi \leq 1$ is taken small enough such that $\phi \in \mathcal{H}(\beta, [0,1]^p, M)$. For a given $\boldsymbol{\sigma} = (\sigma_1, \cdots, \sigma_m) \in \{\pm 1\}^m$, we construct a distribution $\pi_{\boldsymbol{\sigma}}$ on $\mathbb{R}^p \times \{-1, 1\}$ as follows. Let $\mu$ be the Lebesgue measure. For $0 < \omega < \frac{1}{m}$ and $A_0 \subseteq \mathcal{X}_0$ with $\mu(A_0) > 0$, we construct the marginal distribution $\mathbb{P}_{\boldsymbol{X}}$ on $\mathbb{R}^p$ that has the density function

$$\mathbb{P}_{\boldsymbol{X}}(\boldsymbol{x}) = \begin{cases} \frac{\omega}{\mu(\mathcal{B}(0, q^{-1}/4))}, & \boldsymbol{x} \in \mathcal{B}(z, q^{-1}/4) \text{ for some } z \in G_q, \\ (1 - m\omega)/\mu(A_0), & \boldsymbol{x} \in A_0, \\ 0, & \text{otherwise.} \end{cases}$$

The conditional distribution on $\{-1, 1\}$ is defined by

$$\eta_{\boldsymbol{\sigma}}(\boldsymbol{x}) = \mathbb{P}(Y = 1 | \boldsymbol{X} = \boldsymbol{x}) = \begin{cases} \frac{1 + \sigma_j \varphi(\boldsymbol{x})}{2}, & \text{for } \boldsymbol{x} \in \mathcal{X}_j, j = 1, \cdots, m, \\ 1/2, & \boldsymbol{x} \in \mathcal{X}_0, \end{cases}$$

where $\varphi(\boldsymbol{x}) = q^{-\beta} \phi(q(\boldsymbol{x} - n_q(\boldsymbol{x})))$. Correspondingly, for any $\boldsymbol{\sigma}$, $\widetilde{\eta}_{\boldsymbol{\sigma}}(\boldsymbol{x})$ is given as

$$\widetilde{\eta}_{\boldsymbol{\sigma}}(\boldsymbol{x}) = \theta \widetilde{\eta}_{\boldsymbol{\sigma}}(\boldsymbol{x}) + (1 - \theta)(1 - \widetilde{\eta}_{\boldsymbol{\sigma}}(\boldsymbol{x})) = \begin{cases} \frac{1 + (2\theta - 1)\sigma_j \varphi(\boldsymbol{x})}{2}, & \text{for } \boldsymbol{x} \in \mathcal{X}_j, j = 1, \cdots, m, \\ 1/2, & \boldsymbol{x} \in \mathcal{X}_0, \end{cases}$$

Notice that $D^s \varphi = q^{|s| - \beta} D^s(q(\boldsymbol{x} - n_q(\boldsymbol{x})))$ for any $s \in \mathbb{N}$ with $|s| \leq \beta$. Thus, $\eta_{\boldsymbol{\sigma}}(\boldsymbol{x})$ belongs to $\mathcal{H}(\beta, [0,1]^p, M)$.

$$\mathbb{P}\left( |\eta_{\boldsymbol{\sigma}}(\boldsymbol{x}) - 1/2| \leq t \right) = m \mathbb{P}\left( \phi(q(\boldsymbol{x} - n_q(\boldsymbol{x}))) \leq 2t q^\beta \right)$$

$$= m \int_{\mathcal{B}(\boldsymbol{x}_0, (4q)^{-1})} I\left( \phi(q(\boldsymbol{x} - \boldsymbol{x}_0)) \leq 2t q^\beta \right) \frac{w}{\mu(\mathcal{B}(\boldsymbol{0}, (4q)^{-1}))} d\boldsymbol{x}$$

$$= m \int_{\mathcal{B}(\boldsymbol{0}, 1/4)} I\left( \phi(\boldsymbol{x}) \leq 2t q^\beta \right) \frac{w}{\mu(\mathcal{B}(\boldsymbol{0}, 1/4))} d\boldsymbol{x} = m w I\left( t \geq C_\phi/(2q^\beta) \right),$$

where $\boldsymbol{x}_0 = \left( \frac{1}{2K}, \ldots, \frac{1}{2K} \right)$. Clearly, the low-noise assumption of $\eta_{\boldsymbol{\sigma}}$ can be satisfied by setting $mw \leq C_\phi^\gamma/(2q^\beta)^\gamma$. Let $\mathcal{P}_{\gamma, \beta}$ denote the set of joint distributions of $(\boldsymbol{X}, Y)$ satisfying the low-noise assumption and $\eta(\boldsymbol{x}) \in \mathcal{H}(\beta, [0,1]^p, M)$. For any $\boldsymbol{\sigma}$, we have $\pi_{\boldsymbol{\sigma}} \in \mathcal{P}_{\gamma, \beta}$, implying $\mathcal{P}' = \{\pi_{\boldsymbol{\sigma}} : \boldsymbol{\sigma} \in \{-1, 1\}^m\} \subset \mathcal{P}_{\gamma, \beta}$. Therefore,

$$\sup_{\pi \in \mathcal{P}_{\gamma, \beta}} \mathbb{E}_{\widetilde{\mathcal{D}}}\left[ R(\widetilde{s}_{nn}) - R(f^*) \right] \geq \sup_{\pi_{\boldsymbol{\sigma}} \in \mathcal{P}'} \mathbb{E}_{\widetilde{\mathcal{D}}}\left[ R(\widetilde{s}_{nn}) - R(f^*) \right].$$

Let $\widetilde{P}'$ be a set of probability measures on $(\boldsymbol{X}, \widetilde{Y})$ satisfying that for each $\pi_{\boldsymbol{\sigma}} \in \mathcal{P}'$ there exists an $\widetilde{\pi}_{\boldsymbol{\sigma}} \in \widetilde{\mathcal{P}}'$ such that $\pi_{\boldsymbol{\sigma}}$ and $\widetilde{\pi}_{\boldsymbol{\sigma}}$ have the same marginal distribution of $\boldsymbol{X}$ and $\widetilde{\eta}_{\boldsymbol{\sigma}}(\boldsymbol{x}) = \theta \eta_{\boldsymbol{\sigma}}(\boldsymbol{x}) + (1 - \theta)(1 - \eta_{\boldsymbol{\sigma}}(\boldsymbol{x}))$. It follows that

$$\sup_{\pi_{\boldsymbol{\sigma}} \in \mathcal{P}'} \mathbb{E}_{\widetilde{\mathcal{D}}}\left[ R(\widetilde{s}_{nn}) - R(f^*) \right] \geq (2\theta - 1)^{-1} \sup_{\widetilde{\pi}_{\boldsymbol{\sigma}} \in \widetilde{\mathcal{P}}'} \mathbb{E}_{\widetilde{\mathcal{D}}}\left[ \widetilde{R}(\widetilde{s}_{nn}) - \widetilde{R}(f^*) \right].$$

Next, we proceed to bound $\sup_{\widetilde{\pi}_{\boldsymbol{\sigma}}\in\widetilde{\mathcal{P}}'}\mathbb{E}_{\widetilde{\mathcal{D}}}\left[\widetilde{R}(\widetilde{s}_{nn}) - \widetilde{R}(f^*)\right]$. Notice that $f^*$ varies with the value of $\boldsymbol{\sigma}$, therefore we use $f^*_{\boldsymbol{\sigma}}$ to characterize its dependence on $\boldsymbol{\sigma}$.

$$\sup_{\widetilde{\pi}_{\boldsymbol{\sigma}}\in\widetilde{\mathcal{P}}'}\mathbb{E}_{\widetilde{\mathcal{D}}}\left[\widetilde{R}(\widetilde{s}_{nn}) - \widetilde{R}(f^*)\right] = \sup_{\widetilde{\pi}_{\boldsymbol{\sigma}}\in\widetilde{\mathcal{P}}'}\left\{\mathbb{E}_{\widetilde{\mathcal{D}}}\left[\mathbb{E}_{\boldsymbol{X}\sim\mathbb{P}_{\boldsymbol{X}}}\left[|2\widetilde{\eta}_{\boldsymbol{\sigma}}(\boldsymbol{X}) - 1|\mathbb{1}_{\{\widetilde{s}_{nn}(\boldsymbol{X})\neq\sigma_j;\boldsymbol{X}\in\mathcal{X}_j\}}\right]\right]\right\}$$
$$= (2\theta - 1)\sup_{\widetilde{\pi}_{\boldsymbol{\sigma}}\in\widetilde{\mathcal{P}}'}\left\{\mathbb{E}_{\widetilde{\mathcal{D}}}\left[\mathbb{E}_{\boldsymbol{X}\sim\mathbb{P}_{\boldsymbol{X}}}\left[\varphi(\boldsymbol{X})\mathbb{1}_{\{\widetilde{s}_{nn}(\boldsymbol{X})\neq\sigma_j;\boldsymbol{X}\in\mathcal{X}_j\}}\right]\right]\right\},$$

where $\mathbb{1}_{\{x\}} = 1$ if the statement $x$ is true.

Let $\Pi$ be the distribution of a Rademacher random variable $\sigma$, that is, $\Pi(\sigma = 1) = \Pi(\sigma = -1) = 1/2$. Then

$$\sup_{\widetilde{\pi}_{\boldsymbol{\sigma}}\in\widetilde{\mathcal{P}}'}\left\{\mathbb{E}_{\widetilde{\mathcal{D}}}\left[\mathbb{E}_{\boldsymbol{X}\sim\mathbb{P}_{\boldsymbol{X}}}\left[\varphi(\boldsymbol{X})\mathbb{1}_{\{\widetilde{s}_{nn}(\boldsymbol{X})\neq f^*_{\boldsymbol{\sigma}}(\boldsymbol{X})\}}\right]\right]\right\} \geq \mathbb{E}_{\Pi^m}\left\{\mathbb{E}_{\widetilde{\mathcal{D}}}\left[\mathbb{E}_{\boldsymbol{X}\sim\mathbb{P}_{\boldsymbol{X}}}\left[\varphi(\boldsymbol{X})\mathbb{1}_{\{\widetilde{s}_{nn}(\boldsymbol{X})\neq f^*_{\boldsymbol{\sigma}}(\boldsymbol{X})\}}\right]\right]\right\}.$$

Note that for $\boldsymbol{x}\in\mathcal{X}_0$, $\|\boldsymbol{x} - n_q(\boldsymbol{x})\| \geq (2q)^{-1}$ and $\varphi(\boldsymbol{x}) = 0$. Thus, we have

$$\mathbb{E}_{\Pi^m}\left\{\mathbb{E}_{\widetilde{\mathcal{D}}}\left[\mathbb{E}_{\boldsymbol{X}\sim\mathbb{P}_{\boldsymbol{X}}}\left[\varphi(\boldsymbol{X})\mathbb{1}_{\{\widetilde{s}_{nn}(\boldsymbol{X})\neq f^*_{\boldsymbol{\sigma}}(\boldsymbol{X})\}}\right]\right]\right\} = \sum_{j=1}^m\mathbb{E}_{\Pi^m}\left\{\mathbb{E}_{\widetilde{\mathcal{D}}}\left[\mathbb{E}_{\boldsymbol{X}\sim\mathbb{P}_{\boldsymbol{X}}}\left[\varphi(\boldsymbol{X})\mathbb{1}_{\{\widetilde{s}_{nn}(\boldsymbol{X})\neq\sigma_j;\boldsymbol{X}\in\mathcal{X}_j\}}\right]\right]\right\}.$$

Let $\boldsymbol{\sigma}_{j,r} = (\sigma_1,\cdots,\sigma_{j-1},r,\sigma_{j+1},\cdots,\sigma_m)$ for $r\in\{0,\pm 1\}$. Here $\boldsymbol{\sigma}_{j,r}$ denotes a vector deduced from $\boldsymbol{\sigma}$ by fixing its $j$-th element to $r$. We have

$$\mathbb{E}_{\Pi^m}\left\{\mathbb{E}_{\widetilde{\mathcal{D}}}\left[\mathbb{E}_{\boldsymbol{X}\sim\mathbb{P}_{\boldsymbol{X}}}\left[\varphi(\boldsymbol{X})\mathbb{1}_{\{\widetilde{s}_{nn}(\boldsymbol{X})\neq\sigma_j;\boldsymbol{X}\in\mathcal{X}_j\}}\right]\right]\right\}$$
$$=\mathbb{E}_{\Pi^m}\left\{\mathbb{E}_{\widetilde{\pi}^n_{\boldsymbol{\sigma}_{j,0}}}\left[\frac{d\widetilde{\pi}^n_{\boldsymbol{\sigma}}}{d\widetilde{\pi}^n_{\boldsymbol{\sigma}_{j,0}}}\mathbb{E}_{\boldsymbol{X}\sim\mathbb{P}_{\boldsymbol{X}}}\left[\varphi(\boldsymbol{X})\mathbb{1}_{\{\widetilde{s}_{nn}(\boldsymbol{X})\neq\sigma_j;\boldsymbol{X}\in\mathcal{X}_j\}}\right]\right]\right\}$$
$$=\mathbb{E}_{\Pi^{m-1}}\left\{\mathbb{E}_{\widetilde{\pi}^n_{\boldsymbol{\sigma}_{j,0}}}\mathbb{E}_{\sigma_j\sim\Pi}\left[\frac{d\widetilde{\pi}^n_{\boldsymbol{\sigma}}}{d\widetilde{\pi}^n_{\boldsymbol{\sigma}_{j,0}}}\mathbb{E}_{\boldsymbol{X}\sim\mathbb{P}_{\boldsymbol{X}}}\left[\varphi(\boldsymbol{X})\mathbb{1}_{\{\widetilde{s}_{nn}(\boldsymbol{X})\neq\sigma_j;\boldsymbol{X}\in\mathcal{X}_j\}}\right]\right]\right\}.$$

where $\widetilde{\pi}_{\boldsymbol{\sigma}_{j,0}}$ has the same marginal distribution as $\mathbb{P}_{\boldsymbol{X}}$ and $\widetilde{\pi}_{\boldsymbol{\sigma}_{j,0}}$ and $\widetilde{\pi}_{\boldsymbol{\sigma}}$ differ in the conditional distribution over the points in $\mathcal{X}_j$. Specifically,

$$\widetilde{\eta}_{\boldsymbol{\sigma}_{j,0}}(\boldsymbol{x}) = \begin{cases} 1/2, & \text{if } \boldsymbol{x}\in\mathcal{X}_j, \\ \widetilde{\eta}_{\boldsymbol{\sigma}}(\boldsymbol{x}), & \text{otherwise.} \end{cases}$$

It then follows that

$$\mathbb{E}_{\Pi^{m-1}}\left\{\mathbb{E}_{\widetilde{\pi}^n_{\boldsymbol{\sigma}_{j,0}}}\mathbb{E}_{\sigma_j\sim\Pi}\left[\frac{d\widetilde{\pi}^n_{\boldsymbol{\sigma}}}{d\widetilde{\pi}^n_{\boldsymbol{\sigma}_{j,0}}}\mathbb{E}_{\boldsymbol{X}\sim\mathbb{P}_{\boldsymbol{X}}}\left[\varphi(\boldsymbol{X})\mathbb{1}_{\{\widetilde{s}_{nn}(\boldsymbol{X})\neq\sigma_j;\boldsymbol{X}\in\mathcal{X}_j\}}\right]\right]\right\}$$
$$=\mathbb{E}_{\Pi^{m-1}}\left\{\mathbb{E}_{\widetilde{\pi}^n_{\boldsymbol{\sigma}_{j,0}}}\mathbb{E}_{\sigma_j\sim\Pi}\left[\frac{d\widetilde{\pi}^n_{\boldsymbol{\sigma}}}{d\widetilde{\pi}^n_{\boldsymbol{\sigma}_{j,0}}}\mathbb{E}_{\boldsymbol{X}\sim\mathbb{P}_{\boldsymbol{X}}}\left[\varphi(\boldsymbol{X})\mathbb{1}_{\{\boldsymbol{X}\in\mathcal{X}_j\}}\mathbb{1}_{\{\widetilde{s}_{nn}(\boldsymbol{X})\neq\sigma_j\}}\right]\right]\right\}$$
$$=\mathbb{E}_{\Pi^{m-1}}\left\{\mathbb{E}_{\boldsymbol{X}\sim\mathbb{P}_{\boldsymbol{X}}}\left[\varphi(\boldsymbol{X})\mathbb{1}_{\{\boldsymbol{X}\in\mathcal{X}_j\}}\mathbb{E}_{\widetilde{\pi}^n_{\boldsymbol{\sigma}_{j,0}}}\mathbb{E}_{\sigma_j\sim\Pi}\left[\frac{d\widetilde{\pi}^n_{\boldsymbol{\sigma}}}{d\widetilde{\pi}^n_{\boldsymbol{\sigma}_{j,0}}}\mathbb{1}_{\{\widetilde{s}_{nn}(\boldsymbol{X})\neq\sigma_j\}}\right]\right]\right\}$$
$$\geq\mathbb{E}_{\Pi^{m-1}}\left\{\mathbb{E}_{\boldsymbol{X}\sim\mathbb{P}_{\boldsymbol{X}}}\left[\varphi(\boldsymbol{X})\mathbb{1}_{\{\boldsymbol{X}\in\mathcal{X}_j\}}\left(\frac{1}{2}\mathbb{P}_{\widetilde{\pi}^n_{\boldsymbol{\sigma}_{j,1}}}(\widetilde{s}_{nn}(\boldsymbol{X})\neq 1) + \frac{1}{2}\mathbb{P}_{\widetilde{\pi}^n_{\boldsymbol{\sigma}_{j,-1}}}(\widetilde{s}_{nn}(\boldsymbol{X})\neq -1)\right)\right]\right\}$$
$$\geq\frac{1}{2}\mathbb{E}_{\Pi^{m-1}}\left\{\mathbb{E}_{\boldsymbol{X}\sim\mathbb{P}_{\boldsymbol{X}}}\left[\varphi(\boldsymbol{X})\mathbb{1}_{\{\boldsymbol{X}\in\mathcal{X}_j\}}\right]\left(1 - \text{TV}(\widetilde{\pi}^n_{\boldsymbol{\sigma}_{j,1}},\widetilde{\pi}^n_{\boldsymbol{\sigma}_{j,-1}})\right)\right\},$$

where TV is the total variation distance between two distributions. Since $\mathbb{P}_{\boldsymbol{X}}(\mathcal{X}_j) = \omega$, we have

$$\mathbb{E}_{\Pi^m}\left\{\mathbb{E}_{\widetilde{\mathcal{D}}}\left[\mathbb{E}_{\boldsymbol{X}\sim\mathbb{P}_{\boldsymbol{X}}}\left[\varphi(\boldsymbol{X})\mathbb{1}_{\{\widetilde{s}_{nn}(\boldsymbol{X})\neq\sigma_j;\boldsymbol{X}\in\mathcal{X}_j\}}\right]\right]\right\} \geq \frac{\omega}{2}\mathbb{E}_{\boldsymbol{X}\sim\mathbb{P}_{\boldsymbol{X}}}\left(\varphi(\boldsymbol{X})|\boldsymbol{X}\in\mathcal{X}_j\right)\left(1 - \text{TV}(\widetilde{\pi}^n_{\boldsymbol{\sigma}_{j,1}},\widetilde{\pi}^n_{\boldsymbol{\sigma}_{j,-1}})\right),$$

Notice that the above inequality holds for any index $j \in [m]$. In conclusion, we obtain

$$\sum_{j=1}^{m} \mathbb{E}_{\Pi^m} \left\{ \mathbb{E}_{\widetilde{\pi}_{\boldsymbol{\sigma}}^n} \left[ \mathbb{E}_{\boldsymbol{X} \sim \mathbb{P}_{\boldsymbol{X}}} \left[ \varphi(X) \mathbb{1}_{[\widetilde{s}_{nn}(\boldsymbol{X}) \neq \sigma_j; \boldsymbol{X} \in \mathcal{X}_j]} \right] \right] \right\} \geq \frac{m\omega}{2} \mathbb{E}_{\boldsymbol{X} \sim \mathbb{P}_{\boldsymbol{X}}} \left( \varphi(\boldsymbol{X}) | \boldsymbol{X} \in \mathcal{X}_j \right) \left( 1 - \mathrm{TV}(\widetilde{\pi}_{\boldsymbol{\sigma}_{1,1}}^n, \widetilde{\pi}_{\boldsymbol{\sigma}_{1,-1}}^n) \right).$$

Now we bound $\mathrm{TV}(\widetilde{\pi}_{\boldsymbol{\sigma}_{1,1}}^n, \widetilde{\pi}_{\boldsymbol{\sigma}_{1,-1}}^n)$. First, it holds

$$\mathrm{TV}(\widetilde{\pi}_{\boldsymbol{\sigma}_{1,1}}^n, \widetilde{\pi}_{\boldsymbol{\sigma}_{1,-1}}^n) = \sum_{l=1}^{n} \binom{n}{l} \omega^l (1-\omega)^{n-l} \mathcal{V}_l,$$

where $\mathcal{V}_l = \mathrm{TV}(\widetilde{\pi}_{-1}^l, \widetilde{\pi}_1^l)$ with $\widetilde{\pi}_r = \widetilde{\pi}_{\boldsymbol{\sigma}_{1,r}}(\cdot | \boldsymbol{X} \in \mathcal{X}_1)$. Note that

$$\mathcal{V}_l \leq H(\widetilde{\pi}_{-1}^l, \widetilde{\pi}_1^l) = \sqrt{2 \left( 1 - \left[ 1 - \frac{H^2(\widetilde{\pi}_{-1}, \widetilde{\pi}_1)}{2} \right]^l \right)}$$

where $H$ is the Hellinger distance and

$$1 - \frac{H^2(\widetilde{\pi}_{-1}, \widetilde{\pi}_1)}{2} = \mathbb{E}_{\boldsymbol{X} \sim \mathbb{P}_{\boldsymbol{X}}} \left( \sqrt{1 - (2\theta-1)^2 \varphi(\boldsymbol{X})} \Big| \boldsymbol{X} \in \mathcal{X}_1 \right) =: \sqrt{1 - b^2}$$

Since $1 - (1-x^2)^{l/2} \leq \frac{lx^2}{2}$ for $l \geq 2$ and $x > 0$, we have $\mathcal{V}_l \leq b\sqrt{l}$ and

$$\mathrm{TV}(\widetilde{\pi}_{\boldsymbol{\sigma}_{1,1}}^n, \widetilde{\pi}_{\boldsymbol{\sigma}_{1,-1}}^n) \leq b \sum_{l=1}^{n} \mathbb{P}\left( \sum_{i=1}^{n} \epsilon_{i,\omega} = l \right) \sqrt{l} \leq b\sqrt{n\omega},$$

where $\epsilon_i$ are i.i.d. random variables such that $\mathbb{P}(\epsilon_i = 1) = \omega = 1 - \mathbb{P}(\epsilon_i = -1)$. In conclusion, we have

$$\sup_{\widetilde{\pi}_{\boldsymbol{\sigma}} \in \widetilde{\mathcal{P}}'} \left\{ \mathbb{E}_{\widetilde{\mathcal{D}}} \left[ \widetilde{R}(\widetilde{s}_{nn}) \right] - \widetilde{R}^* \right\} \geq m\omega b'(1 - b\sqrt{n\omega}),$$

where

$$b = \left[ 1 - \left( \mathbb{E}\left[ \sqrt{1 - (2\theta-1)^2 \varphi^2(X)} \Big| \boldsymbol{X} \in \mathcal{X}_j \right] \right)^2 \right]^{1/2} \asymp (2\theta-1)q^{-\beta},$$

$$b' = \mathbb{E}\left( (2\theta-1)\varphi(\boldsymbol{X}) \Big| \boldsymbol{X} \in \mathcal{X}_j \right) \asymp (2\theta-1)q^{-\beta}.$$

As a result, we have

$$\sup_{\pi \in \mathcal{P}_{\gamma,\beta}} \mathbb{E}_{\widetilde{\mathcal{D}}} \left[ R(\widetilde{s}_{nn}) - R^* \right] \gtrsim m\omega q^{-\beta}(1 - (2\theta-1)q^{-\beta}\sqrt{n\omega}).$$

Take $\omega = \frac{q^{2\beta}}{4n(2\theta-1)^2}$ and $m = q^p$, we obtain

$$\sup_{\pi \in \mathcal{P}_{\gamma,\beta}} \mathbb{E}_{\widetilde{\mathcal{D}}} \left[ R(\widetilde{s}_{nn}) - R^* \right] \gtrsim \frac{q^{\beta+p}}{n\kappa_\epsilon^2} \asymp \left( \frac{1}{n\kappa_\epsilon^2} \right)^{\frac{(\gamma+1)\beta}{(\gamma+2)\beta+p}}$$

by taking $q \asymp \left( n(2\theta-1)^2 \right)^{\frac{1}{(\gamma+2)\beta+p}}$. Particularly, when $\epsilon \asymp n^{-1/2+\zeta}$, we have

$$\sup_{\pi \in \mathcal{P}_{\gamma,\beta}} \mathbb{E}_{\widetilde{\mathcal{D}}} \left[ R(\widetilde{s}_{nn}) - R^* \right] \lesssim \left( \frac{\log n}{n\kappa_\epsilon^2} \right)^{\frac{2\beta(\gamma+1)}{2\beta(\gamma+2)+p(\gamma+2)}} \asymp \left( \frac{\log(n)}{n^{2\zeta}} \right)^{\frac{2\beta(\gamma+1)}{2\beta(\gamma+2)+p(\gamma+2)}} \to 0 \text{ as } n \to \infty.$$

This completes the proof.

$\square$

