# OpenReview forum: "Binary Classification under Local Label Differential Privacy Using Randomized Response Mechanisms"
_TMLR — Accepted by TMLR_

### Review · Reviewer_3ueR · 2023-07-02

**Summary Of Contributions:**

1. the submission analyzed the impact of Randomized Response on the performance of a binary classifier, including both the Bayes optimal classifier and a constrained neural network.

2. the analysis also showcases the increase of errors in the estimation with local label privacy implemented through differential privacy.



**Audience:**

Yes

**Broader Impact Concerns:**

No concerns

**Claims And Evidence:**

Yes

**Requested Changes:**

I would recommend the authors think more toward class-imbalanced scenarios. For example, in Ads, the number of people who viewed or clicked on an ad is very rare and usually controlled in A/B testing, therefore, classifying every data sample as a negative sample would lead to very high accuracy, but the true positive rate would be low. Also in the medical domain, patients who are diagnosed with a certain disease or treated with a specific drug are sparse. Thus, it might be more interesting in that direction.

**Strengths And Weaknesses:**

1. the impact of the randomized response on the bayes opt classifier is essentially the same as the estimation error when the randomized response is used in surveys, therefore, I am not sure if the novelty really exists here.

2. I also think it is generally understood that reducing the privacy parameter --- epsilon --- would require us to design learning algorithms with fewer parameters, and in terms of neural networks, the structure of it should be simplified.

3. in binary classification, it might be more interesting to analyze the impact on the False Positive Rate and the True Positive Rate since often the two classes are imbalanced and the positive samples are rare. Simply looking at accuracy may not reflect the practical impact of the randomized responses.

4. In section 3.2, the authors claimed that the bayes decision rule would stay invariant to the choice of privacy parameters. However, it only would only stay the same when the positive and negative samples are balanced. Perhaps the authors meant to say that the excess risk would stay the same which is proven by Lemma 2.

---

### Review · Reviewer_vvPN · 2023-07-27

**Summary Of Contributions:**

The authors study use of randomized response to report binary labels under label differential privacy. In particular, they consider the impact of the privacy mechanism on downstream tasks like training neural networks to perform classification. They give several theoretical bounds on the impact on the convergence rate. Their theoretical bounds are supplemented with empirical studies on both synthetic and real datasets. The empirical results provide evidence for the authors 3 main hypotheses (driven by their theoretical results) that 1) for fixed epsilon, the model error converges to 0 as n increases, 2) eps=1/sqrt(n) is a dividing line, above which the model error does not converge to 0 and below which the error does converge to 0. While the first hypothesis is unsurprising, the stark evidence for the second is surprising and convincing.


**Audience:**

Yes

**Broader Impact Concerns:**

I have no ethical concerns for this work.

**Claims And Evidence:**

Yes

**Requested Changes:**

- Justify the use of randomized response as the local randomizer and why no debiasing is required.
- Provide intuition for proofs of Theorems 1 and 2 to make both the theorem statements and the proofs easier to parse.
- Either elaborate on or remove comments that imply that learning the estimator even under DP is a bad thing for privacy.

**Strengths And Weaknesses:**

Comments:
- I thought the experiments were interesting, well setup and nicely corroborated the theoretical results. I was particularly interested in the behavior as epsilon was varied with n. The stark change in the behavior between eps=1/log(n)sqrt(n), 1/sqrt(n) and logn/sqrt(n), while predicted by the theory was nice to see in practice. Figure 4 and Figure 7 were particularly nice.
- The authors fixed at the beginning of the paper that their DP algorithm was going to be to perform randomized response on the labels. They then analyzed the impact on downstream tasks of taking these labels at face value. I have two concerns about this setup:
    - The authors don’t give any justification for the use of binary randomized response in this context. It seems like a somewhat reasonable thing to do, but some justification is required.
    - When using the local reports in downstream tasks, to the best of my understanding, the authors did not use the fact they they knew the statistical disclosure method. For example, when using randomized response to estimate the mean of a Bernoulli, one does a debasing step since the expectation of the mean resulting from the responses using randomized response is not the mean of the original data. As the authors discuss in Lemma 2, since they are rounding some of this effect disappears, but it wasn’t clear to me that the presence of privacy noise should not be taken into account when defining the optimization problem.
- The authors make the comment “Lemma 2 theoretically validates the concern that privatized dataset generated by randomized response mechanism might still maintain utility for learning a consistent classifier in downstream binary classification tasks. Therefore, the estimated classifier can be used to infer the labels in the training dataset and leads to privacy leakage of the labels” and a similar comment later in the paper. These comments seem to indicate that the authors are conflating “learning statistical properties of the underlying dataset” with “violating privacy”. I would argue that the key insight of differential privacy is the argument that these are not the same. The privacy loss of each individual is approximately the same as if they had not participated in the data collection (since the classifier would still have been learnt with approximately the same accuracy). I encourage the authors to read the first chapter of the book “The Algorithmic Foundations of Differential Privacy” for more discussion.
- The main theorems Theorem 1 and Theorem 2 are quite technical and little intuition is given for their proof. In the experimental section the authors give some discussion about what these results imply about how the error trends with various parameters (particularly epsilon). However, it was difficult to absorb these theorems without some discussion of their proof.
- I did not read the technical proofs in detail but did not see any errors.

Minor Comments:
- The statement “generally the privacy protection in differential privacy is achieved by injecting noise implicitly to the raw data” is not really true, even in the local model.
- CDP is not a good acronym for central DP because it is more commonly used for concentrated DP.
- The authors briefly discuss an add/delete version of label DP described as “robust to the presence or absence of a single label”. I don’t think this makes sense? How can only the label be missing?
- Something is amiss about the definition of LLDP in Definition 1. This is just the definition of what it means for an algorithm with binary input and output to be eps-DP. The definition doesn’t mention labels at all. I think the authors are conflating a label DP mechanism (which should take (x,y) pairs as input) and the fact that one way to create a label DP mechanism is to release (x,f(y)) where f is eps-DP. On a minor note, I think the log should be inside the absolute value.
- My understanding is that Lemma 1 is really just a fact about the KL divergence between Bernoulli distributions? If so, I think it would be easier to parse if it was stated in this way, then applied to the specific setting.
- I found “the exponential term of the training size” to be confusing title. Perhaps just specifying that the training size was on a log scale would be easier to parse?
- Typo in the second line of the proof of Lemma 2
- My personal preference would be to have the theorem and lemma statements duplicated in the appendix to minimize flipping between the main body and the appendix. I although think it would be worthwhile to include more intuition in the proofs in the appendix, they are currently quite hard to read.

---

> ### Author Response · Authors · 2023-08-07
> **Author Response**
>
>
> We thank the reviewer for your detailed comments and useful advices. In our revision, we have made a number of changes that we hope will resolve your concerns (most changes are highlighted in blue for your convenience). Below are the answers to your comments and requested changes.
>
> **For requested changes:**
>
> **1. Justify the use of randomized response as the local randomizer and why no debiasing is required.**
>
> We appreciate the constructive suggestion provided by the reviewer. The randomized response mechanism was initially introduced for safeguarding the privacy of individuals who may be hesitant to reveal their true answers to certain questions during sample surveys [1]. Over time, this mechanism has found diverse applications, extending beyond surveys to include pairwise comparisons in ranking data [2] and handling edges in graph data [3,4]. We have expanded the discussion in Section 1 of the revised manuscript.
>
> As demonstrated in Lemma 2, we have established that the Bayes decision rule remains invariant to the randomized response mechanism, regardless of the privacy guarantee $\epsilon>0$ provided. This finding implies that, with a suitable model specification, it becomes feasible to obtain a differentially private classifier that approximates the performance of the Bayes classifier without debiasing. In other words, using perturbed labels for learning does not prevent the empirical risk minimization framework from learning a good classifier, where the debiasing procedure is not necessarily required. Similar results have been reported in [5], where it was shown that the Bayes classifier remains preserved even under certain label noise.
>
> **2. Provide intuition for proofs of Theorems 1 and 2 to make both the theorem statements and the proofs easier to parse.**
>
> We apologize for any lack of clarity in the proofs of Theorems presented in our initial submission. We have now included more demonstrations of Theorems 1 and 2. For the upper bounds in both Theorems 1 and 2, the proofs are derived by establishing a uniform concentration inequality. Compared with Theorem 1, the upper bound of Theorem 2 explicates the approximation error with respect to the structure of neural network, which allows us to make a tradeoff between estimation and approximation errors to achieve the fastest convergence rates. As for the lower bounds, the proofs mainly utilize the Assouad's lemma.
>
> **3.Either elaborate on or remove comments that imply that learning the estimator even under DP is a bad thing for privacy.**
>
> We thank the reviewer for pointing out our misunderstanding on the differential privacy and we have removed corresponding comments in the revised manuscript. We agree that learning a classifier based on a privatized dataset does not lead to privacy violation on labels, since it is widely accepted that statistical inference is not a privacy violation [6].
>
> **For minor comments:**
>
> We sincerely appreciate the reviewer for conducting a thorough examination of our manuscript. We have implemented the following changes to address your minor comments:
>
> * We acknowledge that privacy protection in differential privacy is not solely achieved through noise injection to raw data. Thus, we have removed the statement "generally the privacy protection in differential privacy is achieved by injecting noise implicitly to the raw data."
>
> * To enhance clarity, we now use "central DP" as an abbreviation for "central differential privacy."
>
> * We acknowledge your correct understanding of Lemma 1, which presents the KL divergence between Bernoulli distributions under the setting of differential privacy. We have revised it accordingly.
>
> * We have revised the titles of Figures 2-4 to "The training size on a log scale" for better clarity.
>
> * We have corrected typos in the manuscirpt.
>
> * To facilitate the reviewing process, we have included the main theorems and lemmas in the appendix. Furthermore, we have supplemented both the main text and the appendix with comprehensive explanations and intuitive insights into the proofs of our theoretical results.
>
> **References**
>
> [1] WARNER, Stanley L. Randomized response: A survey technique for eliminating evasive answer bias. Journal of the American Statistical Association, 1965, 60.309: 63-69.
>
> [2] LI, Zhechen, et al. Differentially private condorcet voting. In: Proceedings of the AAAI Conference on Artificial Intelligence. 2023. p. 5755-5763.
>
> [3] HEHIR, Jonathan; SLAVKOVIC, Aleksandra; NIU, Xiaoyue. Consistent Spectral Clustering of Network Block Models under Local Differential Privacy. Journal of Privacy and Confidentiality, 2022, 12.2.
>
> [4] GUO, Xiao, et al. Privacy-Preserving Community Detection for Locally Distributed Multiple Networks. arXiv preprint arXiv:2306.15709, 2023.
>
> [5] CANNINGS, Timothy I.; FAN, Yingying; SAMWORTH, Richard J. Classification with imperfect training labels. Biometrika, 2020, 107.2: 311-330.
>
> [6] https://differentialprivacy.org/inference-is-not-a-privacy-violation/

---

> > ### Comment · Reviewer_vvPN · 2023-08-14
> > **Thank you**
> >
> > Thank you for your response. I wanted to clarify my question about randomized response since one of the other reviewers had the same question. I think the use of a local DP is motivated, my question is why use randomized response over other LLDP mechanisms? If I remember correctly, randomized response can be claimed to be optimal in some settings, but I didn't see a discussion of why it was a good choice for this problem?

---

> > > ### Author Response · Authors · 2023-08-26
> > > **Response to Reviewer**
> > >
> > > We thank the reviewer for pointing out an important property of the randomized response mechanism. We have added some discussion about this in the revised manuscript.
> > >
> > > **Response to your question**:
> > >
> > > The randomized response indeed offers some optimal statistical properties. As discussed in depth by [1], the RR mechanism exhibits smaller expected mean square errors between released and actual values compared to the Laplace mechanism. Furthermore, the effectiveness of the RR mechanism surpasses that of the output perturbation approach, particularly in scenarios where the sensitivity of output functions is high. This result can be understood from the perspective of statistical hypothesis testing [2]. Specifically, we can consider the following hypothesis testing problem:
> > > $$
> > > H_0: \widetilde{y}=\mathcal{A}(y) \mbox{ v.s. }
> > > H_1: \widetilde{y}=\mathcal{A}(1-y)
> > > $$
> > > where $y \in$ {-1,1} represents the real label and $\widetilde{y} \in$ {-1,1} represents the observed label. With $\epsilon$-DP, the Type-I and Type II errors are both $err = 1/(1+e^{\epsilon})$ if we treat $\widetilde{y}$ as real label. Then the tradeoff function of RR mechanism (the curve connecting (0,1), (1,0), and (err,err)) perfectly matches the optimal tradeoff between type-I and type-II errors under $\epsilon$-DP as demonstrated in [3].
> > >
> > >
> > > In the context of label differential privacy (label-DP), both central-DP and label-DP mechanisms exist. For instance, the classical RR is integrated with deep learning in a study found at [4], while another paper [5] introduces PATE (central-DP) and ALIBI (local-DP). The current state-of-the-art approach, outlined in [6], presents a variant of the RR mechanism that incorporates additional information from the loss function to enhance model performance. Analyzing this variant can be challenging due to its construction involving the solution of an optimization problem without a closed-form representation. To address this, we begin by examining the classical RR, which has a succinct representation, in order to derive certain theories related to deep learning.
> > >
> > > [1] WANG, Yue; WU, Xintao; HU, Donghui. Using Randomized Response for Differential Privacy Preserving Data Collection. In: EDBT/ICDT Workshops. 2016. p. 0090-6778.
> > >
> > > [2] WASSERMAN, Larry; ZHOU, Shuheng. A statistical framework for differential privacy. Journal of the American Statistical Association, 2010, 105.489: 375-389.
> > >
> > > [3] DONG, Jinshuo; ROTH, Aaron; SU, Weijie. Gaussian Differential Privacy. Journal of the Royal Statistical Society, 2021.
> > >
> > > [4] GHAZI, Badih, et al. Deep learning with label differential privacy. Advances in neural information processing systems, 2021, 34: 27131-27145.
> > >
> > > [5] MALEK ESMAEILI, Mani, et al. Antipodes of label differential privacy: Pate and alibi. Advances in Neural Information Processing Systems, 2021, 34: 6934-6945.
> > >
> > > [6] GHAZI, Badih, et al. Regression with Label Differential Privacy. arXiv preprint arXiv:2212.06074, 2022.

---

> > > > ### Comment · Reviewer_vvPN · 2023-09-18
> > > > **Debiasing?**
> > > >
> > > > Thank you for your response! Just another question about the use of RR. Why do you not need a debiasing step as you would when using RR to estimate a mean?

---

> > > > > ### Comment · Reviewer_vvPN · 2023-09-18
> > > > > **Definition of label DP**
> > > > >
> > > > > Just another comment: the definition of label DP still doesn't match the definition given in Ghazi et al., which allows for a potentially larger class of mechanisms. I would prefer the authors give the general definition, then describe that one way to satisfy this definition is to randomise the labels independent of the dependent variable x.

---

> > > > > > ### Author Response · Authors · 2023-09-22
> > > > > > **Definition of Label DP**
> > > > > >
> > > > > > We thanks the reviewer for pointing out this for improving the quality of this paper. We have corrected the definition of Label DP in the re-submitted paper, and denote the $\epsilon$-label DP achieved by the randomized response mechanism as $\epsilon$-local label DP.

---

> > > > > ### Author Response · Authors · 2023-09-21
> > > > > **Author Response for Debiasing**
> > > > >
> > > > > Thank you for your comment. In our paper, we propose the use of privatized labels for learning without the need for a debiasing step. This approach is grounded in a two-fold rationale:
> > > > >
> > > > > Firstly, if we debias the labels in order to obtain an unbiased estimate of the mean, the resulting debiased label is given by:
> > > > > $$
> > > > > Z = \frac{\widetilde{Y}(e^{\epsilon}+1)-1}{e^{\epsilon}-1}
> > > > > $$
> > > > > It is important to note that the debiased label is not 0-1 binary, offering a more nuanced representation.
> > > > >
> > > > > Secondly, the Bayes classifier remains invariant to the randomized response mechanism, enabling the learnability of the optimal classifier with the appropriate choice of function class. To illustrate this concept, consider an intuitive yet somewhat inaccurate example: envision two random walks—one with an 80% chance of +1 and a 20% chance of -1, and the other with a 60% chance of +1 and a 40% chance of -1. As these two random walks take many steps, it becomes evident that both random walks with exhibit higher probability of +1.
> > > > >
> > > > > I appreciate your feedback, and you've raised an important point. There is indeed an implicit debiasing step in our paper, as illustrated in Lemma 2. This step is crucial for calculating the excess risk under the real distribution $P(X,Y)$ of the differentially private classifier $\widetilde{f}$ by first debiasing the excess risk under the private distribution $P(X,\widetilde{Y})$. Specifically, under $\epsilon$-Local Label DP (where $\theta = e^{\epsilon}/(e^{\epsilon}+1)$), we get
> > > > > $$
> > > > > \widetilde{R}(\widetilde{f}) - \widetilde{R}(f^*) =
> > > > > \frac{e^{\epsilon}-1}{e^{\epsilon}+1}
> > > > > \left(
> > > > > R(\widetilde{f}) - R(f^*)
> > > > > \right).
> > > > > $$
> > > > > Here $\widetilde{R}(\widetilde{f})$ represents the generalization performance of the differentially private classifier $\widetilde{f}$ on label-privatized data. This equality provides us with a means to deduce the generalization performance of $\widetilde{f}$ on real data by effectively eliminating the noise introduced by the labels, thus allowing us to obtain the generalization of $\widetilde{R}(\widetilde{f})$.

---

### Review · Reviewer_7nL3 · 2023-08-13

**Summary Of Contributions:**

The paper studies the impact of local label DP on the excess risk of binary classification, when the local label DP is achieved by randomized response method. The paper proves consistency of the classifier trained on the LLDP-preserving labels. The analysis shows that the excess risk under LLDP still converges but application of LLDP slows down the convergence depending on the privacy level $\epsilon$. The result is further instantiated for a neural network of finite width and composed of ReLU activation nodes. The additional assumption is that the NN belongs to a bounded $\beta$-Holder space. The analysis shows that (a) the size of the neural network should be smaller for higher privacy level to ensure generalization and (b) the convergence of the excess risk is achievable if $\epsilon$ decreases at a rate slower than $O(1/\sqrt{n})$. The results are further verified experimentally.


**Audience:**

Yes

**Broader Impact Concerns:**

The paper rigorously shows an intuitive phenomenon that having label DP would slower down the convergence of the classifier. The other interesting result is that as privacy barrier increases the NN should have smaller depth to ensure better generalisation. These results are interesting though the problem of interest seems very specific from the present draft.

**Claims And Evidence:**

Yes

**Requested Changes:**

Addressing the weaknesses will be appreciated.

**Strengths And Weaknesses:**

**Strengths**

1. The LLDP and the learning theoretic quantities are well introduced formally.

2. The presented results are rigorously proved.

3. The theoretical findings are experimentally verified.


**Weaknesses**

1. Motivation of the setup: It's very hard to understand why studying such a specific setup. At least the related work does not make it evident to me. Why specifically local DP is interesting here not central DP? Why studying randomised response as the privacy preserving mechanism? May be I am missing some context which I will like to be introduced to.

2. Proof sketches: Addition of proof sketches have improved the readability of the work significantly. But still it is not evident what is non-trivial or challenging in the analysis? Pointing out such challenges would make the proof more appreciable. Also, in the supplementary proof steps are often skipped or sparsely explained. Explaining proofs in the appendix would be very helpful.

3. Second part of Theorem 2: As a corollary of Thm 2, it is stated that the convergence of the excess risk is achievable if $\epsilon$ decreases at a rate slower than $O(1/\sqrt{n})$. But no proof of it is provided in the paper. A proof should be added.

4. Excess EFNE and EFPE results under LLDP: The impact of class imbalance is shown for normal classifier but not the one trained with LLDP labels. It is said that the DP classifier also satisfies the same rate but a proof is needed to claim this result.

5. Holder space: Can you give two examples (its okay even if it is a generalised linear model) where the Holder's assumption does and does not hold?

---

> ### Author Response · Authors · 2023-08-26
> **Response to Reviewer**
>
> We thank the reviewer for your constructive comments and advices. In our revision, we have made some changes that we hope will resolve your concerns. Below are the point-by-point  responses to the weakness.
>
> 1. The randomized response (RR) is a traditional manner to achieve local DP since it offers some optimal statistical properties. This topic has been discussed in depth by [1]. The optimality can be viewed from the perspective of statistical hypothesis testing that RR achieves the optimal trade-off between type I and type II errors under $\epsilon$-DP [2]. Basically, label DP can be achieved in both central and local manners. For instance, the classical RR is integrated with deep learning in a study found at [3], while [4] introduces PATE (central-DP) and ALIBI (local-DP). The current state-of-the-art approach, outlined in [5], presents a variant of RR that incorporates additional information from the loss function to enhance model performance. Analyzing this variant can be challenging due to its construction involving the solution of an optimization problem without a closed-form representation. To study the inherent effects of privacy on generalization, we begin by examining the classical RR, which has a succinct representation, in order to derive certain theories related to deep learning. We have added more discussions about the motivation in the revised manuscript.
>
> 2. We extend our gratitude to the reviewer for providing invaluable advices. Within our theoretical framework, we have rigorously established key insights concerning the generalization performance of classifiers trained on privatized datasets. A pivotal challenge in our context arises from the inherent disparity between the distributions of training and test data. Addressing this challenge, we have effectively established a significant link (Lemma 2) between the excess risk (under the training data distribution) and the corresponding excess risk (under the test data distribution). This linkage serves as a vital bridge, effectively transposing classical classification results into the domain of our specific problem. We have added more discussions in the revised manuscript. Additionally, for improving the readability of our manuscript, we tried our best to augment the proofs of Theorem 1 and 2 for enhanced clarification.
>
> 3. We thank the reviewer for pointing this out. We have added a detailed proof of the corollary of Theorem 2 in the revised manuscript..
>
> 4. We apologize for the confusion regarding the results for EFNE and EFPE (Lemma 4). We would like to clarify that the relationship established between excess EFNE (or excess EFPE) and excess risk, as outlined in Lemma 4, is applicable to *all classifiers*, denoted as $f$ in our context. Notably, this result also encompasses the differentially private classifier, $\widetilde{f}$, *as a specific case*. With this finding, the theoretical results about the convergence rate of the excess risk exhibited by $\widetilde{f}$ can be seamlessly extrapolated to the asymptotic behavior of excess EFNE and EFPE. We have further augmented the explanations for Lemma 4 in our revised manuscript.
>
> 5. The Holder assumption is a common assumption for analyzing the generalization performance of neural networks [6,7]. The Holder space is defined as
> $\mathcal{H}(\beta,\mathcal{X}) = \lbrace f \in \mathcal{C}^{\lfloor \beta\rfloor}(\mathcal{X}):\Vert f \Vert_{\mathcal{H}(\beta,\mathcal{X})}<\infty\rbrace.$
> Functions in the Holder space exhibit a certain level of smoothness (measured by $\beta$) and regularity (bounded Holder norm). Take $\beta=1$ and $\mathcal{X}=[0,1]^p$ for example, $\mathcal{H}(1,[0,1]^p)$ consists of all functions with continous partial derivatives and bounded Holder norm. Clearly, two example can be derived as follows:
>
> (1) When $\eta(x)=(sin(\sum_{i=1}^p x_i)+1)/2$, the Holder assumption holds.
>
> (2) When $\eta(x)=\sum_{i=1}^p |x_i|/p$, the Holder assumption does not hold.
>
> **References**
>
> [1] WANG, Yue; WU, Xintao; HU, Donghui. Using Randomized Response for Differential Privacy Preserving Data Collection. In: EDBT/ICDT Workshops. 2016. p. 0090-6778.
>
> [2] DONG, Jinshuo; ROTH, Aaron; SU, Weijie. Gaussian Differential Privacy. Journal of the Royal Statistical Society, 2021.
>
> [3] GHAZI, Badih, et al. Deep learning with label differential privacy. Advances in neural information processing systems, 2021, 34: 27131-27145.
>
> [4] MALEK ESMAEILI, Mani, et al. Antipodes of label differential privacy: Pate and alibi. Advances in Neural Information Processing Systems, 2021, 34: 6934-6945.
>
> [5] GHAZI, Badih, et al. Regression with Label Differential Privacy. arXiv preprint arXiv:2212.06074, 2022.
>
> [6] Jean-Yves Audibert. Alexandre B. Tsybakov. "Fast learning rates for plug-in classifiers." Ann. Statist. 35 (2) 608 - 633
>
> [7] Schmidt-Hieber, Anselm Johannes. Nonparametric regression using deep neural networks with ReLU activation function. Annals of statistics, 2020, 48.4: 1875-1897.

---

### Review · Reviewer_LvK2 · 2023-09-30

**Summary Of Contributions:**

Label differential privacy was introduced by Hsu and Chaudhari and was implicit in that paper. It has been extended to the local setting by Ghazi et al. recently. For privacy, they use randomized response, which seems intuitive (though I am not convinced it is the optimal one). They give extensive experiments to support their theoretical results.

**Audience:**

Yes

**Broader Impact Concerns:**

None.

**Claims And Evidence:**

Yes

**Requested Changes:**

Give some high-level overview of the result and some intuition why the mechanism is the right choice.

Some discussion on high-privacy regime and why the authors think that there is a sharp threshold where you have model error converging to zero or not.

Unless I am mistaken, the definition of local label DP needs to be checked.

**Strengths And Weaknesses:**

My apologies for such a long delay in submitting the reviews. At a high level, I liked reading the paper. The paper contains a good mix of theoretical as well as experimental results. I was unable to through the entire appendix (even with such a delay) and I trust the authors have done a good job of verifying their results.

In the sense of the main result of the paper, it seems like the paper shows that if we aim for a really low privacy regime ${1 \over \sqrt{n}}$, then we do not have the error to go to $0$. I really did not get the intuition why for an even smaller privacy regime, we have that error that goes to $0$. Perhaps, the authors can clarify it. If they have already clarified it in a revision, then it is great. Also, why would one care for such a low privacy regime? Is there a practical setting where one would consider such a low-privacy regime?

I am not entirely sure that the local label DP definition is the correct one. It is based on my one-time look at the definition given in Ghazi, Golowich, Kumar, Manurangsi, Zhang. Perhaps, I am missing something here. Can the authors inform me that the definition is really the same?

To parse the main theorem and get an intuition as to why it should be true, I really had to go back and forth a lot. Perhaps, the authors can give me a high-level overview of how they arrived at their results. It would also help to understand what are the technical challenges in arriving at their result. Honestly, I do not care that much about the technical lemmas (which end up being just the maths to validate the result), but more about the intuition.

Overall, the paper's result are definitely interesting and deserves to be accepted.

---

> ### Author Response · Authors · 2023-10-03
> **Author Response**
>
> Thank you for your valuable comments and insightful suggestions aimed at enhancing the quality of this paper. To begin with, we would like to clarify that smaller $\epsilon$ actually indicates a stronger privacy. The following are our point-by-point responses for your questions.
>
> **1.** The randomized response is a traditional local differential privacy (local-DP) randomized mechanism that offers some optimal statistical properties. This topic has been discussed in depth by [1]. By setting a privacy budget of $\epsilon$, it is possible to create a randomized response mechanism that attains optimal utility.
>
> Additionally, when viewed from the perspective of statistical hypothesis testing [2], which develops the $f$-DP framework, RR accomplishes the optimal trade-off between type I and type II errors under $\epsilon$-DP. Precisely, let $\mathcal{A}$ be an $\epsilon$-DP local randomizer. The $\epsilon$-DP definition is equivalent to the hypothesis testing problem $$H_0: \mathcal{A}(y)=\widetilde{y} \quad v.s. \quad H_1:\mathcal{A}(y')=\widetilde{y}$$ for two different data points $y,y'$.
> Then, according to [2], the type I error $\alpha$ and the type II error $\beta$ satisfy the trade-off function (flipped ROC curve)
> $$
> \beta(\alpha) = \max(0, 1 - e^{\epsilon}\alpha, e^{-\epsilon}(1 - \alpha)).
> $$
> This trade-off function is matched by the randomized response algorithm. In contrast, the Laplace mechanism fails to perfectly match this trade-off function, as demonstrated in [2]. **This result means that using the $\epsilon$ in the randomized response mechanism is equivalent to using Type-I and Type-II errors to measure privacy protection, which are more practical in real-life applications.**
>
> **2.** In our paper, we demonstrate that $\epsilon=\sqrt{n}$ serves as a critical threshold for the convergence of our estimated model. From a high-level perspective, this phenomenon arises because the increased sample size contributes more information for model estimation, which is counterbalanced by the information loss incurred due to the enhanced privacy protection afforded by the randomized response mechanism. To illustrate this concept, we offer a similar yet somewhat imprecise example. Nevertheless, the underlying intuition in this example bears resemblance to the findings we present in our paper. Let $(x_i)_{i=1}^n$ be a set of sample following the normal distribution with mean $\mu$ and variance $\sigma^2$. By the law of large number, we have $\overline{x}=n^{-1}\sum x_i$ converges to $\mu$ in probability, and the variance of $\overline{x}=\sigma^2/n$. If $\sigma$ is dependent on the sample size and diverges at the order $\sqrt{n}$, then $\overline{x}$ fails to converge to $\mu$. Similarly, in our results, if $\epsilon=\sqrt{n^{-1}}$ (stronger privacy as $n$ increases), then the noise incurred by privacy protection is exactly offset by the increased sample size.
>
> **3.** We would like clarify the definition of the original $\epsilon$-Differential Privacy ($\epsilon$-DP). This definition can be stated as follows: In the context of two neighboring datasets, denoted as $D$ and $D'$ (differing only by a single data point), a randomized mechanism denoted as $\mathcal{A}$ is considered $\epsilon$-Differentially Private if, for any outcome $s$ in the output space $S$, the following inequality holds:
> $$
> \max_{s \in S}
> \Big|
> \log\left(
> \frac{P(\mathcal{A}(D) = s)}{P(\mathcal{A}(D') =s)}\right) \Big|\leq \epsilon.
> $$
> The operation of the randomized mechanism $\mathcal{A}$ involves processing a dataset $D$ and subsequently releasing an output $s$ to attackers, which may include potential privacy attackers. It is crucial to emphasize that this process enables attackers to make inferences about the presence or absence of a specific individual record within the dataset. A lower value of $\epsilon$ corresponds to a stronger privacy guarantee, indicating that the released output $s"$ poses greater challenges for potential attackers in their attempts to deduce the existence of a specific individual record within the dataset $D$.
>
> **Notice that the randomized response mechanism is applied to each data point indenepdently**. Therefore,
> $$
> P(\mathcal{A}(D) = s) =
> P(\mathcal{A}(y_1) = s_1,\mathcal{A}(y_1) = s_2,\ldots,\mathcal{A}(y_n) = s_n),
> $$
> where $s=(s_1,\ldots,s_n)$. Without loss of generality, we suppose $D$ and $D'$ differ in the first instance. Then
> $$
> \frac{P(\mathcal{A}(D) = s)}{P(\mathcal{A}(D') =s)}=
> \frac{P(\mathcal{A}(y_1) = s_1)}{P(\mathcal{A}(y_1') =s_1)},
> $$
> which leads to the definition of $\epsilon$-local DP. **This relation shows that $\epsilon$-local DP is a way to achieve the original $\epsilon$-DP.**
>
> **Reference**
>
> [1] WANG, Yue; WU, Xintao; HU, Donghui. Using Randomized Response for Differential Privacy Preserving Data Collection. In: EDBT/ICDT Workshops. 2016. p. 0090-6778.
>
> [2] DONG, Jinshuo; ROTH, Aaron; SU, Weijie. Gaussian Differential Privacy. Journal of the Royal Statistical Society, 2021.

---

### Author Response · Authors · 2023-10-22
**Thanks for your efforts and a Camera-Ready version is uploaded**

We would like to express our gratitude to the reviewers and editors for their invaluable contributions to reviewing and improving our manuscript. Your constructive feedback and evaluation have played a key role in enhancing the quality and rigor of our research. We truly appreciate their time and expertise. We have uploaded a camera-ready version of our manuscript and attached a link consisting of Python codes for numerical experiments in our paper.

---

### Decision · Action_Editor_z5ZJ · 2023-10-20

**Recommendation:** Accept as is

**Comment:**

The reviewers generally agreed that the results are interesting and sound, albeit focused on a very narrow setting (label DP with randomized response mechanism for downstream binary classification).

**Audience:**

Yes, there is a sizeable community interested in differential privacy

**Claims And Evidence:**

The paper's theoretical claims appear valid according to the reviewers, and are complemented by empirical experiments that nicely match the theoretical predictions